# The Effect of Personalization in FedProx: A Fine-grained Analysis on Statistical Accuracy and Communication Efficiency

## Abstract

FedProx is a simple yet effective federated learning method that enables model personalization via regularization. Despite remarkable success in practice, a rigorous analysis of how such a regularization provably improves the statistical accuracy of each client's local model hasn't been fully established. Setting the regularization strength heuristically presents a risk, as an inappropriate choice may even degrade accuracy. This work fills in the gap by analyzing the effect of regularization on statistical accuracy, thereby providing a theoretical guideline for setting the regularization strength for achieving personalization. We prove that by adaptively choosing the regularization strength under different statistical heterogeneity, FedProx can consistently outperform pure local training and achieve a ~~nearly~~ *minimax-optimal* statistical rate. In addition, to shed light on resource allocation, we design an algorithm, provably showing that stronger personalization reduces communication complexity without increasing the computation cost overhead. Finally, our theory is validated on both synthetic and real-world datasets and its generalizability is verified in a non-convex setting.

## 1 Introduction

Federated Learning (FL) has emerged as an attractive framework for aggregating distributed data, enabling clients to collaboratively train a shared global model while preserving data privacy. In the currently prevalent paradigm (McMahan et al., 2017), FL is formulated as a finite sum minimization problem focusing on a single shared model. Nevertheless, it has been well recognized that one of the key challenges in FL is the statistical heterogeneity of the client datasets. As the participants collect their own local data, it often reflects client-specific characteristics and is not identically distributed. With high statistical heterogeneity, training a single model for all clients by minimizing their average in-sample loss becomes questionable.

To address this challenge, one solution is to relax the common model constraint and solve alternatively the following objective in FedProx (Li et al., 2020):

$$\min_{\boldsymbol{w}^{(g)}, \{\boldsymbol{w}^{(i)}\}_{i \in [m]}} \sum_{i \in [m]} p_i \left( L_i(\boldsymbol{w}^{(i)}, S_i) + \frac{\lambda}{2} \|\boldsymbol{w}^{(g)} - \boldsymbol{w}^{(i)}\|^2 \right), \tag{1}$$

where $L_i(\boldsymbol{w}^{(i)}, S_i)$ is the empirical risk of the $i$-th client, and $S_i$ is the local dataset of the $i$-th client. The set $\{p_i\}_{i \in [m]}$ is a collection of weights for the clients satisfying $p_i \geq 0$ and $\sum_i p_i = 1$. In (1), each client $i$ learns a local personalized model $\boldsymbol{w}^{(i)}$ by fitting its local data $S_i$, while enabling collaborative learning by shrinking all local models towards a common global one $\boldsymbol{w}^{(g)}$ with intensity controlled by $\lambda$. The smaller $\lambda$ is, the weaker the coupling of the local models the formulation would enforce thus the higher personalization is. Adjusting the strength of regularization controls the extent to which the $\boldsymbol{w}^{(i)}$'s is adapted to client $i$'s local data, thus the degree of personalization.

The promising empirical performance of FedProx in learning under heterogeneous data (Li et al., 2020; Donevski et al., 2021; He et al., 2021; An et al., 2023) has led to a sequence of works studying the problem from a pure computation perspective, aiming at developing efficient algorithms under various problem assumptions to reach a (local) solution (Hanzely & Richtárik, 2020; T Dinh et al.,

2020; Li et al., 2021; Acar et al., 2021; Yuan & Li, 2022). However, the fundamental question of how to characterize the accuracy of the models produced by (1) is largely unaddressed. As such, for a given set of clients, it is not even clear if introducing personalization will be beneficial, outperforming the baselines given by training a single global model or pure local training. In addition, communication cost in FL is a critical bottleneck in resource-constrained environments. Therefore, understanding when personalization will improve both training efficiency and statistical accuracy is of paramount importance. We address these questions by providing a fine-grained analysis of the effect of personalization in FedProx. Our contribution is summarized below:

• **Minimax-optimal Statistical Accuracy**. We analyze the statistical accuracy of the objective in (1), taking statistical heterogeneity into account. For each of the models, we establish a state-of-the-art convergence rate $\mathcal{O}(1/(mn)) + \mathcal{O}(1/n \wedge R^2)$, where $R$ characterizes the statistical heterogeneity and $n$ is the sample size on each client. The first term $\mathcal{O}(1/(mn))$ represents the sample complexity of training a single global model under homogeneity, which is attained by (1) when $R$ is close to zero, highlighting the benefit of FL. For the second term, the first component $\mathcal{O}(1/n)$ is the sample complexity of purely local training, and the second component $\mathcal{O}(R^2)$ demonstrates the flexibility of FedProx under different statistical heterogeneity, implying the statistical accuracy of FedProx never gets worse than purely local training's. Such a rate matches the existing lower bound for the problem. Therefore, to the best of our knowledge, this is the first work that demonstrates that with a theoretically-informed adaptive choice of personalization degree characterized by $\lambda$, FedProx achieves a *minimax-optimal* statistical accuracy across different levels of statistical heterogeneity.

• **Provably Enhanced Communication Efficiency**. As the objective (1) could be reformulated as a bi-level problem, we propose an algorithm for (1) with an explicit computation cost assessment. We establish the communication complexity as $\mathcal{O}(\kappa \frac{\lambda+\mu}{\lambda+L} \log \frac{1}{\varepsilon})$ and the local computation complexity in each communication round as $\widetilde{\mathcal{O}}(\frac{\lambda+L}{\lambda+\mu})$. With higher personalization degree, communication complexity would decrease and the computation complexity per round would increase, demonstrating that there is a trade-off between computation and communication costs. Our analysis concludes that model personalization can reduce communication overhead without incurring additional computation expenses.

• **Empirical Validation**. We conduct numerical studies on logistic regression with synthetic data and real-world data under our problem assumptions. The results corroborate our theoretical findings. We further test the numerical performance of (1) for training CNNs, showing empirically the results, although proved under the convexity assumption, generalize to nonconvex losses.

## 2 RELATED WORKS

In recent years, federated learning has become an attractive solution for training models locally on distributed clients, rather than transferring data to centralized data centers (Mammen, 2021; Wen et al., 2023; Beltrán et al., 2023). As each client generates its local data, statistical heterogeneity is common with data being non-identically distributed between clients (Li et al., 2020; Ye et al., 2023). Given the variability of data in network, model personalization is a natural strategy used to improve learning accuracy for each client. Formulations enabling model personalization have been studied independently from multiple fields. For example, meta-learning (Chen et al., 2018; Jiang et al., 2019; Khodak et al., 2019; Fallah et al., 2020) assumes all models follow a common distribution. By minimizing the average loss, these methods aim to learn a meta-model that generalizes well to unseen new tasks. Multi-task learning methods (Agarwal et al., 2020; Liang et al., 2020) and transfer learning methods (Li et al., 2022; He et al., 2024a;b) study formulation similar to (1), but the statistical rate is established for either the average of all models or only the target model. Therefore, it is not clear if all clients can benefit from the procedure and learn a more accurate model than purely local training. Representation learning (Zhou et al., 2020; Wang et al., 2024), on the other hand, focuses on a different model structure where the local models can be represented as the composition of two parts, with one common to all clients and the other specific to each client.

In the FL community, personalized FL methods can be broadly divided into two main strands. Different from the works mentioned previously, studies here focus primarily on the properties of the iterates computed by the algorithms. One line of works (Arivazhagan et al., 2019; Liang et al., 2020; Singhal et al., 2021; Collins et al., 2021) is based on the representation learning formulation, where

both the algorithmic convergence and statistical accuracy of the iterates have been investigated. Another line of works (Smith et al., 2017; Li et al., 2020; Hanzely & Richtárik, 2020; Hanzely et al., 2020; 2023), including FedProx, achieves personalization by relaxing the requirement of learning a common global model through regularization techniques. In particular, algorithms and complexity lower bounds specific to the FedProx objective were studied in works (Li et al., 2020; Hanzely et al., 2020; T Dinh et al., 2020; Hanzely & Richtárik, 2020; Li et al., 2021), see Table 5 for a detailed comparison. These works study the problem from a pure optimization perspective and cannot show personalization improves learning accuracy.

There are only a few recent works we are aware of studying the statistical accuracy of regularization-based PFL in the form of (1). Specifically, Cheng et al. (2023) investigates the asymptotic behavior of the FedProx estimator under an over-parameterized linear regression model; neither a finite-sample rate nor algorithm is provided. Chen et al. (2023c) studies the FedProx estimator under rather strong assumptions like bounded parameter space. By defining a new notion of algorithmic stability, they establish the non-asymptotic sample complexity for FedProx and induce minimax bound. However, even with some overly strong assumptions, their analysis cannot match the statistical lower bound in most cases.

**Notation.** $[n]$ denotes the set of integers $\{1, 2, \cdots, n\}$ for any positive integer $n$. Let $\|\cdot\|_2$ denote the $\ell_2$ vector norm and the Frobenius matrix norm. For two non-negative sequences $a_n$, $b_n$, we denote $a_n = \mathcal{O}(b_n)$ as $a_n/b_n \to 1$. $\tilde{\mathcal{O}}(\cdot)$ hides logarithmic factors.

## 3 Preliminaries

Considering there are $m$ clients, each client has a collection of $i.i.d$ data $S_i = \{z_{ij}\}_{j \in [n_i]}$ over possibly different data distributions $\mathcal{D}_i$. For each client, the ultimate goal is to find the solution that minimizes the local risk, i.e. $\boldsymbol{w}_\star^{(i)} \in \arg\min_{\boldsymbol{w}} \mathbb{E}_{z \sim \mathcal{D}_i} \ell(\boldsymbol{w}, z)$. Here, $\ell(\boldsymbol{w}, z)$ denotes the loss function of interest and $\boldsymbol{w}_\star^{(i)} \in \mathbb{R}^d$ is called the ground truth local model for client $i$. In this paper, we assume the model dimension $d$ is finite. Under such a setup, two commonly adopted approaches are pure local training (LocalTrain) and pure global training (GlobalTrain):

$$\text{LocalTrain:} \quad \min_{\boldsymbol{w}^{(i)}} L_i(\boldsymbol{w}^{(i)}, S_i), \ i \in [m], \qquad \text{GlobalTrain:} \quad \min_{\substack{\boldsymbol{w}^{(i)} = \boldsymbol{w}^{(g)}, \\ i \in [m]}} \sum_{i=1}^m p_i L_i(\boldsymbol{w}^{(i)}, S_i), \ (2)$$

where $L_i(\boldsymbol{w}, S_i) := \sum_{j=1}^{n_i} \ell(\boldsymbol{w}, z_{ij})/n_i$ denotes the empirical risk of client $i$. In LocalTrain, each client individually minimizes its loss, whereas in GlobalTrain, all clients collaboratively minimize their average loss. Compared to LocalTrain, GlobalTrain integrates information from all clients to improve the learning performance of each client. When there are sufficient similarities between the data distribution $\mathcal{D}_i$ of different clients, GlobalTrain would be a better choice as more samples are included. However, if $\mathcal{D}_i$'s are highly unrelated, LocalTrain will be a more robust choice.

In practice, the data distribution is neither totally the same nor completely unrelated across clients. To handle the wide middle ground of the two extremes, FedProx solves the following problem

$$\min_{\boldsymbol{w}^{(g)}, \{\boldsymbol{w}^{(i)}\}_{i \in [m]}} \sum_{i=1}^m p_i \left( L_i(\boldsymbol{w}^{(i)}, S_i) + \frac{\lambda}{2} \|\boldsymbol{w}^{(g)} - \boldsymbol{w}^{(i)}\|^2 \right).$$

The key feature of Fedprox is the incorporation of a proximal (regularization) term, which controls the personalization degree. As $\lambda \to 0$, the FedProx objective increasingly behaves like LocalTrain with higher personalization degree and reduces to $m$ separate LocalTrain problems in the end. On the other hand, as $\lambda \to \infty$, the local models $\boldsymbol{w}^{(i)}$ are shrunk towards the global model $\boldsymbol{w}^{(g)}$ with lower personalization degree and eventually degenerate to optimize a single shared model. Thus, by adjusting $\lambda$, FedProx interpolates between LocalTrain and GlobalTrain, allowing for a flexible choice of personalization degree to achieve higher modeling accuracy. It's noticeable that this choice of $\lambda$ also affects the optimization process of FedProx. With larger $\lambda$, more frequent communication between clients is expected to ensure faster convergence, as local updates are more tightly coupled to the global model. In contrast, smaller $\lambda$ value allows for more independent updates, reducing the need for communication. In the next few sections, we provide a theoretical guarantee for the intuition provided above.

In Section 4, we analyze when the personalization level, characterized by $\lambda$, would improve the statistical accuracy of both the global model $\boldsymbol{w}^{(g)}$ and the local models $\{\boldsymbol{w}^{(i)}\}_{i \in [m]}$. This is in sharp contrast to most of the existing works where the error is established for the sum of them. In Section 5, we explore how such a degree of personalization will affect communication complexity.

# 4 WHEN WILL PERSONALIZATION IMPROVE STATISTICAL ACCURACY

In this section, we analyze how the choice of $\lambda$, or the degree of personalization, would improve the statistical accuracy of the solution obtained from the FedProx problem (1). Following the existing literature (Li et al., 2023; Chen et al., 2023b;c; Duan & Wang, 2023), we define the statistical heterogeneity as the Euclidean distance between the true local models $\boldsymbol{w}_\star^{(i)}$ and the true global model $\boldsymbol{w}_\star^{(g)} := \sum_{i=1}^m p_i \boldsymbol{w}_\star^{(i)}$. Specifically, we adopt the following assumption:

**Assumption 1** (Statistical Heterogeneity). *There exists a nonnegative constant $R$ such that we have* $\|\boldsymbol{w}_\star^{(i)} - \boldsymbol{w}_\star^{(g)}\|^2 \le R^2, \forall i \in [m]$.

Intuitively, a larger $R$ indicates a greater divergence between the clients' data distributions, thus stronger statistical heterogeneity, and vice versa. Therefore, in the FedProx objective, the choice of $\lambda$ should be adjusted according to $R$. More discussion about Assumption 1 can be found in Appendix A.1. Next, we introduce three standard assumptions.

**Assumption 2** (Smoothness). *The loss function $\ell(\cdot, z)$ is $L$-smooth, i.e. for any $x, y \in \mathbb{R}^d$*

$$\|\nabla\ell(x, z) - \nabla\ell(y, z)\| \le L\|x - y\|, \quad \forall z. \tag{3}$$

**Assumption 3** (Strong Convexity). *The loss function $\ell(\cdot, z)$ is almost surely $\mu$-strongly convex:* $\ell(x, z) \ge \ell(y, z) + \langle \nabla\ell(y, z), y - x \rangle + \frac{\mu}{2}\|x - y\|^2, \quad \forall x, y \in \mathbb{R}^d$.

**Assumption 4** (Bound Gradient Variance at Optimum). *There exists a nonnegative constant $\rho$ such that $\forall i \in [m]$, we have $\mathbb{E}_{z \sim \mathcal{D}_i}\|\nabla\ell(\boldsymbol{w}_\star^{(i)}, z)\|^2 \le \rho^2$.*

Here, the strongly convex and smooth assumption is used to establish the estimation error and is widely adopted in the theoretical analysis of regularization-based personalized federated learning (T Dinh et al., 2020; Deng et al., 2020; Hanzely & Richtárik, 2020; Hanzely et al., 2020) and theoretical analysis in federated learning (Chen et al., 2023c; Cheng et al., 2023). Assumption 4 is used to quantify the observation noise, and is also standard in the literature (Duan & Wang, 2023; Chen et al., 2023c).

Next, we establish a statistical rate of the optimal solution to the Fedprox problem (1) with a fine-grained analysis. To facilitate the analysis, we first define some notations. Define $\{\widetilde{\boldsymbol{w}}^{(i)}\}_{i=1}^m$ as the optimal solution for local models $\boldsymbol{w}^{(i)}$ in objective (1). A check on the optimality condition implies that the optimal solution for the global model is $\widetilde{\boldsymbol{w}}^{(g)} = \sum_{i=1}^m p_i \widetilde{\boldsymbol{w}}^{(i)}$. In this paper, we use global statistical error $\mathbb{E}\|\widetilde{\boldsymbol{w}}^{(g)} - \boldsymbol{w}_\star^{(g)}\|^2$ and local statistical error $\mathbb{E}\|\widetilde{\boldsymbol{w}}^{(i)} - \boldsymbol{w}_\star^{(i)}\|^2$ as the measure of statstical accuracy. Here the expectation is taken with respect to all data. Note that some existing literature adopts alternative metrics, such as *average excess risk* and *individual excess risk* (Chen et al., 2023c) to measure the statistical error. The convergence rate of these two definitions is closely related up to the smoothness constant and strong convexity constant. Define $\tilde{\boldsymbol{w}}_{\text{GT}}$ as the optimal solution for the GlobalTrain problem in (2). We first provide a lemma to cast light on the effect of $\lambda$ on the statistical error. See Appendix A.2 for the proof.

**Lemma 1.** *Suppose Assumptions 1, 2 and 3 hold. We have*

$$\|\widetilde{\boldsymbol{w}}^{(g)} - \tilde{\boldsymbol{w}}_{GT}\|^2 \le \frac{4(1 + L/\mu)}{\mu(\mu + \lambda)} \sum_{i \in [m]} p_i \|\nabla L_i(\widetilde{\boldsymbol{w}}^{(g)}, S_i)\|^2, \tag{4}$$

$$\sum_{i \in [m], j \in [n_i]} \frac{p_i}{n_i} \left( \mu\|\widetilde{\boldsymbol{w}}^{(i)} - \boldsymbol{w}_\star^{(i)}\|^2 + \langle \nabla\ell(\boldsymbol{w}_\star^{(i)}, z_{ij}), \widetilde{\boldsymbol{w}}^{(i)} - \boldsymbol{w}_\star^{(i)} \rangle \right) \le \frac{\lambda}{2}R^2. \tag{5}$$

Lemma 1 establishes a relationship between the model estimation error and the tuning parameter $\lambda$. From the bound (4), we can see that a larger $\lambda$ will push the global model solution $\widetilde{\boldsymbol{w}}^{(g)}$ towards

the GlobalTrain solution. On the other hand, from the bound (5), we can set a smaller $\lambda \propto 1/nR^2$ to obtain a rate $O(1/n)$, independent of $R$. This suggests that $\lambda$ can be chosen adaptively w.r.t. $R$ to achieve the optimal statistical rate. A detailed discussion can be found in Appendix A.3. In the following theorem, we provide such a choice and establish a state-of-the-art statistical rate.

**Theorem 1.** *Suppose Assumptions 1, 2, 3 and 4 hold and $n_i = n, \forall i \in [m]$. Consider the FedProx objective in (1) with $p_i = \frac{1}{m}, \forall i \in [m]$. If it takes $\lambda \geq \max \left\{ 64\kappa^2 L, (2\kappa \vee 5)\, \mu \frac{2L^2R^2 + \rho^2/n}{L^2R^2 + \rho^2/N} \right\} - \mu$ when $R \leq \frac{1}{\sqrt{n}}$, and $\lambda \leq \frac{\rho^2}{n\mu R^2}$ when $R > \frac{1}{\sqrt{n}}$, the global model $\widetilde{w}^{(g)}$ and local models $\{\widetilde{w}^{(i)}\}_{i=1}^m$ obtained from solving the objective (1) satisfy*

$$\mathbb{E}\left\| \widetilde{w}^{(g)} - w_\star^{(g)} \right\|^2 \leq \frac{C_1}{N} + C_2 \left( \frac{1}{n} \wedge R^2 \right), \tag{6}$$

$$\mathbb{E}\left\| \widetilde{w}^{(i)} - w_\star^{(i)} \right\|^2 \leq \frac{C_3}{N} + C_4\left(\frac{1}{n} \wedge R^2\right) \quad \text{for all } i \in [m], \tag{7}$$

*where $N = mn$ and*

$$C_1 = 32\frac{\rho^2}{\mu^2}, C_2 = \big(\frac{(1+\sqrt{3})\rho}{\mu}\big)^2 \vee \frac{32L^2}{\mu^2}, C_3 = \frac{192\rho^2}{\mu^2}, C_4 = \left( \frac{\rho(1 + (4C_2 + 5)^{\frac{1}{2}})}{\mu} \right)^2 \vee \frac{198L^2}{\mu^2}.$$

*Proof.* See Appendix A.3. □

Theorem 1 shows that with an appropriate choice of $\lambda$, both the global and local models achieve a fast statistical rate across a wide range of $R$. To interpret the result, we first analyze the error bound provided in (6) and (7). The two error bounds have the same order and can be decomposed into two key components. The first term, of order $\mathcal{O}(N^{-1})$, captures the benefit of collaborative learning, as $N = mn$ is the total sample size across $m$ clients. This term reflects the sample efficiency gained through collaborative learning. The second term, of order $\mathcal{O}(n^{-1} \wedge R^2)$, measures the cost of statistical heterogeneity. As $R$ increases, the convergence rate gradually degrades, transitioning from $\mathcal{O}(N^{-1})$ to $\mathcal{O}(n^{-1})$. Notably, the guarantee is minimax-optimal, as the derived bound matches the problem's lower bound under the given assumptions (Chen et al., 2023c). Remark 2 in Appendix A.3 discusses about how such an interpolation is achieved through different personalization degrees.

Table 1: Results Comparison. $D := \sup_{w,w' \in \mathcal{D}} \|w - w'\|$ is diameter of the parameter space ($D \geq R$) and $\|\ell\|_\infty = \inf\{M \in \mathbb{R} : \ell(\cdot, z) \leq M, \text{for any } z\}$ is an uniform upper bound on the loss function. Statistical rate refers to $\mathbb{E}\|w_\star^{(i)} - \widetilde{w}^{(i)}\|^2$, with constants neglected.

| Source | Assumption | Paradigm | Statistical Rate | | |
|---|---|---|---|---|---|
| | | | $R > \frac{1}{\sqrt{n}}$ | $\frac{1}{m\sqrt{n}} < R \leq \frac{1}{\sqrt{n}}$ | $R \leq \frac{1}{m\sqrt{n}}$ |
| Chen et al. (2023c) | A1,2,3,4, $D \vee \|\ell\|_\infty \leq C$ | Two-stage | $\frac{D^2 + \|\ell\|_\infty}{n}$ | $\frac{\|\ell\|_\infty}{\sqrt{n}}R$ | $\frac{\|\ell\|_\infty}{\sqrt{mn}}$ |
| Ours | A1,2,3,4 | One-stage | $\frac{1}{n}$ | $R^2 \wedge \frac{1}{mn}$ | $\frac{1}{mn}$ |
| Lower Bound | - | - | $\frac{1}{n}$ | $R^2 \wedge \frac{1}{mn}$ | $\frac{1}{mn}$ |

We now compare our theoretical results with the result up to date (Chen et al., 2023c), as summarized in Table 1. For a fair comparison, it is noticeable that we have transformed their result into our metrics $\mathbb{E}\|w_\star^{(i)} - \widetilde{w}^{(i)}\|^2$, by removing their optimization error and problem-dependent constant. Our contribution can be summarized into three aspects:

• Our results are derived under more realistic assumptions. Chen et al. (2023c) imposes additional stringent conditions including a bounded parameter space $D$ and a uniform bound on the loss function $\ell(\cdot, z)$. Such conditions are difficult to satisfy in practice. For example, even a simple linear regression model with sub-Gaussian covariates would violate these assumptions.

• Our result is established for both global and local models with the same choice of $\lambda$. In contrast, Chen et al. (2023c) proposes a two-stage process where the global model is first estimated, followed

by an additional local training phase with a different choice of $\lambda$. This two-step approach is likely the artifact of their analysis, as existing work on FedProx also shows that a single-stage optimization with a chosen $\lambda$ suffices to achieve the desired performance (T Dinh et al., 2020).

• We establish a state-of-the-art minimax statistical rate. As shown in Table 1, the established upper bound matches the lower bound established in Chen et al. (2023c). In a heterogeneous case when $R \geq n^{-1/2}$, Chen et al. (2023c) establishes a rate that is at least of the order $\mathcal{O}\left(n^{-1}R^2\right)$ as $D \geq R$. This bound becomes increasingly loose with larger $R$. In contrast, our bound is $\mathcal{O}\left(n^{-1}\right)$, which shows that properly-tuned FedProx is always no worse than LocalTrain, independent of $R$. In a homogeneous case when $R \leq m^{-1}n^{-1/2}$, they can only establish a rate slower than $\mathcal{O}\left(1/(\sqrt{m}n)\right)$, while our result achieves a rate of $\mathcal{O}\left(1/(mn)\right)$, leveraging all $mn$ samples and matching the rate of GlobalTrain on the IID data. As the client number could be extremely large ($10^5$ devices in Chen et al. (2023a)), order of $m$ is non-trivial. During the transition period when $m^{-1}n^{-1/2} < R < n^{-1/2}$, we achieve a rate of $\mathcal{O}(R^2)$, again matching the lower bound, and is strictly faster than the rate $\mathcal{O}(n^{-1/2}R)$ established in Chen et al. (2023c).

## 5 WHEN WILL PERSONALIZATION IMPROVE COMMUNICATION EFFICIENCY

Notice that if we define the local objective of each client $i$ as $h_i(\boldsymbol{w}^{(i)}, \boldsymbol{w}^{(g)}) := L_i(\boldsymbol{w}^{(i)}, S_i) + \frac{\lambda}{2}\|\boldsymbol{w}^{(g)} - \boldsymbol{w}^{(i)}\|^2$, then the FedProx problem given by (1) can be rewritten in the following bilevel (iterated minimization) form:

$$\min_{\boldsymbol{w}^{(g)}} F(\boldsymbol{w}^{(g)}) := \frac{1}{m}\sum_{i=1}^{m} F_i(\boldsymbol{w}^{(g)}), \quad \text{where} \quad F_i(\boldsymbol{w}^{(g)}) := \min_{\boldsymbol{w}^{(i)}} h_i(\boldsymbol{w}^{(i)}, \boldsymbol{w}^{(g)}). \tag{8}$$

The reformulation (8) is a finite-sum minimization problem, with each component $F_i$ being the Moreau envelope (Moreau, 1965; Yosida, 1964) of $L_i$. Let $\boldsymbol{w}_\star^{(i)}(\boldsymbol{w}^{(g)})$ be the minimizer of $h_i(\,\cdot\,, \boldsymbol{w}^{(g)})$, i.e.,

$$\boldsymbol{w}_\star^{(i)}(\boldsymbol{w}^{(g)}) := \text{prox}_{L_i/\lambda}(\boldsymbol{w}^{(g)}) = \arg\min_{\boldsymbol{w}^{(i)}} h_i(\boldsymbol{w}^{(i)}, \boldsymbol{w}^{(g)}). \tag{9}$$

Based on Assumption 2 and the new notations in (8) and (9), the following lemma describes the optimizing properties for $F_i$, $h_i$ and $\boldsymbol{w}_\star^{(i)}$. The proof can be found in Appendix B.2.

**Lemma 2.** *Under Assumption 3 and 2, $F_i$ is $\mu_g$-strongly convex and $L_g$-smooth, with $\mu_g = \frac{\lambda\mu}{\lambda+\mu}$ and $L_g = \frac{\lambda L}{\lambda+L}$, each $h_i$ is $\mu_\ell$-strongly convex and $L_\ell$-smooth, with $\mu_\ell = \mu + \lambda$ and $L_\ell = L + \lambda$, and the mapping $\boldsymbol{w}_\star^{(i)} : \mathbb{R}^d \to \mathbb{R}^d$ is $L_w$-Lipschitz with $L_w = \frac{\lambda}{\lambda+\mu}$.*

Next, we recall the properties of the Moreau envelope to guide the algorithm design.

**Lemma 3** (Lemaréchal & Sagastizábal (1997)). *Under Assumption 3, each $F_i : \mathbb{R}^d \to \mathbb{R}$ is continuously differentiable, and the gradient is given by $\nabla F_i(\boldsymbol{w}^{(g)}) = \lambda(\boldsymbol{w}^{(g)} - \boldsymbol{w}_\star^{(i)}(\boldsymbol{w}^{(g)}))$.*

From Lemma 2 and Lemma 3, it suggests applying a simple gradient algorithm to optimize $\boldsymbol{w}^{(g)}$:

$$\boldsymbol{w}_{t+1}^{(g)} = \boldsymbol{w}_t^{(g)} - \gamma \cdot \nabla F(\boldsymbol{w}_t^{(g)}) = \boldsymbol{w}_t^{(g)} - \gamma\lambda \cdot \frac{1}{m}\sum_{i=1}^{m}\left(\boldsymbol{w}_t^{(g)} - \boldsymbol{w}_\star^{(i)}(\boldsymbol{w}_t^{(g)})\right), \tag{10}$$

where $\boldsymbol{w}_t^{(g)}$ is the update at the communication round $t$, $\gamma > 0$ is a step size to be properly set (cf. Theorem 2). To implement (10), in each communication round $t$ the server broadcasts $\boldsymbol{w}_t^{(g)}$ to all clients, then each client $i$ updates its local model $\boldsymbol{w}_\star^{(i)}(\boldsymbol{w}_t^{(g)})$ and upload to the server.

Note that executing (10) requires each client $i$ computing the minimizer $\boldsymbol{w}_\star^{(i)}(\boldsymbol{w}_t^{(g)})$. In general, the subproblem (9) does not have a closed-form solution. Therefore, computing the exact gradient $\nabla F$ will incur a high computation cost as well as high latency for the learning process. Leveraging recent advancements in bilevel optimization algorithm design (Ji et al., 2022), we address this issue by approximating $\boldsymbol{w}_\star^{(i)}(\boldsymbol{w}_t^{(g)})$ with a finite number of $K$ gradient steps. Specifically, we let each client

$i$ maintain a local model $\boldsymbol{w}_{t,k}^{(i)}$. Per communication round $t$, $\boldsymbol{w}_{t,k}^{(i)}$ is initialized to be $\boldsymbol{w}_{t,0}^{(i)} = \boldsymbol{w}_{t-1,K}^{(i)}$ as a warm start, and updated according to

$$\boldsymbol{w}_{t,k+1}^{(i)} = \boldsymbol{w}_{t,k}^{(i)} - \eta \nabla h_i(\boldsymbol{w}_{t,k}^{(i)}, \boldsymbol{w}_t^{(g)}), \quad \forall k = 0, \ldots, K-1, \tag{11}$$

where $\nabla h_i(\boldsymbol{w}_{t,k}^{(i)}, \boldsymbol{w}_t^{(g)})$, for notation simplicity, denotes the partial gradient of $h_i$ with respect to $\boldsymbol{w}^{(i)}$. The overall procedure is summarized in Alg. 1, see Appendix B.1.

**Remark 1.** *Alg. 1 is a customization of the general bi-level problem (Ji et al., 2022). While the algorithm design is not new, our contribution lies in analyzing how the personalization degree influences the trade-off between communication and local computation leveraging the FedProx problem structure. Extending to noisy settings to consider stochastic gradient or variance reduction would be interesting future work, which requires additional technical effort.*

**Theorem 2.** *Suppose Assumptions 3 and 2 hold. Let $\{\boldsymbol{w}_t^{(g)}\}_{t \geq 0}$ and $\{\boldsymbol{w}_{t,K}^{(i)}\}_{t \geq 0}$ be the sequence generated by Algorithm 1 with $\gamma < 1/L_g$, $\eta \leq 1/L_\ell$, and the inner loop iteration number satisfying*

$$\left(2 + 64 L_w^2 (1/\mu_g)^2 \lambda^2\right)(1 - \eta \mu_\ell)^K \leq (1 - \gamma L_g)^4, \tag{12}$$

*then $\boldsymbol{w}_t^{(g)}$ converges to $\widetilde{\boldsymbol{w}}^{(g)}$ linearly at rate $1 - (\gamma \mu_g)/2 - (\gamma \mu_g)^2/2$, and $\boldsymbol{w}_{t,0}^{(i)}$ converges to $\widetilde{\boldsymbol{w}}^{(i)}$ linearly at the same rate for any $i \in [m]$.*

*Proof.* See Appendix B.3 and B.5. $\square$

Theorem 2 shows under the step size constraint $\gamma < 1/L_g$, $\boldsymbol{w}_t^{(g)}$ converges to $\tilde{\boldsymbol{w}}^{(g)}$ at the same rate as the iteration (10) with exact gradient computation. Furthermore, the larger $\gamma$ is, the larger $K$ should be chosen so as to satisfy (12). This indicates a tradeoff between communication and computation: the fewer communication rounds are performed, then the more local computation is required for the algorithm to converge.

To better interpret the result and obtain insights on the influence of $\lambda$ on the convergence, we provide the communication and computation complexity under specific choices the algorithm's tuning parameters, as described in the following Corollary 1. See Appendix B.4 for the proof.

**Corollary 1.** *In the same setting as Theorem 2, if choosing the step size $\eta = 1/L_\ell = (\lambda + L)^{-1}$, $\gamma = 1/(2L_g) = \frac{\lambda + L}{2\lambda L}$, and the inner loop iteration number $K = \tilde{\mathcal{O}}\left(\frac{\lambda + L}{\lambda + \mu}\right)$, then the communication complexity and computation complexity for Algorithm 1 to find an $\varepsilon$-solution (i.e., $\|\boldsymbol{w}_T^{(g)} - \tilde{\boldsymbol{w}}^{(g)}\|^2 \leq \varepsilon$ and $\|\boldsymbol{w}_{T,0}^{(i)} - \tilde{\boldsymbol{w}}^{(i)}\|^2 \leq \varepsilon$) are as follows ($\kappa := L/\mu$):*

$$(i) \text{ \# communication rounds} = \mathcal{O}\left(\kappa_g \log(1/\varepsilon)\right) = \mathcal{O}\left(\kappa \cdot \frac{\lambda + \mu}{\lambda + L} \cdot \log \frac{1}{\varepsilon}\right), \tag{13}$$

$$(ii) \text{ \# gradient evaluations} = \tilde{\mathcal{O}}\left(\kappa \cdot \log \frac{1}{\varepsilon}\right). \tag{14}$$

• **Communication cost.** Corollary 1 shows as $\lambda$ increases from 0 to $\infty$, the communication complexity increases from $\tilde{\mathcal{O}}(1)$ to $\tilde{\mathcal{O}}(\kappa)$. The result reveals a higher personalization degree requires less communication cost to collaboratively learn the models. In the full personalization limit with $\lambda \to 0$, the order of the communication cost reduces to near constant $\tilde{\mathcal{O}}(1)$, which corresponds to pure local training. It is worth noting that even personalization can reduce communication complexity, as our ultimate goal is to accurately estimate each model so we can not simply pursue higher communication efficiency.

• **Computation cost.** In contrast to the communication complexity, the gradient evaluations per round $K$ decrease with $\lambda$ increasing. Therefore, a higher degree of personalization requires a more intensive update of the local models once receiving a new global model $\boldsymbol{w}^{(g)}$, so as to adapt to each client's local data set. Note that although a small $\lambda$ requires a larger $K$, the overall computation cost given by (14) is independent of $\lambda$. As such, model personalization (small $\lambda$) can provably reduce the communication cost without any extra computation overhead.

Table 2: Comparison of the communication and computation complexity in existing works of analyzing the objective in (1) (cvx denotes convexity and s-cvx denotes strong convexity).

| Methods | Convexity | Bounded Domain | Computation Cost per Round | Communication Cost |
|---|---|---|---|---|
| FedProx (Li et al., 2020) | cvx | ✗ | $-^{(1)}$ | $\mathcal{O}(\Delta/(\rho\varepsilon))$ |
| pFedMe (T Dinh et al., 2020) | s-cvx | ✗ | $-^{(2)}$ | $\mathcal{O}(1/(mR\varepsilon))$ |
| Ditto (Li et al., 2021) | s-cvx | ✗ | $\mathcal{O}(1)$ | $\mathcal{O}(1/\varepsilon)$ |
| L2GD (Hanzely & Richtárik, 2020) | s-cvx | ✗ | $\mathcal{O}(1+\frac{L}{\lambda})$ | $\mathcal{O}(\frac{2\lambda\kappa}{\lambda+L}\log(\frac{1}{\varepsilon}))$ |
| Algorithm 3 (Chen et al., 2023c) | s-cvx | ✔ | $\mathcal{O}(\lambda/\varepsilon)$ | $\mathcal{O}((\lambda \vee 1)/\varepsilon)$ |
| Our Result | s-cvx | ✗ | $\mathcal{O}(\frac{\lambda+L}{\lambda+\mu})$ | $\mathcal{O}(\kappa\frac{\lambda+L}{\lambda+L}\log\frac{1}{\varepsilon})$ |

$^{(1)}$ Controlled by the precision of inexact solution. $\rho > 0$ measures the subproblem solution accuracy.
$^{(2)}$ Related to the subiteration number $R$ and the precision $v$.

We end this section with a comparison between our results and the existing algorithms for solving FedProx, as reported in Table 5. Our result shows how the personalization degree trades off the communication cost and local computation cost, whereas most of the existing works either don't explicitly analyze the local computational cost (Li et al., 2020; T Dinh et al., 2020), or fail to demonstrate such a trade-off (Chen et al., 2023c). Hanzely & Richtárik (2020) also reveals similar tradeoffs, but their work is a randomized method that requires more coordination between server and clients, and more importantly, when $\lambda$ is small, the computation cost per round is unbounded.

## 6 SIMULATION

In this section, we provide empirical validation for our theoretical results, evaluating FedProx on both synthetic and real datasets with convex and non-convex loss functions. We first examine the effect of personalization on statistical accuracy under different levels of heterogeneity (Section 6.2), followed by an analysis of its impact on communication efficiency (Section 6.3). Detailed descriptions of the experimental setup are available in Section 6.1 and Appendix C, and the complete anonymized codebase is accessible at https://anonymous.4open.science/r/fedprox-bilevel.

### 6.1 EXPERIMENTAL DETAILS

**Synthetic Dataset.** To generate the synthetic data, we follow a similar procedure to prior works (Li et al., 2020) but with some modifications to align with the setup in this paper. Specifically, for each client we generate samples $(\boldsymbol{X}_k, \boldsymbol{y}_k)$, where the labels $\boldsymbol{y}_k$ are produced by a logistic regression model $\boldsymbol{y}_k = \text{argmax}(\sigma(\boldsymbol{w}_k^\top \boldsymbol{X}_k))$, with $\sigma$ being the sigmoid function. The feature vectors $\boldsymbol{X}_k$ are drawn from a multivariate normal distribution $\mathcal{N}(v_k, \boldsymbol{\Sigma})$, where $v_k \sim \mathcal{N}(0, 1)$ and the covariance matrix $\boldsymbol{\Sigma}$ follows a diagonal structure $\boldsymbol{\Sigma}_{j,j} = j^{-1.2}$. The heterogeneity is introduced by sampling the model weights $\boldsymbol{w}_k$ for each client from a normal distribution $\mathcal{N}(0, R)$, where $R \in [0, 3]$ controls the degree of variation (statistical heterogeneity) across clients' data distributions.

**Real Dataset.** We use the MNIST, EMNIST, and CIFAR10 datasets for the real dataset analysis. We first formulate the problem as a convex classification task using multinomial logistic regression, then test the generalization of our theory to a more general, non-convex scenario by conducting experiments using different CNN classifiers. Following Li et al. (2020), to impose statistical heterogeneity, we distribute the data across clients in the way that each client only has access to a fixed number of classes. The fewer classes each client has access to, the higher the statistical heterogeneity. The detail of data processing to set data heterogeneity for each dataset can be found in Appendix C.

**Implementation and Evaluation.** For each setup, we evaluate the FedProx algorithm with three different choices of $\lambda$: small, medium, and large, with specific values varying based on the dataset (details provided in Appendix C). For the synthetic dataset, in alignment with theorem 2, we implement Algorithm 1 using a global step size $\gamma = (\lambda + L)/(\lambda L)$ and a local step size $\eta = (L + \lambda)^{-1}$. For the real dataset, since $L$ is unknown, we implement the same algorithm with a global step size $\gamma$ set to $1/(\lambda)$, while the local step size is determined via grid search. In terms of evaluation, for

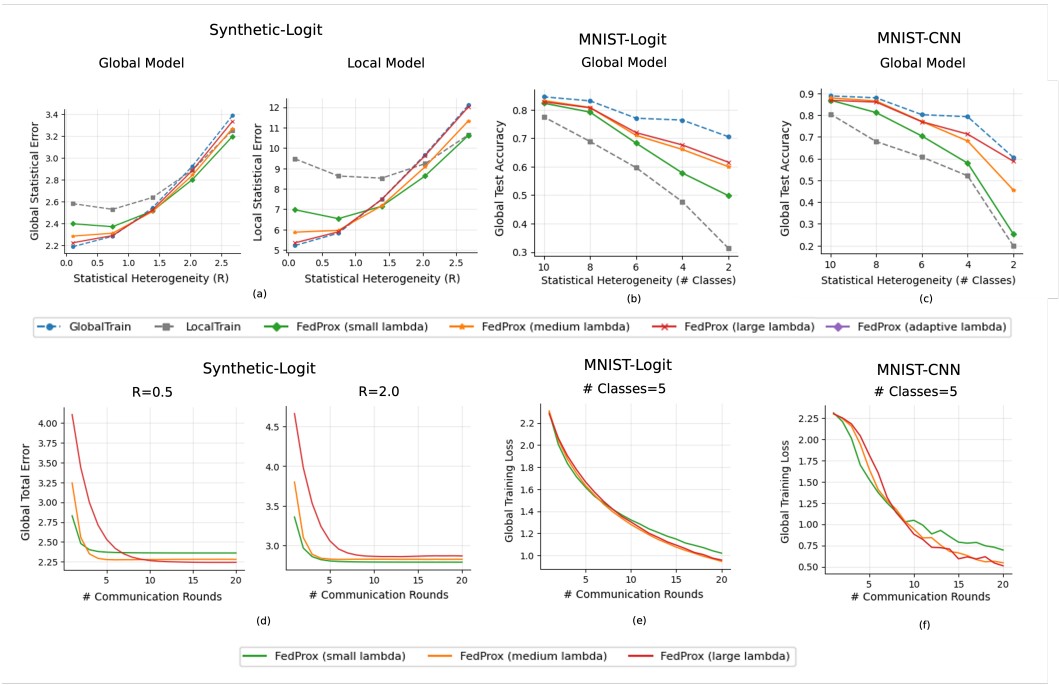

Figure 1: Statistical and total error of different methods. In the top row, subfigures (a), (b), and (c) represent statistical error, showing the difference between the minimizer and the underlying true model for Synthetic-Logit (logistic regression on synthetic data), MNIST-Logit (logistic regression on MNIST), and MNIST-CNN (CNN on MNIST), respectively. In the bottom row, subfigures (d), (e), and (f) represent total error, showing the error between the model at each round and the true model for the same datasets. (1) FedProx adapts between GlobalTrain and LocalTrain, achieving low error across varying levels of heterogeneity with a properly chosen $\lambda$; (2) Larger $\lambda$ (less personalization) requires more communication to converge, and vice versa.

the synthetic dataset, we evaluate by tracking the ground-truth models and measuring error as the distance from these true models. For the real dataset, we report training loss and testing accuracy. Additional details, including hyperparameter setting and evaluation metric definition, can be found in Appendix C.

## 6.2 STATISTICAL ASPECT: BALANCING LOCAL AND GLOBAL LEARNING

We now explore how FedProx balances between GlobalTrain and LocalTrain by achieving different levels of personalization. For comparison, in addition to implementing FedProx with different choices of $\lambda$, we also implemented LocalTrain and GlobalTrain, with details specified in Appendx C. We run all methods to a stable convergence and measure the statistical error. The top row of Figure 1 shows the performance of these methods under different impact of statistical heterogeneity.

From Figure 1(a), in a controlled convex setting, we observe that FedProx consistently interpolates between GlobalTrain and LocalTrain. As $\lambda$ increases, FedProx transitions from behaving similarly to LocalTrain towards GlobalTrain. When statistical heterogeneity is low, both GlobalTrain and FedProx with larger $\lambda$ values achieve better statistical performance. However, as heterogeneity increases, LocalTrain and FedProx with smaller $\lambda$ values show better results. The figure also demonstrates that with *an adaptive choice of $\lambda$ in theoretical analysis* outlined in Theorem 1, Fed-Prox consistently achieves a low error across varying levels of heterogeneity. See more results and discussion in Appendix C. Note that as statistical heterogeneity increases, LocalTrain experiences a slight rise in error due to the increased difficulty of the statistical problem. However, it remains the least affected by heterogeneity since it relies solely on local data. In Figure 1(b) and 1(c), for the MNIST dataset, which includes both convex logistic regression and non-convex CNN settings, we observe similar patterns. FedProx again strikes a balance between GlobalTrain and LocalTrain,

adapting its performance as $\lambda$ changes. While GlobalTrain performs best in this scenario due to the relatively weak statistical heterogeneity, FedProx, with increasing $\lambda$, moves closer to Global-Train performance. In Figure 4 in Appendix C, we also evaluate the local model's statistical error and observe that as statistical heterogeneity increases, FedProx with smaller $\lambda$ values can surpass GlobalTrain.

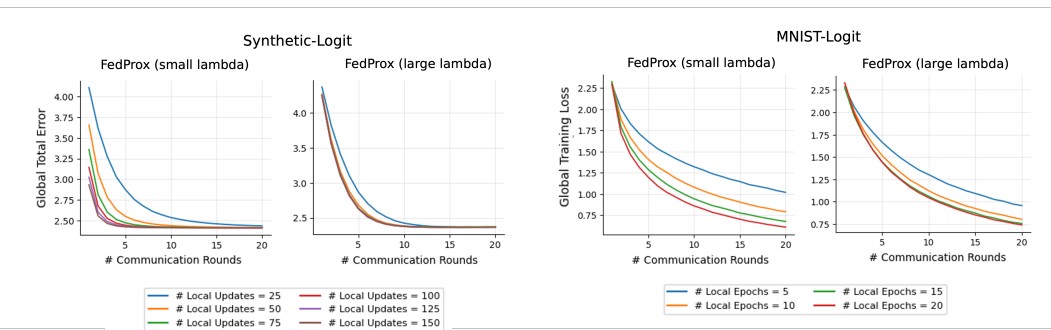

Figure 2: Effect of local updates on FedProx performance. Smaller $\lambda$ reduces communication with more local updates, while larger $\lambda$ benefits marginally from increasing local updates and requires more frequent communication.

### 6.3 OPTIMIZATION ASPECT: COMMUNICATION-COMPUTATION TRADE-OFF

In this section, we explore the trade-off between communication and computation in FedProx, controlled by varying the personalization degree.

**Communication Cost.** In the bottom row of Figure 1, we show the change in the total error of the algorithm (i.e., the difference of the model in $t$th round between the true underlying model) over communication rounds. The first observation is that as $\lambda$ increases, the convergence rate w.r.t. communication rounds slows across all settings. FedProx with a large $\lambda$ (closer to GlobalTrain) needs the most communication rounds to converge, while smaller $\lambda$ values (closer to LocalTrain) result in faster convergence. This result aligns with our theory (c.f. Corollary 1), showing that Fedprox with less personalization relies more heavily on communication. On the statistical side, Figure 1(d) shows that when statistical heterogeneity is low, larger $\lambda$ leads to the smallest total error. As $R$ increases, smaller $\lambda$ (more personalization) results in better performance.

**Computation Cost.** In Figure 2, we investigate how the number of local updates per communication round impacts the performance of FedProx at different personalization degree. Additional results in the non-convex setting can be found in Appendix C. Notice that for models with smaller $\lambda$ (higher personalization), increasing the number of local updates significantly improves convergence, making the algorithm less dependent on frequent communication. This aligns with our analysis: more personalized models rely less on communication and benefit more from local computation.

## 7 CONCLUSION

In this paper, we analyzed of the effect of personalization in FedProx. With a theoretically-informed adaptive choice of personalization degree, FedProx achieves a minimax-optimal statistical accuracy under different statistical heterogeneity degrees and consistently outperforms pure local training. As a minor contribution, we provably show that personalization reduces communication costs without increasing the computation overhead. The results are validated through extensive experiments.

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

# A    PROOF OF STATISTICAL CONVERNGENCE

To facilitate our analysis, we denote

$$\tilde{\Delta}_{\text{stat}}^{(g)} = \boldsymbol{w}_\star^{(g)} - \widetilde{\boldsymbol{w}}^{(g)}, \tag{15}$$

$$\tilde{\Delta}_{\text{stat}}^{(i)} = \boldsymbol{w}_\star^{(i)} - \widetilde{\boldsymbol{w}}^{(i)} \quad \forall i \in [m], \tag{16}$$

where $\tilde{\Delta}_{\text{stat}}^{(g)}$ is the difference between the true global model and the optimal solution of the global model in (1) and $\tilde{\Delta}_{\text{stat}}^{(i)}$ is the difference between the true local model and the optimal solution of the local model in (1) for any $i \in [m]$. The statistical error bound is established as $\mathbb{E}\|\tilde{\Delta}_{\text{stat}}^{(g)}\|^2$ and $\mathbb{E}\|\tilde{\Delta}_{\text{stat}}^{(i)}\|^2$, where the expectation is taken over all the data.

And we also denote

$$\boldsymbol{\delta}_\star^{(i)} := \boldsymbol{w}_\star^{(g)} - \boldsymbol{w}_\star^{(i)}, \quad \tilde{\boldsymbol{\delta}}^{(i)} := \tilde{\boldsymbol{w}}^{(g)} - \tilde{\boldsymbol{w}}^{(i)}, \tag{17}$$

where $\boldsymbol{\delta}_\star^{(i)}$ measures the difference between the true global model $\boldsymbol{w}_\star^{(g)}$ and the true local model $\boldsymbol{w}_\star^{(i)}$, and $\tilde{\boldsymbol{\delta}}^{(i)}$ is the estimator of such a difference.

## A.1    DISCUSSION ON ASSUMPTION 1

When assuming the statistical heterogeneity of each client is different ($R$ is related to the client index $i$), it would provide a more delicate description of statistical heterogeneity, and it also requires different personalization degree per client. Therefore, it is better to consider the objective (18), where the $i$-th client will be shrunk to the global with strength $\lambda_i$. Therefore, solving (18) will consider different personalization degree for different clients. Solving the objective (18) would be open for future work.

$$\min_{\boldsymbol{w}^{(g)}, \{\boldsymbol{w}^{(i)}\}_{i \in [m]}} \sum_{i=1}^m p_i \left( L_i(\boldsymbol{w}^{(i)}, S_i) + \frac{\lambda_i}{2}\|\boldsymbol{w}^{(g)} - \boldsymbol{w}^{(i)}\|^2 \right), \tag{18}$$

Under Assumptions 2 and 3 as we assume, Assumption 1 about statistical heterogeneity would be equivalent to many other similar assumptions for statistical heterogeneity, like $B$-dissimilarity (Li et al., 2020), parameter difference (Chen et al., 2023c) and gradient diversity (T Dinh et al., 2020; Deng et al., 2020).

## A.2    LEMMAS AND THE PROOF

To facilitate the proof of the theorem, we first present two additional lemmas and provide their proof.

**Lemma 4.** *Under Assumption 3, for the optimal solution $\tilde{\boldsymbol{w}}^{(g)}$ and $\tilde{\boldsymbol{w}}^{(i)}$'s in Problem (1), we have*

$$\|\tilde{\boldsymbol{w}}^{(g)} - \tilde{\boldsymbol{w}}^{(i)}\|_2 \le \frac{2\|\nabla L_i(\tilde{\boldsymbol{w}}^{(g)}, S_i)\|_2}{\mu + \lambda}.$$

By the $\mu$-strongly convexity of $\nabla L_i$, we have that for any $\boldsymbol{w} \in \mathbb{R}^d$,

$$L_i(\boldsymbol{w}, S_i) + \frac{\lambda}{2}\|\boldsymbol{w} - \tilde{\boldsymbol{w}}^{(g)}\|^2 \ge L_i(\tilde{\boldsymbol{w}}^{(g)}, S_i) + \left\langle \nabla L_i(\tilde{\boldsymbol{w}}^{(g)}, S_i), \boldsymbol{w} - \tilde{\boldsymbol{w}}^{(g)} \right\rangle + \frac{\mu + \lambda}{2}\|\boldsymbol{w} - \tilde{\boldsymbol{w}}^{(g)}\|^2$$

$$\ge L_i(\tilde{\boldsymbol{w}}^{(g)}, S_i) + \|\boldsymbol{w} - \tilde{\boldsymbol{w}}^{(g)}\| \left( \frac{\mu + \lambda}{2}\|\boldsymbol{w} - \tilde{\boldsymbol{w}}^{(g)}\| - \|\nabla L_i(\tilde{\boldsymbol{w}}^{(g)}, S_i)\| \right), \tag{19}$$

where in the last step we used the Cauchy inequality.

In addition, notice that $\tilde{\boldsymbol{w}}^{(i)}$ is the optimal solution for the local model $\boldsymbol{w}^{(i)}$ in the FedProx problem (1), thus we have

$$\tilde{\boldsymbol{w}}^{(i)} = \underset{\boldsymbol{w}}{\operatorname{argmin}} \left\{ L_i(\boldsymbol{w}, S_i) + \frac{\lambda}{2}\|\boldsymbol{w} - \tilde{\boldsymbol{w}}^{(g)}\|^2 \right\}.$$

This result together with (19) implies that

$$L_i(\tilde{\boldsymbol{w}}^{(g)}, S_i) = L_i(\tilde{\boldsymbol{w}}^{(g)}, S_i) + \frac{\lambda}{2}\|\tilde{\boldsymbol{w}}^{(g)} - \tilde{\boldsymbol{w}}^{(g)}\|^2 \geq L_i(\tilde{\boldsymbol{w}}^{(i)}, S_i) + \frac{\lambda}{2}\|\tilde{\boldsymbol{w}}^{(i)} - \tilde{\boldsymbol{w}}^{(g)}\|^2$$

$$\geq L_i(\tilde{\boldsymbol{w}}^{(g)}, S_i) + \|\tilde{\boldsymbol{w}}^{(i)} - \tilde{\boldsymbol{w}}^{(g)}\| \left( \frac{\mu + \lambda}{2}\|\tilde{\boldsymbol{w}}^{(i)} - \tilde{\boldsymbol{w}}^{(g)}\| - \|\nabla L_i(\tilde{\boldsymbol{w}}^{(g)}, S_i)\| \right).$$

Therefore, since $\mu + \lambda > 0$ and $\|\tilde{\boldsymbol{w}}^{(i)} - \tilde{\boldsymbol{w}}^{(g)}\| \geq 0$, rearranging the terms leads to

$$\|\tilde{\boldsymbol{w}}^{(g)} - \tilde{\boldsymbol{w}}^{(i)}\| \leq \frac{2\|\nabla L_i(\tilde{\boldsymbol{w}}^{(g)}, S_i)\|}{\mu + \lambda}, \tag{20}$$

as claimed.

**Lemma 5.** *Under Assumption 1, 2, 3, 4, for $\tilde{\boldsymbol{w}}_{GT}$, the solution of the GlobalTrain problem in (2), we have*

$$\mathbb{E}\|\tilde{\boldsymbol{w}}_{GT} - \boldsymbol{w}_\star^{(g)}\| \leq \frac{\rho\sqrt{\sum_{i \in [m]} \frac{p_i^2}{n_i}} + LR}{\mu/2}.$$

By the optimality condition of the GlobalTrain problem in (2) and the strong convexity of $L_i$'s, we have

$$0 \geq \sum_{i \in [m]} p_i \left( L_i(\tilde{\boldsymbol{w}}_{\text{GT}}, S_i) - L_i(\boldsymbol{w}_\star^{(g)}, S_i) \right)$$

$$\geq \sum_{i \in [m]} p_i \left\langle \nabla L_i(\boldsymbol{w}_\star^{(g)}, S_i), \tilde{\boldsymbol{w}}_{\text{GT}} - \boldsymbol{w}_\star^{(g)} \right\rangle + \frac{\mu}{2}\|\tilde{\boldsymbol{w}}_{\text{GT}} - \boldsymbol{w}_\star^{(g)}\|^2$$

$$\geq -\left\| \sum_{i \in [m]} p_i \nabla L_i(\boldsymbol{w}_\star^{(g)}, S_i) \right\| \left\| \tilde{\boldsymbol{w}}_{\text{GT}} - \boldsymbol{w}_\star^{(g)} \right\| + \frac{\mu}{2}\|\tilde{\boldsymbol{w}}_{\text{GT}} - \boldsymbol{w}_\star^{(g)}\|^2.$$

Therefore, we have

$$\|\tilde{\boldsymbol{w}}_{\text{GT}} - \boldsymbol{w}_\star^{(g)}\| \leq \frac{\left\| \sum_{i \in [m]} p_i \nabla L_i(\boldsymbol{w}_\star^{(g)}, S_i) \right\|}{\mu/2}$$

$$\leq \frac{\left\| \sum_{i \in [m]} p_i \nabla L_i(\boldsymbol{w}_\star^{(i)}, S_i) \right\| + \sum_{i \in [m]} p_i \left\| \nabla L_i(\boldsymbol{w}_\star^{(g)}, S_i) - \nabla L_i(\boldsymbol{w}_\star^{(i)}, S_i) \right\|}{\mu/2}$$

$$\tag{21}$$

By the $L$-smoothness property of $L_i$'s in Assumption 2 and the bounded gradient property listed in Assumption 4, we have

$$\sum_{i \in [m]} p_i \left\| \nabla L_i(\boldsymbol{w}_\star^{(g)}, S_i) - \nabla L_i(\boldsymbol{w}_\star^{(i)}, S_i) \right\| \leq \sum_{i \in [m]} p_i L \|\boldsymbol{w}_\star^{(g)} - \boldsymbol{w}_\star^{(i)}\| \leq LR,$$

$$\mathbb{E}\left\| \sum_{i \in [m]} p_i \nabla L_i(\boldsymbol{w}_\star^{(i)}, S_i) \right\| = \mathbb{E}\left\| \sum_{i \in [m]} \sum_{j \in [n_i]} \frac{p_i}{n_i} \nabla \ell(\boldsymbol{w}_\star^{(i)}, z_{ij}) \right\| \leq \rho\sqrt{\sum_{i \in [m]} \frac{p_i^2}{n_i}}.$$

These results together with (21) implies

$$\mathbb{E}\|\tilde{\boldsymbol{w}}_{\text{GT}} - \boldsymbol{w}_\star^{(g)}\| \leq \frac{\rho\sqrt{\sum_{i \in [m]} \frac{p_i^2}{n_i}} + LR}{\mu/2}$$

as claimed.

### A.3 PROOF OF LEMMA 1 AND THEOREM 1

#### A.3.1 PROOF OF LEMMA 1

Lemma 1 is a direct consequence of the proof of Theorem 1. To see this, from (46) and the fact that $(\mu + \lambda)^2 \geq \mu(\mu + \lambda)$, we have

$$\|\tilde{\boldsymbol{w}}^{(g)} - \tilde{\boldsymbol{w}}_{\text{GT}}\|^2 \leq \frac{4}{\mu} \left( \frac{1}{\mu + \lambda} + \frac{L}{(\mu + \lambda)^2} \right) \sum_i p_i \|\nabla L_i(\tilde{\boldsymbol{w}}^{(g)}, S_i)\|^2$$

$$\leq \frac{4(1 + L/\mu)}{\mu(\mu + \lambda)} \sum_i p_i \|\nabla L_i(\tilde{\boldsymbol{w}}^{(g)}, S_i)\|^2,$$

thus (4) holds. In addition, with the fact that $\|\tilde{\boldsymbol{\delta}}^{(i)}\|_2 \geq 0$, from (28) we find

$$0 \geq -\sum_i \sum_j \frac{p_i}{n_i} \left\langle \nabla \ell(\boldsymbol{w}_\star^{(i)}, z_{ij}), \tilde{\Delta}_{\text{stat}}^{(i)} \right\rangle + \frac{\mu}{2} \sum_i \sum_j \frac{p_i}{n_i} \left\| \tilde{\Delta}_{\text{stat}}^{(i)} \right\|^2 + \frac{\lambda}{2} \sum_i p_i \left\| \tilde{\boldsymbol{\delta}}^{(i)} \right\|^2 - \frac{\lambda}{2} R^2$$

$$\geq -\sum_i \sum_j \frac{p_i}{n_i} \left\langle \nabla \ell(\boldsymbol{w}_\star^{(i)}, z_{ij}), \tilde{\Delta}_{\text{stat}}^{(i)} \right\rangle + \frac{\mu}{2} \sum_i \sum_j \frac{p_i}{n_i} \left\| \tilde{\Delta}_{\text{stat}}^{(i)} \right\|^2 - \frac{\lambda}{2} R^2.$$

which implies the result in (5).

#### A.3.2 PROOF OF GLOBAL STATISTICAL ERROR BOUND IN THEOREM 1

**Case 1**: We first consider the case when $R > \sqrt{\sum_i \frac{p_i}{n_i}}$.

Recall that $\tilde{\boldsymbol{w}}^{(g)}$ and $\{\tilde{\boldsymbol{w}}^{(i)}\}_{i \in [m]}$ are the minimizers of the objective (1). According to the first-order condition, we have

$$\nabla_{\boldsymbol{w}^{(i)}} L_i \left( \boldsymbol{w}^{(i)}, S_i \right) \big|_{\boldsymbol{w}^{(i)} = \tilde{\boldsymbol{w}}^{(i)}} = \lambda \left( \tilde{\boldsymbol{w}}^{(g)} - \tilde{\boldsymbol{w}}^{(i)} \right), \tag{22}$$

$$\tilde{\boldsymbol{w}}^{(g)} = \sum_i p_i \tilde{\boldsymbol{w}}^{(i)}. \tag{23}$$

We start with the optimality condition of $\tilde{\boldsymbol{w}}^{(i)}$ and $\tilde{\boldsymbol{w}}^{(g)}$, which yields

$$0 \geq \sum_{i \in [m]} p_i \left[ L_i(\tilde{\boldsymbol{w}}^{(i)}, S_i) + \frac{\lambda}{2} \|\tilde{\boldsymbol{w}}^{(g)} - \tilde{\boldsymbol{w}}^{(i)}\|^2 \right] - \sum_{i \in [m]} p_i \left[ L_i(\boldsymbol{w}_\star^{(i)}, S_i) + \frac{\lambda}{2} \|\boldsymbol{w}_\star^{(g)} - \boldsymbol{w}_\star^{(i)}\|^2 \right]. \tag{24}$$

Reorganizing the terms, we obtain

$$0 \geq \sum_{i \in [m]} \sum_{j \in [n_i]} \frac{p_i}{n_i} \left( \ell(\tilde{\boldsymbol{w}}^{(i)}, z_{ij}) - \ell(\boldsymbol{w}_\star^{(i)}, z_{ij}) \right) + \sum_{i \in [m]} p_i \frac{\lambda}{2} \|\tilde{\boldsymbol{w}}^{(g)} - \tilde{\boldsymbol{w}}^{(i)}\|^2 - \sum_{i \in [m]} p_i \frac{\lambda}{2} \|\boldsymbol{w}_\star^{(g)} - \boldsymbol{w}_\star^{(i)}\|^2. \tag{25}$$

For the first term on the RHS in (25), apply the $\mu$-strongly convexity of the loss function $\ell$ (cf. Assumption 3), we then have

$$\sum_{i \in [m]} \sum_{j \in [n_i]} \frac{p_i}{n_i} \left( \ell(\tilde{\boldsymbol{w}}^{(i)}, z_{ij}) - \ell(\boldsymbol{w}_\star^{(i)}, z_{ij}) \right) \geq -\sum_{i \in [m]} \sum_{j \in [n_i]} \frac{p_i}{n_i} \left\langle \nabla \ell(\boldsymbol{w}_\star^{(i)}, z_{ij}), \tilde{\Delta}_{\text{stat}}^{(i)} \right\rangle$$

$$+ \frac{\mu}{2} \sum_{i \in [m]} \sum_{j \in [n_i]} \frac{p_i}{n_i} \left\| \tilde{\Delta}_{\text{stat}}^{(i)} \right\|^2, \tag{26}$$

where we use the definition of $\tilde{\Delta}_{\text{stat}}^{(i)}$ in (16).

Plugging (26) back into (25), it yields

$$0 \geq -\sum_i \sum_j \frac{p_i}{n_i} \left\langle \nabla \ell(\boldsymbol{w}_\star^{(i)}, z_{ij}), \tilde{\Delta}_{\text{stat}}^{(i)} \right\rangle + \frac{\mu}{2} \sum_i \sum_j \frac{p_i}{n_i} \left\| \tilde{\Delta}_{\text{stat}}^{(i)} \right\|^2$$
$$+ \sum_{i \in [m]} p_i \frac{\lambda}{2} \|\tilde{\boldsymbol{w}}^{(g)} - \tilde{\boldsymbol{w}}^{(i)}\|^2 - \sum_{i \in [m]} p_i \frac{\lambda}{2} \|\boldsymbol{w}_\star^{(g)} - \boldsymbol{w}_\star^{(i)}\|^2. \tag{27}$$

Recall the definition in (17) and the Assumption 1 about statistical heterogeneity

$$0 \geq -\sum_i \sum_j \frac{p_i}{n_i} \left\langle \nabla \ell(\boldsymbol{w}_\star^{(i)}, z_{ij}), \tilde{\Delta}_{\text{stat}}^{(i)} \right\rangle + \frac{\mu}{2} \sum_i \sum_j \frac{p_i}{n_i} \left\| \tilde{\Delta}_{\text{stat}}^{(i)} \right\|^2 + \frac{\lambda}{2} \sum_i p_i \left\| \tilde{\boldsymbol{\delta}}^{(i)} \right\|^2 - \frac{\lambda}{2} R^2. \tag{28}$$

Applying Cauchy inequality on the first term of (28), it yields

$$\sum_i \sum_j \frac{p_i}{n_i} \left\langle \nabla \ell(\boldsymbol{w}_\star^{(i)}, z_{ij}), \tilde{\Delta}_{\text{stat}}^{(i)} \right\rangle = \sum_i p_i \left\langle \sum_j \frac{1}{n_i} \nabla \ell(\boldsymbol{w}_\star^{(i)}, z_{ij}), \tilde{\Delta}_{\text{stat}}^{(i)} \right\rangle$$
$$\leq \sum_i p_i \left\| \sum_j \frac{1}{n_i} \nabla \ell(\boldsymbol{w}_\star^{(i)}, z_{ij}) \right\|_2 \left\| \tilde{\Delta}_{\text{stat}}^{(i)} \right\|_2. \tag{29}$$

Consider the Assumption 4 and $z_{ij}$ are i.i.d data

$$\left( \mathbb{E} \left\| \sum_j \frac{1}{n_i} \nabla \ell(\boldsymbol{w}_\star^{(i)}, z_{ij}) \right\|_2 \right)^2 \leq \mathbb{E} \left\| \sum_j \frac{1}{n_i} \nabla \ell(\boldsymbol{w}_\star^{(i)}, z_{ij}) \right\|^2 \leq \frac{\rho^2}{n_i}. \tag{30}$$

Combining (29) and (30), it yields

$$\sum_i \sum_j \frac{p_i}{n_i} \mathbb{E} \left\langle \nabla \ell(\boldsymbol{w}_\star^{(i)}, z_{ij}), \tilde{\Delta}_{\text{stat}}^{(i)} \right\rangle \leq \sum_i \rho \frac{p_i}{\sqrt{n_i}} \mathbb{E} \|\tilde{\Delta}_{\text{stat}}^{(i)}\|_2. \tag{31}$$

Plugging (31) back to (28) yields

$$0 \geq -\rho \sum_i \frac{p_i}{\sqrt{n_i}} \mathbb{E} \left\| \tilde{\Delta}_{\text{stat}}^{(i)} \right\|_2 + \frac{\mu}{2} \sum_i p_i \mathbb{E} \left\| \tilde{\Delta}_{\text{stat}}^{(i)} \right\|^2 + \frac{\lambda}{2} \sum_i p_i \mathbb{E} \left\| \tilde{\boldsymbol{\delta}}^{(i)} \right\|^2 - \frac{\lambda}{2} R^2. \tag{32}$$

After dropping the third term and applying Cauchy inequality, we obtain

$$0 \geq -\rho \left( \left( \sum_{i \in [m]} \frac{p_i}{n_i} \right) \left( \sum_i p_i \mathbb{E} \left\| \tilde{\Delta}_{\text{stat}}^{(i)} \right\|^2 \right) \right)^{\frac{1}{2}} + \frac{\mu}{2} \sum_i p_i \mathbb{E} \left\| \tilde{\Delta}_{\text{stat}}^{(i)} \right\|^2 - \frac{\lambda}{2} R^2. \tag{33}$$

The equation (33) would be regarded as a quadratic function of $\sum_i p_i \mathbb{E} \left\| \tilde{\Delta}_{\text{stat}}^{(i)} \right\|^2$. If we set the $\lambda$ as

$$\lambda \leq \frac{\rho^2}{\mu R^2} \sum_{i \in [m]} \frac{p_i}{n_i}, \tag{34}$$

then solving the inequality in (33) w.r.t $\sum_i p_i \mathbb{E} \left\| \tilde{\Delta}_{\text{stat}}^{(i)} \right\|^2$ yields

$$\mathbb{E}\left\|\tilde{\Delta}_{\text{stat}}^{(g)}\right\|_2 \leq \left(\sum_i p_i \mathbb{E}\left\|\tilde{\Delta}_{\text{stat}}^{(i)}\right\|^2\right)^{\frac{1}{2}} \leq \frac{1}{\mu}\left(\rho\left(\sum_{i\in[m]}\frac{p_i}{n_i}\right)^{\frac{1}{2}} + \left(\rho^2\sum_{i\in[m]}\frac{3p_i}{n_i}\right)^{\frac{1}{2}}\right)$$

$$= \frac{(1+\sqrt{3})\rho}{\mu}\left(\sum_{i\in[m]}\frac{p_i}{n_i}\right)^{\frac{1}{2}}. \tag{35}$$

**Case 2:** Then we consider $R \leq \sqrt{\sum_i \frac{p_i}{n_i}}$. Recall the definition of the global minimizer of Global-Training $\tilde{\boldsymbol{w}}_{\text{GT}} = \arg\min_w \sum_i p_i L_i(w; S_i)$, based on the optimality condition of $\widetilde{\boldsymbol{w}}_i, \widetilde{\boldsymbol{w}}_g, i \in [m]$, we obtain

$$\sum_i p_i\left(L_i(\tilde{\boldsymbol{w}}^{(i)}, S_i) + \frac{\lambda}{2}\|\tilde{\boldsymbol{w}}^{(i)} - \tilde{\boldsymbol{w}}^{(g)}\|^2\right) \leq \sum_i p_i\left(L_i(\tilde{\boldsymbol{w}}_{\text{GT}}, S_i) + \frac{\lambda}{2}\|\tilde{\boldsymbol{w}}_{\text{GT}} - \tilde{\boldsymbol{w}}_{\text{GT}}\|^2\right)$$

$$= \sum_i p_i L_i(\tilde{\boldsymbol{w}}_{\text{GT}}, S_i). \tag{36}$$

Therefore, we obtain

$$\sum_i p_i L_i(\tilde{\boldsymbol{w}}_{\text{GT}}, S_i) \geq \sum_i p_i L_i(\tilde{\boldsymbol{w}}^{(i)}, S_i). \tag{37}$$

In addition, applying the smoothness assumption in Assumption 2 yields

$$\left|L_i(\tilde{\boldsymbol{w}}^{(i)}, S_i) - L_i(\tilde{\boldsymbol{w}}^{(g)}, S_i) - \nabla L_i(\tilde{\boldsymbol{w}}^{(g)}, S_i)^T(\tilde{\boldsymbol{w}}^{(i)} - \tilde{\boldsymbol{w}}^{(g)})\right| \leq \frac{L}{2}\|\tilde{\boldsymbol{w}}^{(i)} - \tilde{\boldsymbol{w}}^{(g)}\|^2. \tag{38}$$

Combining (37) and (38), we have

$$\sum_i p_i L_i(\tilde{\boldsymbol{w}}_{\text{GT}}, S_i) \geq \sum_i p_i L_i(\tilde{\boldsymbol{w}}^{(i)}, S_i) \tag{39}$$

$$\geq \sum_i p_i L_i(\tilde{\boldsymbol{w}}^{(g)}, S_i) + \sum_i p_i \nabla L_i(\tilde{\boldsymbol{w}}^{(g)}, S_i)^\top(\tilde{\boldsymbol{w}}^{(i)} - \tilde{\boldsymbol{w}}^{(g)}) - \sum_i p_i\frac{L}{2}\|\tilde{\boldsymbol{w}}^{(i)} - \tilde{\boldsymbol{w}}^{(g)}|^2. \tag{40}$$

Using Cauchy inequality on the second term of the R.H.S.,

$$\sum_i p_i L_i(\tilde{\boldsymbol{w}}_{\text{GT}}, S_i) \geq \sum_i p_i L_i(\tilde{\boldsymbol{w}}^{(g)}, S_i) - \sum_i p_i\|\nabla L_i(\tilde{\boldsymbol{w}}^{(g)}, S_i)\|_2\|\tilde{\boldsymbol{w}}^{(i)} - \tilde{\boldsymbol{w}}^{(g)}\|_2$$

$$- \sum_i p_i\frac{L}{2}\|\tilde{\boldsymbol{w}}^{(i)} - \tilde{\boldsymbol{w}}^{(g)}\|^2. \tag{41}$$

On the other hand, note that the optimality condition of $\tilde{\boldsymbol{w}}_{\text{GT}}$ is

$$\sum_i p_i \nabla L_i(\tilde{\boldsymbol{w}}_{\text{GT}}, S_i) = 0, \tag{42}$$

Using the strong-convexity assumption in Assumption 3, we can show that

$$\sum_i p_i \left\{ L_i(\tilde{\boldsymbol{w}}^{(g)}, S_i) - L_i(\tilde{\boldsymbol{w}}_{\mathrm{GT}}, S_i) - \nabla L_i(\tilde{\boldsymbol{w}}_{\mathrm{GT}}, S_i)^T (\tilde{\boldsymbol{w}}^{(g)} - \tilde{\boldsymbol{w}}_{\mathrm{GT}}) \right\}$$

$$= \sum_i p_i \left\{ L_i(\tilde{\boldsymbol{w}}^{(g)}, S_i) - L_i(\tilde{\boldsymbol{w}}_{\mathrm{GT}}, S_i) \right\}$$

$$\geq \sum_i p_i \frac{\mu}{2} \|\tilde{\boldsymbol{w}}^{(g)} - \tilde{\boldsymbol{w}}_{\mathrm{GT}}\|^2$$

$$= \frac{\mu}{2} \|\tilde{\boldsymbol{w}}^{(g)} - \tilde{\boldsymbol{w}}_{\mathrm{GT}}\|^2. \tag{43}$$

Combining the results (41) and (43), it yields

$$\sum_i p_i L_i(\tilde{\boldsymbol{w}}_{\mathrm{GT}}, S_i) \geq \sum_i p_i L_i(\tilde{\boldsymbol{w}}_{\mathrm{GT}}, S_i) + \frac{\mu}{2} \|\tilde{\boldsymbol{w}}^{(g)} - \tilde{\boldsymbol{w}}_{\mathrm{GT}}\|^2$$

$$- \sum_i p_i \|\nabla L_i(\tilde{\boldsymbol{w}}^{(g)}, S_i)\|_2 \|\tilde{\boldsymbol{w}}^{(i)} - \tilde{\boldsymbol{w}}^{(g)}\|_2 - \sum_i p_i \frac{L}{2} \|\tilde{\boldsymbol{w}}^{(i)} - \tilde{\boldsymbol{w}}^{(g)}\|^2. \tag{44}$$

Reorganizing these terms yields

$$\frac{\mu}{2} \|\tilde{\boldsymbol{w}}^{(g)} - \tilde{\boldsymbol{w}}_{\mathrm{GT}}\|^2 \leq \sum_i p_i \left\{ \|\nabla L_i(\tilde{\boldsymbol{w}}^{(g)}, S_i)\|_2 \|\tilde{\boldsymbol{w}}^{(i)} - \tilde{\boldsymbol{w}}^{(g)}\|_2 + \frac{L}{2} \|\tilde{\boldsymbol{w}}^{(i)} - \tilde{\boldsymbol{w}}^{(g)}\|^2 \right\}. \tag{45}$$

Using Lemma 4 to bound the term $\|\tilde{\boldsymbol{w}}^{(i)} - \tilde{\boldsymbol{w}}^{(g)}\|^2$, we have

$$\|\tilde{\boldsymbol{w}}^{(g)} - \tilde{\boldsymbol{w}}_{\mathrm{GT}}\|^2 \leq \frac{2}{\mu} \sum_i p_i \left[ \frac{2}{\mu + \lambda} + \frac{2L}{(\mu + \lambda)^2} \right] \|\nabla L_i(\tilde{\boldsymbol{w}}^{(g)}, S_i)\|^2$$

$$= \frac{4}{\mu} \left( \frac{1}{\mu + \lambda} + \frac{L}{(\mu + \lambda)^2} \right) \sum_i p_i \|\nabla L_i(\tilde{\boldsymbol{w}}^{(g)}, S_i)\|^2. \tag{46}$$

Note that by the smoothness Assumption and triangle inequality,

$$\|\nabla L_i(\tilde{\boldsymbol{w}}^{(g)}, S_i)\|_2^2 \leq 2\|\nabla L_i(\tilde{\boldsymbol{w}}^{(g)}, S_i) - \nabla L_i(\boldsymbol{w}_\star^{(i)}, S_i)\|^2 + 2\|\nabla L_i(\boldsymbol{w}_\star^{(i)}, S_i)\|^2$$

$$\leq 2L^2 \|\tilde{\boldsymbol{w}}^{(g)} - \boldsymbol{w}_\star^{(i)}\|^2 + 2\|\nabla L_i(\boldsymbol{w}_\star^{(i)}, S_i)\|^2. \tag{47}$$

Take expectation w.r.t $S_i$ and use Assumption 1 and 4, then we have

$$\mathbb{E}[\|\nabla L_i(\tilde{\boldsymbol{w}}^{(g)}, S_i)\|^2] \leq 2L^2 \mathbb{E}[\|\tilde{\boldsymbol{w}}^{(g)} - \boldsymbol{w}_\star^{(i)}\|^2] + 2\mathbb{E}\left[ \left\| \frac{1}{n_i} \sum_{j=1}^{n_i} \nabla L(\boldsymbol{w}_\star^{(i)}, z_{ij}) \right\|^2 \right]$$

$$\leq 4L^2 \mathbb{E}[\|\tilde{\boldsymbol{w}}^{(g)} - \boldsymbol{w}_\star^{(g)}\|^2] + 4L^2 R^2 + 2\frac{\rho^2}{n}, \tag{48}$$

where the last term comes from (30).

Plugging (48) back into (46), it yields

$$\mathbb{E}[\|\tilde{\boldsymbol{w}}^{(g)} - \tilde{\boldsymbol{w}}_{\mathrm{GT}}\|^2] \leq \frac{4}{\mu} \left( \frac{1}{\mu + \lambda} + \frac{L}{(\mu + \lambda)^2} \right) \left\{ 4L^2 \mathbb{E}[\|\tilde{\boldsymbol{w}}^{(g)} - \boldsymbol{w}_\star^{(g)}\|^2] + 4L^2 R^2 + 2\frac{\rho^2}{n} \right\}. \tag{49}$$

Next, we are ready to establish the global error bound. Using the result (49) and Lemma 5, we obtain

$$\mathbb{E}[\|\tilde{\boldsymbol{w}}^{(g)} - \boldsymbol{w}_\star^{(g)}\|^2] \leq \frac{4}{\mu} \left( \frac{1}{\mu+\lambda} + \frac{L}{(\mu+\lambda)^2} \right) \left\{ 4L^2 \mathbb{E}[\|\tilde{\boldsymbol{w}}^{(g)} - \boldsymbol{w}_\star^{(g)}\|^2] + 4L^2 R^2 + 2\frac{\rho^2}{n} \right\}$$
$$+ \frac{8\rho^2}{\mu^2} \frac{1}{N} + \frac{8L^2}{\mu^2} R^2. \tag{50}$$

Let's define $g(\lambda) = \frac{4}{\mu} \left( \frac{1}{\mu+\lambda} + \frac{L}{(\mu+\lambda)^2} \right)$ and reorganize (50), which yields

$$\mathbb{E}[\|\tilde{\boldsymbol{w}}^{(g)} - \boldsymbol{w}_\star^{(g)}\|^2] \leq \frac{g(\lambda) \left( 4L^2 R^2 + 2\frac{\rho^2}{n} \right) + \frac{8\rho^2}{\mu^2} \frac{1}{N} + \frac{8L^2}{\mu^2} R^2}{1 - 4L^2 g(\lambda)}. \tag{51}$$

If we assume

$$g(\lambda) \leq \frac{1}{8L^2} \wedge \left( \left( \frac{4\rho^2}{\mu^2 N} + \frac{4L^2 R^2}{\mu^2} \right) \Big/ \left( 2L^2 R^2 + \frac{\rho^2}{n} \right) \right), \tag{52}$$

we have

$$\mathbb{E}\|\tilde{\boldsymbol{w}}^{(g)} - \boldsymbol{w}_\star^{(g)}\|^2 \leq 32 \left( \frac{\rho^2}{\mu^2} \frac{1}{N} + \frac{L^2}{\mu^2} R^2 \right). \tag{53}$$

Furthermore, we can get a simplified condition for $\lambda$ as

$$\lambda \geq \frac{8\kappa}{\mu} \left\{ 8L^2 \vee \left( \left( 2L^2 R^2 + \frac{\rho^2}{n} \right) \Big/ \left( \frac{4\rho^2}{\mu^2 N} + \frac{4L^2 R^2}{\mu^2} \right) \right) \right\} - \mu \tag{54}$$

where $\kappa = \frac{L}{\mu} \geq 1$ and the condition that $\lambda$ is non-negative holds trivially as $\frac{\kappa L^2}{\mu} = \kappa^2 L \geq \mu$.

Combine two-case arguments, we have

$$\Rightarrow \mathbb{E}\|\tilde{\boldsymbol{w}}^{(g)} - \boldsymbol{w}_\star^{(g)}\|^2 \leq \begin{cases} \left( \frac{(1+\sqrt{3})\rho}{\mu} \right)^2 \sum_{i \in [m]} \frac{p_i}{n_i} & R > \sqrt{\sum_i \frac{p_i}{n_i}} \\ 32 \left( \frac{\rho^2}{\mu^2} \frac{1}{N} + \frac{L^2}{\mu^2} R^2 \right) & R \leq \sqrt{\sum_i \frac{p_i}{n_i}}. \end{cases} \tag{55}$$

Thus it yields a one-line rate

$$\mathbb{E}\|\tilde{\boldsymbol{w}}^{(g)} - \boldsymbol{w}_\star^{(g)}\|^2 \leq C_1 \frac{1}{N} + C_2 \Big( \sum_{i \in [m]} \frac{p_i}{n_i} \wedge R^2 \Big), \tag{56}$$

where

$$C_1 = 32 \frac{\rho^2}{\mu^2},$$
$$C_2 = \left( \frac{(1+\sqrt{3})\rho}{\mu} \right)^2 \vee \frac{32L^2}{\mu^2}. \tag{57}$$

Moreover, if we assume data balanced, i.e $n_i = n, \forall i \in [m]$ and $p_i = \frac{1}{m}$, we obtain

$$\mathbb{E}\|\tilde{\boldsymbol{w}}^{(g)} - \boldsymbol{w}_\star^{(g)}\|^2 \leq C_1 \frac{1}{N} + C_2 \left( \frac{1}{n} \wedge R^2 \right), \tag{58}$$

as claimed.

### A.3.3 Proof of Local Statistical Error Bound in Theorem 1

**Case 1**: first we consider the case when $R > \sqrt{\sum_i \frac{p_i}{n_i}}$.

To prove the local statistical error bound, we start with the optimality condition of $\tilde{\boldsymbol{w}}^{(i)}$ and $\tilde{\boldsymbol{w}}^{(g)}$ for a single client, which yields that for $i \in [m]$,

$$0 \geq L_i\left(\tilde{\boldsymbol{w}}^{(i)}, S_i\right) + \frac{\lambda}{2}\left\|\tilde{\boldsymbol{w}}^{(g)} - \tilde{\boldsymbol{w}}^{(i)}\right\|^2 - L_i\left(\boldsymbol{w}_\star^{(i)}, S_i\right) - \frac{\lambda}{2}\left\|\tilde{\boldsymbol{w}}^{(g)} - \boldsymbol{w}_\star^{(i)}\right\|^2. \tag{59}$$

Reorganized the terms, we obtain

$$0 \geq \sum_{j \in [n_i]} \frac{1}{n_i}\left(\ell\left(\tilde{\boldsymbol{w}}^{(i)}, z_{ij}\right) - \ell\left(\boldsymbol{w}_\star^{(i)}, z_{ij}\right)\right) + \frac{\lambda}{2}\left\|\tilde{\boldsymbol{w}}^{(g)} - \tilde{\boldsymbol{w}}^{(i)}\right\|^2 - \frac{\lambda}{2}\left\|\tilde{\boldsymbol{w}}^{(g)} - \boldsymbol{w}_\star^{(i)}\right\|^2. \tag{60}$$

For the first term on the RHS, applying the $\mu$-strongly convex of the loss function $\ell$ (cf. Assumption 3) yields

$$\sum_{j \in [n_i]} \frac{1}{n_i}\left(\ell(\tilde{\boldsymbol{w}}^{(i)}, z_{ij}) - \ell(\boldsymbol{w}_\star^{(i)}, z_{ij})\right) \geq -\sum_{j \in [n_i]} \frac{1}{n_i}\left\langle\nabla\ell(\boldsymbol{w}_\star^{(i)}, z_{ij}), \tilde{\Delta}_{\text{stat}}^{(i)}\right\rangle + \frac{\mu}{2}\sum_{j \in [n_i]} \frac{1}{n_i}\left\|\tilde{\Delta}_{\text{stat}}^{(i)}\right\|^2, \tag{61}$$

where we use the definition of $\tilde{\Delta}_{\text{stat}}^{(i)}$ in (16).

Pluggging (61) back into (60), it yields

$$0 \geq -\sum_j \frac{1}{n_i}\left\langle\nabla\ell(\boldsymbol{w}_\star^{(i)}, z_{ij}), \tilde{\Delta}_{\text{stat}}^{(i)}\right\rangle + \frac{\mu}{2}\sum_j \frac{1}{n_i}\left\|\tilde{\Delta}_{\text{stat}}^{(i)}\right\|^2 + \frac{\lambda}{2}\|\tilde{\boldsymbol{w}}^{(g)} - \tilde{\boldsymbol{w}}^{(i)}\|^2 - \frac{\lambda}{2}\|\tilde{\boldsymbol{w}}^{(g)} - \boldsymbol{w}_\star^{(i)}\|^2. \tag{62}$$

Apply Cauchy inequality on the first term of (62)

$$\begin{aligned}
\sum_j \frac{1}{n_i}\left\langle\nabla\ell(\boldsymbol{w}_\star^{(i)}, z_{ij}), \tilde{\Delta}_{\text{stat}}^{(i)}\right\rangle &= \left\langle\sum_j \frac{1}{n_i}\nabla\ell(\boldsymbol{w}_\star^{(i)}, z_{ij}), \tilde{\Delta}_{\text{stat}}^{(i)}\right\rangle \\
&\leq \left\|\sum_j \frac{1}{n_i}\nabla\ell(\boldsymbol{w}_\star^{(i)}, z_{ij})\right\|_2 \left\|\tilde{\Delta}_{\text{stat}}^{(i)}\right\|_2.
\end{aligned} \tag{63}$$

Consider the Assumption 4 and $z_{ij}$ are i.i.d data

$$\left(\mathbb{E}\left\|\sum_j \frac{1}{n_i}\nabla\ell(\boldsymbol{w}_\star^{(i)}, z_{ij})\right\|_2\right)^2 \leq \mathbb{E}\left\|\sum_j \frac{1}{n_i}\nabla\ell(\boldsymbol{w}_\star^{(i)}, z_{ij})\right\|^2 \leq \frac{\rho^2}{n_i}. \tag{64}$$

Combining (63) and (64), it yields

$$\sum_j \frac{1}{n_i}\mathbb{E}\left\langle\nabla\ell(\boldsymbol{w}_\star^{(i)}, z_{ij}), \tilde{\Delta}_{\text{stat}}^{(i)}\right\rangle \leq \rho\frac{1}{\sqrt{n_i}}\mathbb{E}\|\tilde{\Delta}_{\text{stat}}^{(i)}\|_2. \tag{65}$$

Plugging (65) back to (62) yields

$$0 \geq -\rho \frac{1}{\sqrt{n_i}} \mathbb{E} \left\| \tilde{\Delta}_{\text{stat}}^{(i)} \right\|_2 + \frac{\mu}{2} \mathbb{E} \left\| \tilde{\Delta}_{\text{stat}}^{(i)} \right\|^2 + \frac{\lambda}{2} \mathbb{E} \| \tilde{\boldsymbol{w}}^{(g)} - \tilde{\boldsymbol{w}}^{(i)} \|^2 - \frac{\lambda}{2} \mathbb{E} \| \tilde{\boldsymbol{w}}^{(g)} - \boldsymbol{w}_\star^{(i)} \|^2. \tag{66}$$

To utilize the statistical global convergence rate shown in Appendix A.3.2, we need to discuss $R$ in the same way. Recall the upper bound of $\tilde{\Delta}_{\text{stat}}^{(g)}$ in (58):

$$\mathbb{E} \| \tilde{\Delta}_{\text{stat}}^{(g)} \|^2 \leq \begin{cases} \frac{C_2}{n} & R > \frac{1}{\sqrt{n}} \\ \frac{C_1}{N} + C_2 R^2 & R \leq \frac{1}{\sqrt{n}}. \end{cases} \tag{67}$$

For simplicity, in the following arguments, we denote the upper bound of $\mathbb{E} \| \tilde{\Delta}_{\text{stat}}^{(g)} \|_2$ as $U_0$.

Note that

$$-\| \tilde{\boldsymbol{w}}^{(g)} - \boldsymbol{w}_\star^{(i)} \|^2 = -\left\| \tilde{\boldsymbol{w}}^{(g)} - \boldsymbol{w}_\star^{(g)} + \boldsymbol{w}_\star^{(g)} - \boldsymbol{w}_\star^{(i)} \right\|^2$$

$$\geq -\| \tilde{\Delta}_{\text{stat}}^{(g)} \|^2 - \| \boldsymbol{w}_\star^{(g)} - \boldsymbol{w}_\star^{(i)} \|^2 - 2\| \boldsymbol{w}_\star^{(g)} - \boldsymbol{w}_\star^{(i)} \|_2 \left\| \tilde{\Delta}_{\text{stat}}^{(g)} \right\|_2 \tag{68}$$

$$\geq -\| \tilde{\Delta}_{\text{stat}}^{(g)} \|^2 - R^2 - 2R \left\| \tilde{\Delta}_{\text{stat}}^{(g)} \right\|_2.$$

Plug (68) into (66)

$$0 \geq -\frac{\rho}{\sqrt{n_i}} \mathbb{E} \left\| \tilde{\Delta}_{\text{stat}}^{(i)} \right\|_2 + \frac{\mu}{2} \mathbb{E} \left\| \tilde{\Delta}_{\text{stat}}^{(i)} \right\|^2 + \frac{\lambda}{2} \mathbb{E} \left\| \tilde{\boldsymbol{\delta}}^{(i)} \right\|^2 - \frac{\lambda}{2} \mathbb{E} \left\| \tilde{\boldsymbol{w}}^{(g)} - \boldsymbol{w}_\star^{(g)} + \boldsymbol{w}_\star^{(g)} - \boldsymbol{w}_\star^{(i)} \right\|^2$$

$$\geq -\frac{\rho}{\sqrt{n_i}} \mathbb{E} \left\| \tilde{\Delta}_{\text{stat}}^{(i)} \right\|_2 + \frac{\mu}{2} \mathbb{E} \left\| \tilde{\Delta}_{\text{stat}}^{(i)} \right\|^2 + \frac{\lambda}{2} \mathbb{E} \left\| \tilde{\boldsymbol{\delta}}^{(i)} \right\|^2 - \frac{\lambda}{2} \mathbb{E} \left\| \tilde{\Delta}_{\text{stat}}^{(g)} \right\|^2 - \frac{\lambda}{2} R^2 - \lambda R \mathbb{E} \left\| \tilde{\Delta}_{\text{stat}}^{(g)} \right\|_2.$$
$$\tag{69}$$

Putting (67) into (69) and reorganizing, we obtain

$$0 \geq -\frac{\rho}{\sqrt{n_i}} \mathbb{E} \left\| \tilde{\Delta}_{\text{stat}}^{(i)} \right\|_2 + \frac{\mu}{2} \mathbb{E} \left\| \tilde{\Delta}_{\text{stat}}^{(i)} \right\|^2 + \frac{\lambda}{2} \mathbb{E} \left\| \tilde{\boldsymbol{\delta}}^{(i)} \right\|^2 - \frac{\lambda}{2} U_0^2 - \frac{\lambda}{2} R^2 - \lambda R U_0$$

$$\geq \frac{\mu}{2} \mathbb{E} \left\| \tilde{\Delta}_{\text{stat}}^{(i)} \right\|^2 - \frac{\rho}{\sqrt{n_i}} \mathbb{E} \left\| \tilde{\Delta}_{\text{stat}}^{(i)} \right\|_2 - \frac{\lambda}{2} (U_0 + R)^2. \tag{70}$$

In addition, recall that in this case we set $\lambda^\star \leq \frac{\rho^2}{\mu R^2} \sum_{i \in [m]} \frac{p_i}{n_i}$ in the statistical convergence analysis of the global model. If we assume data balanced, i.e. $p_i = \frac{1}{m}$ and $n_i = n, \forall i \in [m]$, it yields $\lambda^\star \leq \frac{\rho^2}{\mu n R^2}$ and

$$\mathbb{E} \| \tilde{\Delta}_{\text{stat}}^{(i)} \|_2 \leq \frac{1}{\mu} \left[ \frac{\rho}{\sqrt{n}} + \left( \frac{\rho^2}{n} + \frac{2\rho^2}{nR^2} (U_0 + R)^2 \right)^{\frac{1}{2}} \right].$$

$$= \frac{1}{\sqrt{n}} \frac{1}{\mu} \left( \rho + \left( \rho^2 + \frac{2\rho^2}{R^2} (U_0 + R)^2 \right)^{\frac{1}{2}} \right)$$

$$\leq \frac{1}{\sqrt{n}} \frac{\rho + (\rho^2 + 4\rho^2 (C_2 + 1))^{\frac{1}{2}}}{\mu} \tag{71}$$

$$= \frac{1}{\sqrt{n}} \frac{\rho(1 + (1 + 4(C_2 + 1))^{\frac{1}{2}})}{\mu}.$$

**Case 2:** we then consider the case when $R \leq \sqrt{\sum_i \frac{p_i}{n_i}}$.

$$\mathbb{E}[\|\tilde{\boldsymbol{w}}^{(i)} - \boldsymbol{w}_\star^{(i)}\|^2]$$

$$\leq 3\mathbb{E}[\|\tilde{\boldsymbol{w}}^{(i)} - \tilde{\boldsymbol{w}}^{(g)}\|^2] + 3\mathbb{E}[\|\tilde{\boldsymbol{w}}^{(g)} - \boldsymbol{w}_\star^{(g)}\|^2] + 3\mathbb{E}[\|\boldsymbol{w}_\star^{(g)} - \boldsymbol{w}_\star^{(i)}\|^2]$$

$$\overset{(a)}{\leq} \frac{12}{(\mu+\lambda)^2}\mathbb{E}[\|\nabla L_i(\tilde{\boldsymbol{w}}^{(g)})\|^2] + 3\mathbb{E}[\|\tilde{\boldsymbol{w}}^{(g)} - \boldsymbol{w}_\star^{(g)}\|^2] + 3R^2$$

$$\overset{(b)}{\leq} \frac{12}{(\mu+\lambda)^2}\left[4L^2\mathbb{E}[\|\tilde{\boldsymbol{w}}^{(g)} - \boldsymbol{w}_\star^{(g)}\|^2] + 4L^2R^2 + 2\frac{\rho^2}{n}\right] + 3\mathbb{E}[\|\tilde{\boldsymbol{w}}^{(g)} - \boldsymbol{w}_\star^{(g)}\|^2] + 3R^2$$

$$\overset{(c)}{\leq} \frac{12}{(\mu+\lambda)^2}\left[128L^2\left(\frac{\rho^2}{\mu^2}\frac{1}{N} + \frac{L^2}{\mu^2}R^2\right) + 4L^2R^2 + 2\frac{\rho^2}{n}\right] + 96\left(\frac{\rho^2}{\mu^2}\frac{1}{N} + \frac{L^2}{\mu^2}R^2\right) + 3R^2$$

$$= \frac{12}{(\mu+\lambda)^2}\left[\frac{128L^2\rho^2}{\mu^2}\frac{1}{N} + (128L^4/\mu^2 + 4L^2)R^2 + 2\frac{\rho^2}{n}\right] + \frac{96\rho^2}{\mu^2}\frac{1}{N} + (96\frac{L^2}{\mu^2} + 3)R^2 \tag{72}$$

where step (a) comes from Assumption 1 and Lemma 4, step (b) comes from result (48) and step (c) is a direct result of applying the error bound of $\mathbb{E}\|\tilde{\Delta}_{\text{stat}}^{(g)}\|^2$.

If we assume $\lambda$ satisfies the following condition

$$\frac{12}{(\mu+\lambda)^2} \leq \frac{96\rho^2/(\mu^2N) + 99\kappa^2R^2}{128\rho^2/N + 132L^2R^2 + 2\rho^2/n} \tag{73}$$

we have

$$\mathbb{E}[\|\tilde{\boldsymbol{w}}^{(i)} - \boldsymbol{w}_\star^{(i)}\|^2] \leq \frac{192\rho^2}{\mu^2}\frac{1}{N} + 198\kappa^2R^2 \tag{74}$$

Furthermore, we can get a simplified condition for $\lambda$ as

$$\lambda \geq \sqrt{\frac{8(64\rho^2/N + 66L^2R^2 + \rho^2/n)}{32\rho^2/(\mu^2N) + 33\kappa^2R^2}} - \mu \tag{75}$$

Combining two-case arguments, we have:

$$\Rightarrow \mathbb{E}\left\|\tilde{\Delta}_{\text{stat}}^{(i)}\right\|^2 \leq \begin{cases} \frac{1}{n}\left(\frac{\rho(1+(1+4(C_2+1))^{\frac{1}{2}})}{\mu}\right)^2 & R > \sqrt{\frac{m}{N}} \\ \frac{192\rho^2}{\mu^2}\frac{1}{N} + 198\kappa^2R^2 & R \leq \sqrt{\frac{m}{N}} \end{cases}. \tag{76}$$

And it yields a one-line rate

$$\mathbb{E}\left\|\tilde{\Delta}_{\text{stat}}^{(i)}\right\|^2 \leq C_3\frac{1}{N} + C_4(R^2 \wedge \frac{1}{n}), \tag{77}$$

where

$$C_3 = \frac{192\rho^2}{\mu^2}, \tag{78}$$

$$C_4 = \left(\frac{\rho(1 + (4C_2+5)^{\frac{1}{2}})}{\mu}\right)^2 \vee 198\kappa^2, \tag{79}$$

$$\tag{80}$$

and $C_2$ are specified in (57).

Combining the results in A.3.2 and A.3.3, we leave a remark below to further interpret the effect of personalization in obj (1) and discuss how to incorporate the conditions of $\lambda$ when establishing the global and local statistical accuracy.

**Remark 2.** *To better understand how such an interpolation is achieved through an adaptive personalization degree, we examine the role of $\lambda$ in navigating between the two extreme cases: GlobalTrain and LocalTrain. As the statistical heterogeneity $R \to \infty$, we have $\lambda \to 0$, leading to a high degree of personalization. The statistical error bound becomes $\mathcal{O}(n^{-1})$, matching the rate of LocalTrain; as the statistical heterogeneity $R \to 0$, $\lambda$ increases, leading to a low degree of personalization. The statistical error bound eventually converges to $\mathcal{O}(N^{-1})$, matching the rate of GlobalTrain under a homogeneous setting. As $R \in (0, \infty)$, $\lambda$ transits between the two extremes, making FedProx perform no worse than LocalTrain while benefiting from global training when clients share similarities.*

*To make the global statistical error bound and local statistical error bound hold simultaneously, we should be aware about the condition of the adaptive strategy of $\lambda$ specified in our proof. It could be summarized as*

$$\begin{cases} \lambda \geq max(a_1, a_2) & R < \sqrt{\frac{m}{N}} \\ \lambda \leq \frac{\rho^2}{n\mu R^2} & R \geq \sqrt{\frac{m}{N}} \end{cases}, \tag{81}$$

*where*

$$a_1 = \frac{8\kappa}{\mu} \left\{ 8L^2 \vee \left( \frac{2L^2R^2 + \rho^2/n}{4\rho^2/(\mu^2 N) + 4\kappa^2 R^2} \right) \right\} - \mu,$$

*and*

$$a_2 = \sqrt{\frac{8(64\rho^2/N + 66L^2R^2 + \rho^2/n)}{32\rho^2/(\mu^2 N) + 33\kappa^2 R^2}} - \mu.$$

*We can induce a sufficient condition of $\lambda$ for $R < \frac{1}{\sqrt{n}}$ as*

$$\lambda \geq max \left\{ 64\kappa^2 L, (2\kappa \vee 5)\,\mu \frac{2L^2R^2 + \rho^2/n}{L^2R^2 + \rho^2/N} \right\} - \mu. \tag{82}$$

## B  OPTIMIZATION CONVERGENCE RATE

### B.1  ALGORITHM

---

**Algorithm 1:** One-stage FedProx

---

**Input:** Initial global model $\boldsymbol{w}_1^{(g)}$, initial local models $\{\boldsymbol{w}_{0,K}^{(i)}\}_{i \in [m]}$, global rounds $T$, global step sizes $\gamma$, local rounds $K$, local step sizes $\eta$

**Output:** Local models $\{\boldsymbol{w}_{T,K}^{(i)}\}_{i \in [m]}$ and global model $\boldsymbol{w}_T^{(g)}$

**for** $t = 1, \ldots, T$ **do**

    The server sends $\boldsymbol{w}_t^{(g)}$ to client $i$, $\forall i \in [m]$

    Set $\boldsymbol{w}_{t,0}^{(i)} = \boldsymbol{w}_{t-1,K}^{(i)}$;

    **for** $k = 0, \ldots, K-1$ **do**

        $\boldsymbol{w}_{t,k+1}^{(i)} = \boldsymbol{w}_{t,k}^{(i)} - \frac{\eta}{n_i} \sum_{j \in [n_i]} \left\{ \nabla \ell(\boldsymbol{w}_{t,k}^{(i)}, z_{i,j}) + \lambda \left( \boldsymbol{w}_{t,k}^{(i)} - \boldsymbol{w}_t^{(g)} \right) \right\}$

    Push $\widehat{\nabla} F_i(\boldsymbol{w}_t^{(g)}) = \lambda(\boldsymbol{w}_t^{(g)} - \boldsymbol{w}_{t,K}^{(i)})$ to the server;

    $\boldsymbol{w}_{t+1}^{(g)} \leftarrow \boldsymbol{w}_t^{(g)} - \frac{\gamma}{m} \sum_{i \in [m]} \widehat{\nabla} F_i(\boldsymbol{w}_t^{(g)})$;

---

### B.2  PROOF OF LEMMAS

#### B.2.1  PROOF OF LEMMA 2

To facilitate our analysis, we state a claim first.

**Claim**: If $f$ is $\mu$-strongly-convex, then the proximal operator

$$\text{prox}_{f/\lambda}(x) = \arg \min_v \left( f(v) + \frac{\lambda}{2} \|x - v\|^2 \right), \tag{83}$$

is $L_w$- Lipschitz continuous with $L_w = \frac{\lambda}{\lambda + \mu}$.

*Proof.* Based on the definition of $\text{prox}_{f/\lambda}(x)$, it will satisfy the first order condition

$$\nabla f(\text{prox}_{\frac{1}{\lambda} f}(x)) - \lambda(x - \text{prox}_{\frac{1}{\lambda} f}(x)) = 0. \tag{84}$$

Then we have

$$\nabla f(x_1) - \lambda(v_1 - x_1) = 0 \quad \text{where } x_1 = \text{prox}_{\frac{1}{\lambda} f}(v), \tag{85}$$

$$\nabla f(x_2) - \lambda(v_2 - x_2) = 0 \quad \text{where } x_2 = \text{prox}_{\frac{1}{\lambda} f}(v_2). \tag{86}$$

Based on the $\mu$-strongly convexity of $f$, we have

$$\langle \nabla f(x_1) - \nabla f(x_2), x_1 - x_2 \rangle \geq \mu \|x_1 - x_2\|^2 \tag{87}$$

$$\Rightarrow \|\text{prox}_{\frac{1}{\lambda} f}(x_1) - \text{prox}_{\frac{1}{\lambda} f}(x_2)\| \leq \frac{1}{1 + \frac{\mu}{\lambda}} \|x_1 - x_2\|. \tag{88}$$

$\square$

Now, we are ready to prove Lemma 2.

First, from Mishchenko et al. (2023), we can directly obtain the analytic properties of $F_i$. Next, the analytic properties of $h_i$ are direct results from Assumption 3 and 2. Finally, to analyze the property of the mapping $w_\star^{(i)}$, we just need to use the result from the above claim as we know the definition of $\boldsymbol{w}_\star^{(i)}(\cdot)$ in (9).

### B.2.2 INNER LOOP CONVERGENCE

**Lemma 6** ((Ji et al., 2022), Inner Loop Error Contraction). *Suppose Assumption 3 and 2 hold, then if $\eta \leq 1/L_\ell$, for all $t \geq 0$ and $\epsilon > 0$, the inner loop error bound of all clients can be formulated as:*

$$e_{t+1}^{(i)} := \|\boldsymbol{w}_{t+1,K}^{(i)} - \boldsymbol{w}_{t+1,\star}^{(i)}\|^2 \leq (1+\epsilon)(1-\eta\mu_\ell)^K e_t^{(i)} + \left(1 + \frac{1}{\epsilon}\right)(1-\eta\mu_\ell)^K \|\boldsymbol{w}_{t,\star}^{(i)} - \boldsymbol{w}_{t-1,\star}^{(i)}\|^2. \tag{89}$$

*Proof.* The proof can be found in Lemma 2 (Ji et al., 2022). ∎

### B.2.3 OUTER LOOP CONVERGENCE

**Lemma 7** (Outer Loop Error Contraction). *Suppose Assumption 3 and 2 hold, then if $\gamma \leq L_g$, we have for all $t > 0$ and $\epsilon_g > 0$:*

$$\|\boldsymbol{w}_{t+1}^{(g)} - \widetilde{\boldsymbol{w}}^{(g)}\|^2 \leq (1 - \gamma\mu_g + \gamma\epsilon_g)\|\boldsymbol{w}_t^{(g)} - \widetilde{\boldsymbol{w}}^{(g)}\|^2 - \gamma^2\|\nabla F(\boldsymbol{w}_t^{(g)})\|^2$$
$$+ (\gamma^3 L_g)\|\widehat{\nabla} F(\boldsymbol{w}_t^{(g)})\|^2 + \left(\frac{\gamma}{\epsilon_g} + \gamma^2\right)\|\nabla F(\boldsymbol{w}_t^{(g)}) - \widehat{\nabla} F(\boldsymbol{w}_t^{(g)})\|^2. \tag{90}$$

*Proof.* Note

$$\widehat{\nabla} F_i(\boldsymbol{w}_t^{(g)}) = \lambda(\boldsymbol{w}_t^{(g)} - \boldsymbol{w}_{t,K}^{(i)}). \tag{91}$$

We define

$$\widehat{\nabla} F(\boldsymbol{w}_t^{(g)}) := \frac{1}{m} \sum_{i \in [m]} \widehat{\nabla} F_i(\boldsymbol{w}_t^{(g)}) = \frac{1}{m} \sum_{i \in [m]} \lambda(\boldsymbol{w}_t^{(g)} - \boldsymbol{w}_{t,K}^{(i)}). \tag{92}$$

Thus the global update rule can be written as $\boldsymbol{w}_{t+1}^{(g)} = \boldsymbol{w}_t^{(g)} - \gamma\widehat{\nabla} F(\boldsymbol{w}_t^{(g)})$. For the outer loop,

$$\|\boldsymbol{w}_{t+1}^{(g)} - \widetilde{\boldsymbol{w}}^{(g)}\|^2$$
$$= \|\boldsymbol{w}_t^{(g)} - \widetilde{\boldsymbol{w}}^{(g)} - \gamma\widehat{\nabla} F(\boldsymbol{w}_t^{(g)})\|^2$$
$$= \|\boldsymbol{w}_t^{(g)} - \widetilde{\boldsymbol{w}}^{(g)}\|^2 - 2\gamma\langle\boldsymbol{w}_t^{(g)} - \widetilde{\boldsymbol{w}}^{(g)}, \widehat{\nabla} F(\boldsymbol{w}_t^{(g)})\rangle + \gamma^2\|\widehat{\nabla} F(\boldsymbol{w}_t^{(g)})\|^2 \tag{93}$$
$$\leq (1 + \epsilon_g\gamma)\|\boldsymbol{w}_t^{(g)} - \widetilde{\boldsymbol{w}}^{(g)}\|^2 - 2\gamma\langle\boldsymbol{w}_t^{(g)} - \widetilde{\boldsymbol{w}}^{(g)}, \nabla F(\boldsymbol{w}_t^{(g)})\rangle + \gamma^2\|\widehat{\nabla} F(\boldsymbol{w}_t^{(g)})\|^2$$
$$+ \left(\frac{\gamma}{\epsilon_g}\right)\|\widehat{\nabla} F(\boldsymbol{w}_t^{(g)}) - \nabla F(\boldsymbol{w}_t^{(g)})\|^2,$$

where we apply the Young's inequality with $\epsilon_g > 0$ in the last line.

Next, we bound the inner product term in the last line using the $\mu_g$-strong convexity of $F$:

$$-\gamma\langle\boldsymbol{w}_t^{(g)} - \widetilde{\boldsymbol{w}}^{(g)}, \nabla F(\boldsymbol{w}_t^{(g)})\rangle \leq -\gamma\left(F(\boldsymbol{w}_t^{(g)}) - \widehat{F} + \frac{\mu_g}{2}\|\boldsymbol{w}_t^{(g)} - \widetilde{\boldsymbol{w}}^{(g)}\|^2\right)$$
$$= -\gamma(F(\boldsymbol{w}_{t+1}^{(g)}) - \widehat{F}) - \frac{\gamma\mu_g}{2}\|\boldsymbol{w}_t^{(g)} - \widetilde{\boldsymbol{w}}^{(g)}\|^2 + \gamma\left(F(\boldsymbol{w}_{t+1}^{(g)}) - F(\boldsymbol{w}_t^{(g)})\right). \tag{94}$$

where $\hat{F} := F(\widetilde{\boldsymbol{w}}^{(g)})$ and the last term $F(\boldsymbol{w}_{t+1}^{(g)}) - F(\boldsymbol{w}_t^{(g)})$ can be upper bounded using the $L_g$-smoothness of $F$:

$$F(\boldsymbol{w}_{t+1}^{(g)}) \leq F(\boldsymbol{w}_t^{(g)}) - \gamma\langle\nabla F(\boldsymbol{w}_t^{(g)}), \widehat{\nabla} F(\boldsymbol{w}_t^{(g)})\rangle + \frac{L_g\gamma^2}{2}\|\widehat{\nabla} F(\boldsymbol{w}_t^{(g)})\|^2$$
$$\overset{(a)}{=} F(\boldsymbol{w}_t^{(g)}) + \frac{\gamma}{2}\left(\|\nabla F(\boldsymbol{w}_t^{(g)}) - \widehat{\nabla} F(\boldsymbol{w}_t^{(g)})\|^2 - \|\nabla F(\boldsymbol{w}_t^{(g)})\|^2 - \|\widehat{\nabla} F(\boldsymbol{w}_t^{(g)})\|^2\right)$$
$$+ \frac{L_g\gamma^2}{2}\|\widehat{\nabla} F(\boldsymbol{w}_t^{(g)})\|^2$$
$$= F(\boldsymbol{w}_t^{(g)}) - \frac{\gamma}{2}\|\nabla F(\boldsymbol{w}_t^{(g)})\|^2 - \left(\frac{\gamma}{2} - \frac{L_g\gamma^2}{2}\right)\|\widehat{\nabla} F(\boldsymbol{w}_t^{(g)})\|^2 + \frac{\gamma}{2}\|\nabla F(\boldsymbol{w}_t^{(g)}) - \widehat{\nabla} F(\boldsymbol{w}_t^{(g)})\|^2, \tag{95}$$

where the step (a) uses the fact $-\langle a, b \rangle = \frac{1}{2}\|a - b\|^2 - \frac{1}{2}\|a\|^2 - \frac{1}{2}\|b\|^2$.

Substituting (94) and (95) into (93) leads to

$$
\begin{aligned}
&\|\boldsymbol{w}_{t+1}^{(g)} - \widetilde{\boldsymbol{w}}^{(g)}\|^2 \\
&\leq (1 + \epsilon_g \gamma)\|\boldsymbol{w}_t^{(g)} - \widetilde{\boldsymbol{w}}^{(g)}\|^2 + \gamma^2\|\widehat{\nabla}F(\boldsymbol{w}_t^{(g)})\|^2 + \left(\frac{\gamma}{\epsilon_g}\right)\|\widehat{\nabla}F(\boldsymbol{w}_t^{(g)}) - \nabla F(\boldsymbol{w}_t^{(g)})\|^2 \\
&\quad - 2\gamma(F(\boldsymbol{w}_{t+1}^{(g)}) - \widehat{F}) - \gamma\mu_g\|\boldsymbol{w}_t^{(g)} - \widetilde{\boldsymbol{w}}^{(g)}\|^2 \\
&\quad - \gamma^2\|\nabla F(\boldsymbol{w}_t^{(g)})\|^2 - \gamma\left(\gamma - L_g\gamma^2\right)\|\widehat{\nabla}F(\boldsymbol{w}_t^{(g)})\|^2 + \gamma^2\|\nabla F(\boldsymbol{w}_t^{(g)}) - \widehat{\nabla}F(\boldsymbol{w}_t^{(g)})\|^2 \\
&\leq (1 - \gamma\mu_g + \gamma\epsilon_g)\|\boldsymbol{w}_t^{(g)} - \widetilde{\boldsymbol{w}}^{(g)}\|^2 - \gamma^2\|\nabla F(\boldsymbol{w}_t^{(g)})\|^2 + (\gamma^3 L_g)\|\widehat{\nabla}F(\boldsymbol{w}_t^{(g)})\|^2 \\
&\quad + \left(\frac{\gamma}{\epsilon_g} + \gamma^2\right)\|\nabla F(\boldsymbol{w}_t^{(g)}) - \widehat{\nabla}F(\boldsymbol{w}_t^{(g)})\|^2.
\end{aligned}
\tag{96}
$$

$\square$

### B.3 PROOF OF THEOREM 2

Recall the definition of $e_t^{(i)}$ in Lemma 6, we define

$$
e_t = \frac{1}{m}\sum_{i \in [m]} e_t^{(i)}.
\tag{97}
$$

Recall the definition $\nabla F(\boldsymbol{w}_t^{(g)}), \widehat{\nabla}F(\boldsymbol{w}_t^{(g)})$ in Lemma 3 and eq (92), the gradient approximation error is

$$
\|\nabla F(\boldsymbol{w}_t^{(g)}) - \widehat{\nabla}F(\boldsymbol{w}_t^{(g)})\|^2 = \lambda^2\left\|\frac{1}{m}\sum_{i=1}^m \left(\boldsymbol{w}_{t,K}^{(i)} - \boldsymbol{w}_{t,\star}^{(i)}\right)\right\|^2 \leq \lambda^2 e_t.
\tag{98}
$$

Therefore, the recursion of optimization error given by Lemma 7 can be further bounded as:

$$
\begin{aligned}
&\|\boldsymbol{w}_{t+1}^{(g)} - \widetilde{\boldsymbol{w}}^{(g)}\|^2 \\
&\stackrel{(a)}{\leq} (1 - \gamma\mu_g + \gamma\epsilon_g)\|\boldsymbol{w}_t^{(g)} - \widetilde{\boldsymbol{w}}^{(g)}\|^2 - \left(\gamma^2 - \gamma^3 L_g(1 + \zeta)\right)\|\nabla F(\boldsymbol{w}_t^{(g)})\|^2 \\
&\quad + \left(\frac{\gamma}{\epsilon_g} + \gamma^2 + L_g\gamma^3(1 + \zeta^{-1})\right)\|\nabla F(\boldsymbol{w}_t^{(g)}) - \widehat{\nabla}F(\boldsymbol{w}_t^{(g)})\|^2 \\
&\stackrel{(b)}{\leq} (1 - \gamma\mu_g + \gamma\epsilon_g)\|\boldsymbol{w}_t^{(g)} - \widetilde{\boldsymbol{w}}^{(g)}\|^2 - \gamma^2\left(1 - \gamma L_g(1 + \zeta)\right)\|\nabla F(\boldsymbol{w}_t^{(g)})\|^2 \\
&\quad + \left(\frac{\gamma}{\epsilon_g} + \gamma^2(2 + \zeta^{-1})\right)\lambda^2 e_t,
\end{aligned}
\tag{99}
$$

where the step (a) comes from the Young's inequality with $\zeta > 0$ to be chosen and the step (b) follows from the step size condition $\gamma \leq 1/L_g$.

Invoking Lemma 2 and Lemma 6, the inner loop optimization error is updated as

$$
\begin{aligned}
e_{t+1} &\leq (1 + \epsilon)(1 - \eta\mu_\ell)^K e_t + \left(1 + \frac{1}{\epsilon}\right)(1 - \eta\mu_\ell)^K \frac{1}{m}\sum_{i=1}^m \|\boldsymbol{w}_{t+1,\star}^{(i)} - \boldsymbol{w}_{t,\star}^{(i)}\|^2 \\
&\stackrel{(a)}{\leq} (1 + \epsilon)(1 - \eta\mu_\ell)^K e_t + \left(1 + \frac{1}{\epsilon}\right)(1 - \eta\mu_\ell)^K L_w^2\|\boldsymbol{w}_t^{(g)} - \boldsymbol{w}_{t+1}^g\|^2 \\
&\stackrel{(b)}{\leq} (1 + \epsilon)(1 - \eta\mu_\ell)^K e_t + \left(1 + \frac{1}{\epsilon}\right)(1 - \eta\mu_\ell)^K L_w^2\gamma^2\|\widehat{\nabla}F(\boldsymbol{w}_t^{(g)})\|^2 \\
&\stackrel{(c)}{\leq} (1 + \epsilon)(1 - \eta\mu_\ell)^K e_t + \left(1 + \frac{1}{\epsilon}\right)(1 - \eta\mu_\ell)^K L_w^2\gamma^2\left(2\|\nabla F(\boldsymbol{w}_t^{(g)})\|^2 + 2\lambda^2 e_t\right) \\
&= \underbrace{\left(1 + \epsilon + 2(1 + \epsilon^{-1})L_w^2\gamma^2\lambda^2\right)(1 - \eta\mu_\ell)^K}_{:=q} e_t + 2\left(1 + \frac{1}{\epsilon}\right)(1 - \eta\mu_\ell)^K L_w^2\gamma^2\|\nabla F(\boldsymbol{w}_t^{(g)})\|^2,
\end{aligned}
\tag{100}
$$

where step (a) is due to Lemma 2, step (b) uses the global update: $\boldsymbol{w}_{t+1}^{(g)} - \boldsymbol{w}_t^{(g)} = -\gamma \widehat{\nabla} F(\boldsymbol{w}_t^{(g)})$ and in step (c), we just apply Cauchy inequality instead of Young's inequality.

Under $\eta \leq 1/L_\ell$, we have $1 - \eta\mu_\ell < 1$. Therefore, by choosing a large $K$ we can always drive $q$ arbitrarily small. Let $K$ be such that

$$q \leq 1 - \gamma\mu_g \leq 1 - \gamma\mu_g + \gamma\epsilon_g. \tag{101}$$

Then, we can rewrite (100) as:

$$
\begin{aligned}
e_{t+1} \leq &\frac{1 - \gamma\mu_g + \gamma\epsilon_g + q}{2}e_t - \frac{1 - \gamma\mu_g + \gamma\epsilon_g - q}{2}e_t \\
&+ 2\left(1 + \frac{1}{\epsilon}\right)(1 - \eta\mu_\ell)^K L_w^2 \gamma^2 \|\nabla F(\boldsymbol{w}_t^{(g)})\|^2.
\end{aligned} \tag{102}
$$

Letting $c_1 = \frac{2}{1 - \gamma\mu_g + \gamma\epsilon_g - q} \cdot \left(\frac{\gamma}{\epsilon_g} + \gamma^2(2 + \zeta^{-1})\right)\lambda^2$, we can combine (102) with (99) and obtain

$$
\begin{aligned}
&\|\boldsymbol{w}_{t+1}^{(g)} - \widetilde{\boldsymbol{w}}^{(g)}\|^2 + c_1 \cdot e_{t+1} \\
\leq &(1 - \gamma\mu_g + \gamma\epsilon_g)\left(\|\boldsymbol{w}_t^{(g)} - \widetilde{\boldsymbol{w}}^{(g)}\|^2 + c_1 \cdot e_t\right) - \gamma^2\left(1 - \gamma L_g(1 + \zeta)\right)\|\nabla F(\boldsymbol{w}_t^{(g)})\|^2 \\
&+ \left(\frac{2}{1 - \gamma\mu_g + \gamma\epsilon_g - q} \cdot \left(\frac{\gamma}{\epsilon_g} + \gamma^2(2 + \zeta^{-1})\right)\lambda^2\right) \cdot 2\left(1 + \frac{1}{\epsilon}\right)(1 - \eta\mu_\ell)^K L_w^2 \gamma^2 \|\nabla F(\boldsymbol{w}_t^{(g)})\|^2.
\end{aligned} \tag{103}
$$

Therefore, if the condition (101) and

$$\frac{4\lambda^2\gamma}{1 - \gamma\mu_g + \gamma\epsilon_g - q}\left(\frac{1}{\epsilon_g} + \gamma(2 + \zeta^{-1})\right)\left(1 + \frac{1}{\epsilon}\right)(1 - \eta\mu_\ell)^K L_w^2 \leq 1 - \gamma L_g(1 + \zeta) \tag{104}$$

hold then (103) implies both $\|\boldsymbol{w}_t^{(g)} - \widetilde{\boldsymbol{w}}^{(g)}\|^2$ and $e_t$ converge to zero at rate $1 - \gamma\mu_g + \gamma\epsilon_g$.

It remains to specify the free parameters $(\epsilon_g, \epsilon, \zeta)$ in (101), (104) and others listed in Lemma 7 and Lemma 6.

• Convergence Rate For the connection term in (103), if we set

$$\epsilon_g = \frac{\mu_g}{2} \cdot (1 - \gamma\mu_g) > 0, \tag{105}$$

which, substituting into $1 - \gamma\mu_g + \gamma\epsilon_g$, gives the convergence rate

$$r := 1 - \frac{\gamma\mu_g}{2} - \frac{(\gamma\mu_g)^2}{2}. \tag{106}$$

Under condition $\gamma \leq 1/L_g$, one can verify that $r \in (0, 1)$.

• Step size conditions. Under the requirement (101), we have

$$1 - \gamma\mu_g + \gamma\epsilon_g - q \geq \gamma\epsilon_g, \tag{107}$$

which gives the following sufficient condition for (104):

$$\frac{2}{\epsilon_g} \cdot \left(\frac{1}{\epsilon_g} + \gamma(2 + \zeta^{-1})\right)\lambda^2 \cdot 2\left(1 + \frac{1}{\epsilon}\right)(1 - \eta\mu_\ell)^K L_w^2 \leq 1 - \gamma L_g(1 + \zeta). \tag{108}$$

Further restricting $\gamma$ such that

$$\gamma(2 + \zeta^{-1})\frac{\mu_g}{2} \leq 1 \quad \Longrightarrow \quad \gamma(2 + \zeta^{-1}) \leq \frac{1}{\epsilon_g}, \tag{109}$$

it suffices to require the following for (108) to hold:

$$\left(\frac{2}{\epsilon_g}\right)^2 \lambda^2 \cdot 2\left(1 + \frac{1}{\epsilon}\right)(1 - \eta\mu_\ell)^K L_w^2 \leq 1 - \gamma L_g(1 + \zeta). \tag{110}$$

Letting $\epsilon = 1$ and $\zeta = 1 - \gamma L_g > 0$, we collect all the conditions (105), (110) and (101) on $\gamma$ respectively as follows:

$$1 - \gamma L_g > 0, \quad 1 - \gamma \mu_g > 0, \tag{111}$$

$$(1 - \gamma \mu_g)^2 (1 - \gamma L_g)^2 \geq 4\lambda^2 \cdot 4(1 - \eta \mu_\ell)^K L_w^2 \cdot \left(\frac{2}{\mu_g}\right)^2, \tag{112}$$

$$1 - \gamma \mu_g \geq \left(2 + 4L_w^2 \gamma^2 \lambda^2\right)(1 - \eta \mu_\ell)^K. \tag{113}$$

Using the fact that $\mu_g \leq L_g$, the above conditions simply to

$$\gamma < 1/L_g, \quad \left(2 + 64L_w^2(1/\mu_g)^2\lambda^2\right)(1 - \eta \mu_\ell)^K \leq (1 - \gamma L_g)^4. \tag{114}$$

### B.4 PROOF OF COROLLARY 1

With $\eta = 1/L_\ell = (\lambda + L)^{-1}$, $\gamma = 1/(2L_g) = \frac{\lambda + L}{2\lambda L}$, and using the fact that $L_w = \frac{\lambda}{\lambda + \mu}$ and $\mu_g = \frac{\lambda \mu}{\lambda + \mu}$ (cf. Lemma 2), condition (114) for $K$ becomes

$$\left(2 + 64L_w^2(1/\mu_g)^2\lambda^2\right)(1 - \eta \mu_\ell)^K$$
$$= \left(2 + 64\frac{\lambda^2}{(\lambda + \mu)^2}\frac{(\lambda + \mu)^2}{(\lambda \mu)^2}\lambda^2\right)\left(1 - \frac{\lambda + \mu}{\lambda + L}\right)^K \tag{115}$$
$$= \left(2 + 64\frac{\lambda^2}{\mu^2}\right)\left(\frac{L - \mu}{\lambda + L}\right)^K \leq \frac{1}{16}.$$

If $L = \mu$, then the above condition trivially holds. Otherwise, a sufficient condition for it is

$$66\kappa^2 \left(\frac{L - \mu}{\lambda + L}\right)^{K-2} \leq \frac{1}{16}, \tag{116}$$

where we have used the fact that

$$\frac{\lambda^2}{\mu^2}\left(\frac{L - \mu}{\lambda + L}\right)^K = \left(\frac{\lambda}{\lambda + L}\right)^2 \cdot \left(\frac{L - \mu}{\mu}\right)^2 \left(\frac{L - \mu}{\lambda + L}\right)^{K-2} \leq \kappa^2 \left(\frac{L - \mu}{\lambda + L}\right)^{K-2} \tag{117}$$

and $\kappa := L/\mu \geq 1$. Finally, using the inequality $\log(1/x) \geq 1 - x$ for $0 < x \leq 1$ we obtain

$$K \geq 2 + \frac{\lambda + L}{\lambda + \mu} \cdot \log(1056\kappa^2). \tag{118}$$

### B.5 PROOF OF THE LOCAL MODEL CONVERGENCE RATE IN THEOREM 2

*Proof.* Note that

$$\|\boldsymbol{w}_{t,K}^{(i)} - \widetilde{\boldsymbol{w}}^{(i)}\|^2 \leq 2\|\boldsymbol{w}_{t,K}^{(i)} - \boldsymbol{w}_{t,\star}^{(i)}\|^2 + 2\|\boldsymbol{w}_{t,\star}^{(i)} - \widetilde{\boldsymbol{w}}^{(i)}\|^2. \tag{119}$$

For the first part on the right-hand side, recalling the definition of $e_t^{(i)}$ in Lemma 6

$$2\|\boldsymbol{w}_{t,K}^{(i)} - \boldsymbol{w}_{t,\star}^{(i)}\|^2 = 2\|e_t^{(i)}\|^2. \tag{120}$$

For the second term on the right-hand side, first, note the property in (9)

$$\boldsymbol{w}_{t,\star}^{(i)} = \text{prox}_{L_i/\lambda}(\boldsymbol{w}_t^{(g)}). \tag{121}$$

Based on the first order condition of (1)

$$\widetilde{\boldsymbol{w}}^{(i)} = \text{prox}_{L_i/\lambda}(\widetilde{\boldsymbol{w}}^{(g)}). \tag{122}$$

Next, plugging (121) and (122) into (119), it yields

$$\|\boldsymbol{w}_{t,K}^{(i)} - \widetilde{\boldsymbol{w}}^{(i)}\|^2 \leq 2\|e_t^{(i)}\|^2 + 2\|\operatorname{prox}_{L_i/\lambda}(\boldsymbol{w}^{(g)}) - \operatorname{prox}_{L_i/\lambda}(\widetilde{\boldsymbol{w}}^{(g)})\|^2$$
$$\leq 2\|e_t^{(i)}\|^2 + 2L_w^2\|\boldsymbol{w}_{t,\star}^{(i)} - \widetilde{\boldsymbol{w}}^{(i)}\|^2, \tag{123}$$

where the last we use Lemma 2.

In the proof of Theorem 2, we have established that both $\|\boldsymbol{w}_t^{(g)} - \widetilde{\boldsymbol{w}}^{(g)}\|^2$ and $e_t$ converges to zero linearly at rate $1 - \gamma\mu_g/2 - (\gamma\mu_g)^2/2$, combining with (123), the proof is finished. $\qquad\square$

## C  ADDITIONAL EXPERIMENT DETAILS AND RESULTS

### C.1  ADDITIONAL EXPERIMENT DETAILS

In this section, we provide more details on the implementation of the methods used in the empirical analysis.

**Real Dataset.** The MNIST dataset consists of 70,000 grayscale images of handwritten digits, each 28x28 pixels, classified into 10 classes (digits 0-9). The EMNIST dataset extends MNIST, including handwritten letters and digits. The Balanced split contains 131,600 28x28 grayscale images across 47 balanced classes (digits and uppercase/lowercase letters). And the CIFAR-10 dataset contains 60,000 color images (32x32 pixels, 3 RGB channels), categorized into 10 classes representing real-world objects (e.g., airplane, car, and dog).

**Algorithms**. To fully investigate the effect of personalization in FedProx with the Algorithm B.1, we compare the two extreme algorithms (GlobalTraining and LocalTraining). Furthermore, since there exist other methods solving the objective (1) to do personalized federated learning, we also try to investigate the effect of personalization in other methods, like pFedMe (T Dinh et al., 2020). The algorithm pFedMe has a similar design with a double loop to solve the subproblem in each communication round.

**Selection of $\lambda$.** For the logistic regression on the synthetic dataset, we selected three values for $\lambda$: small $\lambda = 0.02$, medium $\lambda = 0.1$, and large $\lambda = 0.5$. For logistic regression on the MNIST dataset, we used small $\lambda = 0.5$, medium $\lambda = 1.5$, and large $\lambda = 2.5$. For CNN on the MNIST dataset, we used small $\lambda = 0.2$, medium $\lambda = 0.5$, and large $\lambda = 2.5$. Such a choice is to make sure that each line in the figure is clearly separable. Additionally, we implemented an adaptive choice of $\lambda$ following the formula outlined in Theorem 1, with $\rho$ set to 2. The value of $\rho$ was determined through grid search over the range $[1, 10]$ to ensure optimal performance. For other real datasets, the regularization term $\lambda$ is set as small $\lambda = 0.1$, medium $\lambda = 0.5$, and large $\lambda = 1$.

**Selection of Step Size.** For the synthetic dataset, aligned with Theorem 2, we set the global step size $\gamma = \frac{\lambda+L}{\lambda L}$ and the local step size $\eta = \frac{1}{L+\lambda}$. The smoothness constant $L$ is computed as the upper bound of the $L$-smoothness constant for logistic regression, specifically $4^{-1}\Lambda_{\max}$, where $\Lambda_{\max}$ denotes the largest eigenvalue of the Gram matrix $\boldsymbol{X}^\top\boldsymbol{X}$. The local step size for the synthetic dataset was further tuned through grid search in $\{0.001, 0.01, 0.1\}$. For the real dataset, the global learning rate was set to $\frac{1}{\lambda}$, while the local learning rate was set to 0.01, chosen via grid search. For both GlobalTrain and LocalTrain training baselines, the learning rate was also chosen via grid search within the range $\{0.001, 0.01, 0.1\}$. LocalTrain is implemented using a local (S)GD, and GlobalTrain is implemented using FedAvg (McMahan et al., 2017).

**Selection of Hyperparameters and Convolutional Neural Network Model.** For the MNIST dataset, we implemented logistic regression using the SGD solver with 5 epochs, a batch size of 32, and 20 total runs, and the same hyperparameter setup was used, except the batch size was reduced to 16 to ensure stable convergence in CNN with two convolution layers. To make our experiment more comprehensive, we also utilized a five-layer CNN to train the MNIST dataset with 10 epochs, a batch size of 16, and 50 total runs. For the EMNIST dataset, we used the same five-layer CNN with the same settings. And for CIFAR10, we implemented a three-layer CNN with 10 epochs, a batch size of 16, and 50 total runs. To study the data heterogeneity, we set the number of classes for each client in the MNIST, EMNIST, and CIFAR10 datasets as $\{2, 6, 10\}$, $\{30, 40, 47\}$, and $\{2, 6, 10\}$, respectively.

**Optimizer.** For the synthetic dataset, following the analysis in the paper, the loss function is optimized using gradient descent (GD). For the MNIST dataset, all models are implemented in PyTorch, and optimization is performed using the stochastic gradient descent (SGD) solver.

**Evaluation Criterion.** In terms of evaluation, for the synthetic dataset, we track the ground-truth global model $\boldsymbol{w}_\star^{(g)}$ and local model $\boldsymbol{w}_\star^{(i)}$. We compute the global statistical error as $\|\widetilde{\boldsymbol{w}}^{(g)} - \boldsymbol{w}_\star^{(g)}\|^2$ and the local statistical error as $\|\widetilde{\boldsymbol{w}}^{(i)} - \boldsymbol{w}_\star^{(i)}\|^2$. To capture the total error at the $t$-th iteration, we report both the global total error $\|\boldsymbol{w}_t^{(g)} - \boldsymbol{w}_\star^{(g)}\|^2$ and the local total error $\|\boldsymbol{w}_{t,0}^{(i)} - \boldsymbol{w}_\star^{(i)}\|^2$, which combine both optimization and statistical errors. For the MNIST dataset, we report test accuracy on the MNIST testing set as a measure of the statistical error, and training loss for each client's training set as total error at the $t$-th iteration.

## C.2 Additional Experiment Results

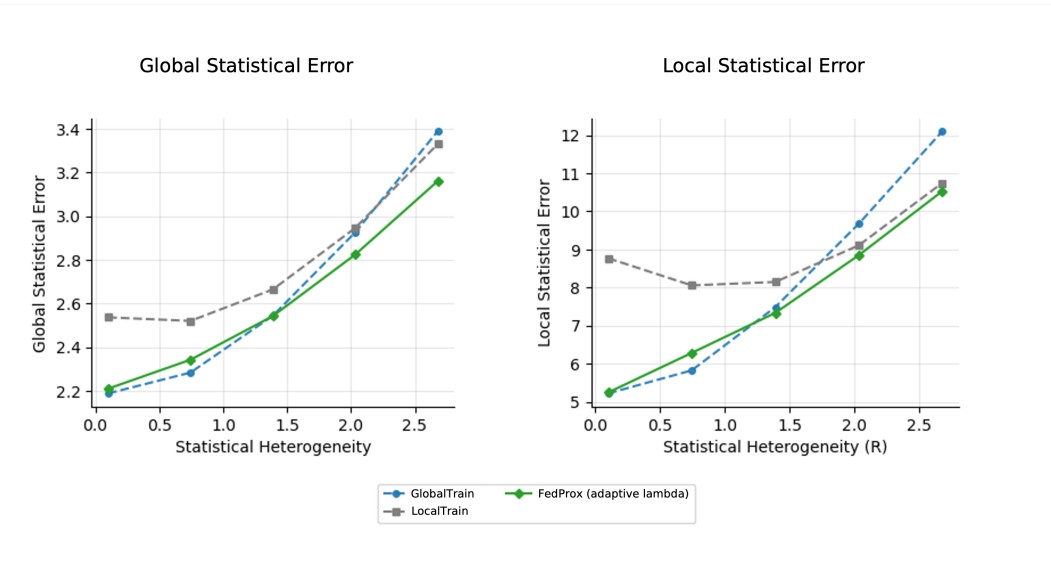

Figure 3: Performance of the FedProx algorithm with an adaptive choice of $\lambda$ (from Theorem 1) under varying levels of statistical heterogeneity.

In Figure C.2, Statistical error of the FedProx algorithm with an adaptive choice of $\lambda$ (from Theorem 1) under varying levels of statistical heterogeneity. With the adaptive choice of $\lambda$ as defined in Theorem 1, FedProx consistently achieves the lowest statistical error across most scenarios, outperforming both GlobalTrain and LocalTrain baselines. This is particularly evident under moderate and high levels of statistical heterogeneity. However, as $R$ approaches very small values, the global baseline performs slightly better, which aligns with the slower rate of FedProx with small $R$ described in our theoretical results. Overall, these results validate our choice of adaptive $\lambda$ and confirm the statistical rate provided by our analysis.

In Figure 4(a), we compare FedProx, LocalTrain, and GlobalTrain on the MNIST dataset by plotting testing accuracy against varying levels of statistical heterogeneity. As heterogeneity increases, each client has access to fewer classes, resulting in fewer classification tasks. This change simplifies the problem, which explains why all methods show improved evaluation accuracy as the number of classes decreases. However, this does not diminish the insights from the relative comparison between methods. GlobalTrain and FedProx with larger $\lambda$ perform better at lower heterogeneity levels, while LocalTrain and FedProx with smaller $\lambda$ perform better as heterogeneity increases.

In Figure 4(b), we track testing loss as a function of communication rounds. Across all sub-figures, FedProx with smaller $\lambda$ (higher personalization) demonstrates faster convergence, especially in the early stages of training. The final training loss achieved by different methods also aligns with the results shown in Figure 4(a) and Figure 1.

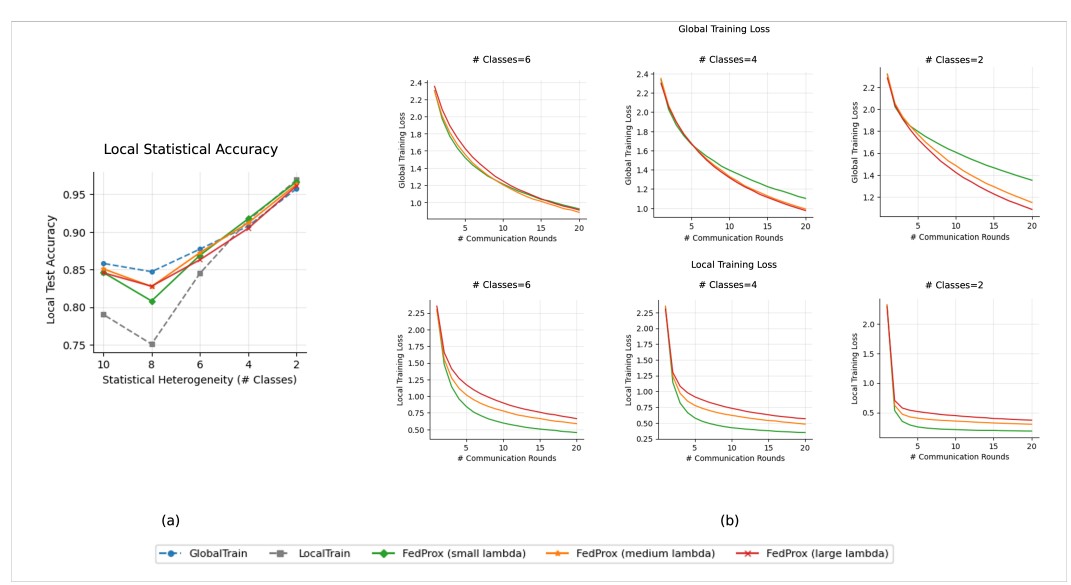

Figure 4: (a) Testing accuracy of different methods (FedProx, LocalTrain, GlobalTrain) versus statistical heterogeneity levels on the MNIST dataset (logistic regression). (b) Training loss versus communication rounds for the same methods.

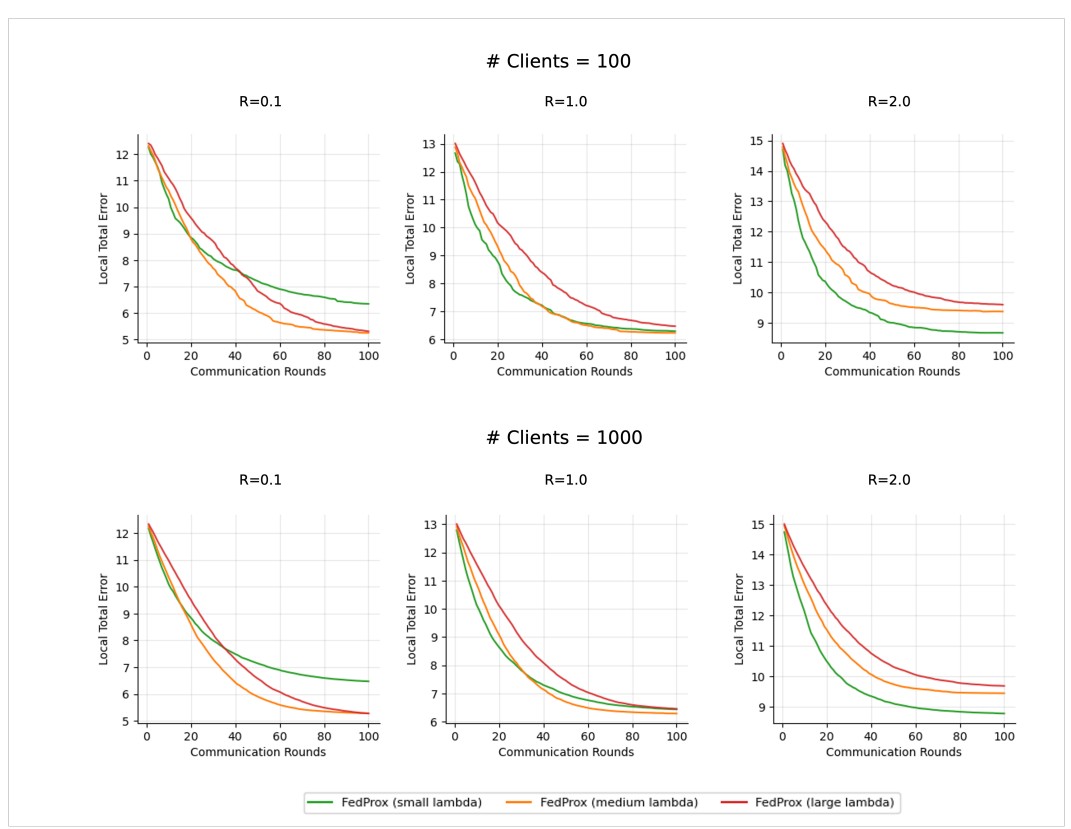

Figure 5: Performance of FedProx with 100 clients (top row) and 1,000 clients (bottom row) with 10% client sampling under different levels of statistical heterogeneity ($R$).

In Figure 5, the top row shows results for 100 clients, and the bottom row shows results for 1,000 clients, and in both results we adopt a client sampling of 10% in each communication round. This tests the scalability and generalizability of the our results under typical federated learning setups. Consistent with the findings from Figure 1, FedProx with smaller $\lambda$ (greater personalization) consistently achieves faster convergence at the early stages of training, while FedProx with larger $\lambda$ (less personalization) takes more communication rounds to converge. As statistical heterogeneity increases (measured by $R$), FedProx with smaller $\lambda$ gradually outperforms models with larger $\lambda$, showcasing the adaptability of FedProx to different levels of heterogeneity. These results further validate the theoretical insights from Theorem 1 and Theorem 2, highlighting the influence of personalization on statistical and optimization performance in federated setups with larger numbers of clients.

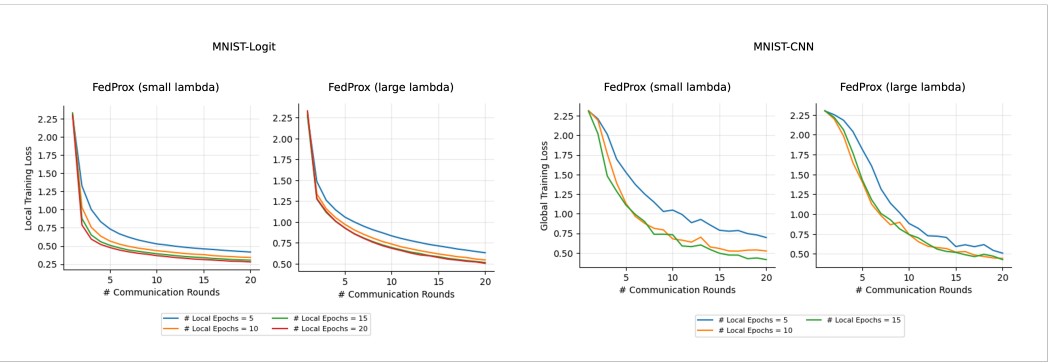

Figure 6: Effect of local updates on FedProx performance for local models in MNIST logistic regression and global models ni MNIST CNN, complementing the results shown in Figure 2.

Figure 6 demonstrates the impact of varying the number of local updates on FedProx performance, complementing the results shown in Figure 2. For FedProx with smaller $\lambda$ (higher personalization), increasing the number of local updates notably reduces the need for communication, leading to faster convergence. In contrast, for FedProx with larger $\lambda$ (lower personalization), increasing local updates yields minimal benefit and only adds to computation costs without reducing communication demands. The results indicate that for larger $\lambda$, the optimal strategy is to prioritize frequent communication rather than increasing local computation. This highlights the trade-off between local computation and communication efficiency under different levels of personalization, validating our theoretical findings in Corollary 1.

| Dataset | Client Class | One-stage FedProx | | | GlobalTrain | LocalTrain | pFedMe | | |
|---|---|---|---|---|---|---|---|---|---|
| | | (small) | (medium) | (large) | | | (small) | (medium) | (large) |
| MNIST CNN2 | 2 | 0.3567 | 0.4721 | 0.6019 | 0.6123 | 0.3019 | 0.3973 | 0.5828 | 0.6172 |
| | 6 | 0.7001 | 0.7833 | 0.7843 | 0.8092 | 0.6231 | 0.7918 | 0.8093 | 0.8312 |
| | 10 | 0.8753 | 0.8893 | 0.9032 | 0.9098 | 0.8194 | 0.8756 | 0.8771 | 0.8749 |
| MNIST CNN5 | 2 | 0.8891 | 0.9000 | 0.9333 | 0.9377 | 0.6536 | 0.9000 | 0.9003 | 0.9380 |
| | 6 | 0.8451 | 0.8941 | 0.9392 | 0.9446 | 0.7333 | 0.9288 | 0.9186 | 0.9306 |
| | 10 | 0.9091 | 0.9102 | 0.9421 | 0.9433 | 0.7633 | 0.9340 | 0.9385 | 0.9440 |
| EMNIST CNN5 | 30 | 0.6084 | 0.6089 | 0.6011 | 0.6102 | 0.5583 | 0.6133 | 0.6129 | 0.6122 |
| | 40 | 0.6097 | 0.6123 | 0.6357 | 0.6423 | 0.5644 | 0.6012 | 0.6102 | 0.6134 |
| | 47 | 0.6278 | 0.6415 | 0.6672 | 0.6611 | 0.6111 | 0.6345 | 0.6532 | 0.6717 |
| CIFAR10 CNN3 | 2 | 0.6101 | 0.6292 | 0.6340 | 0.6444 | 0.5801 | 0.6056 | 0.6012 | 0.6033 |
| | 6 | 0.6211 | 0.6311 | 0.6712 | 0.6949 | 0.5623 | 0.6163 | 0.6163 | 0.6439 |
| | 10 | 0.6102 | 0.6712 | 0.7302 | 0.8041 | 0.6238 | 0.6744 | 0.7443 | 0.7732 |

Table 3: Evaluation accuracy of algorithms (FedProx, GlobalTrain, LocalTrain, and pFedMe) across datasets and client classes. (CNN$k$ means the CNN has $k$ convolutional layers and small, medium, and large indicate different values of $\lambda$, and client class indicates the number of class on each client)

To further validate our conclusions about the effect of personalization on the statistical accuracy of FedProx (Objective 1), we conduct additional experiments on real datasets, including EMNIST and

CIFAR-10. For each dataset, we explore three levels of statistical heterogeneity, controlled by varying the number of classes each client has access to, which follows a widely used setup in federated learning literature. The models are trained using different convolutional neural network (CNN) architectures, denoted as CNN2 and CNN5, where the numbers indicate the number of convolutional layers in the network. As the true underlying model is not accessible for these real datasets, we evaluate model performance based on classification accuracy on the test set, as shown in Tables 3 and 4. We also conduct experiments another regularization-based personalization method, pFedMe (T Dinh et al., 2020), to demonstrate that our results could be generalized to other regularization-based personalized federated learning methods as well. For the columns corresponding to One-stage FedProx and pFedMe, we conduct experiments with varying regularization parameter $\lambda$, investigating the role of personalization in the algorithms. Specifically, we experiment with small, medium, and large $\lambda$ values (see Appendix C.1 for detailed parameter settings).

In Table 3, we can see that consistently across datasets, as $\lambda$ decreases (indicating a higher degree of personalization), the accuracy approaches that of LocalTrain, which trains separate models for each client. Conversely, as $\lambda$ increases (indicating lower personalization and more collaboration), the accuracy approaches that of GlobalTrain, which represents purely global training. This interpolation behavior between LocalTrain and GlobalTrain, enabled by FedProx's personalized objective, aligns well with our theoretical results in Theorem 1 and highlights the adaptive capability of the FedProx method.

Moreover, as dataset homogeneity increases (i.e., clients share more similar data distributions), we observe a general improvement in accuracy across all methods, regardless of the level of personalization. This result demonstrates that while personalization offers significant benefits in highly heterogeneous settings, its necessity diminishes as heterogeneity decreases. Additionally, experiments with different CNN architectures confirm the robustness of these findings across model designs. Although we did not provide a theoretical analysis for pFedMe in this paper, our results suggest that the empirical behavior of pFedMe is consistent with the theoretical insights derived for FedProx. In particular, we observe that pFedMe also exhibits interpolation between local and global learning under varying levels of data heterogeneity.

| Dataset | Client Class | One-stage FedProx (small $\lambda$) | One-stage FedProx (large $\lambda$) | Dichotomous Strategy |
|---|---|---|---|---|
| MNIST CNN2 | 2 | 0.4721 | 0.6019 | 0.2567 |
| MNIST CNN2 | 6 | 0.7833 | 0.7843 | 0.7001 |
| MNIST CNN2 | 10 | 0.8893 | 0.9032 | 0.8753 |
| MNIST CNN5 | 2 | 0.8891 | 0.9333 | 0.7032 |
| MNIST CNN5 | 6 | 0.8451 | 0.8941 | 0.7941 |
| MNIST CNN5 | 10 | 0.9091 | 0.9102 | 0.8922 |
| EMNIST CNN5 | 30 | 0.6084 | 0.6089 | 0.5291 |
| EMNIST CNN5 | 40 | 0.6097 | 0.6123 | 0.5732 |
| EMNIST CNN5 | 47 | 0.6278 | 0.6415 | 0.5856 |

Table 4: Comparison of evaluation accuracy for One-stage FedProx with different values of $\lambda$ and Dichotomous Strategy across various datasets and client classes.

## C.3 COMPARISON OF FEDPROX WITH A DICHOTOMOUS STRATEGY

A simple baseline in personalized federated learning is the *dichotomous strategy*, where the dataset on each client is split into two subsets: one is used for global training (GlobalTrain) with pooled client data, while the other is used for purely local training (LocalTrain). After training, the two models are compared on the test set, and the best-performing model is selected for each client. Such a strategy is known to achieve minimax-optimal statistical accuracy under certain conditions; see Chen et al. (2023c) for a detailed discussion.

Even though the convergence rate coincides with such a dichotomous strategy, we may reiterate that the key contribution of our work lies in the fine-grained analysis of the FedProx method—a method that has demonstrated strong empirical performance in federated learning [2]. Through our analysis, we have, for the first time, established the minimax optimality of FedProx, showing that its statistical performance matches the lower bound of the estimation problem. If one seeks to understand why FedProx could be advantageous over the dichotomous strategy in practice, here are some of the reasons:

- **Stability in Heterogeneous Environments:** Empirical studies [1] demonstrate that Fed-Prox achieves more stable convergence compared to methods such as FedAvg, especially in settings with significant in a heterogeneous network.

- **Adaptivity and Flexibility:** FedProx seamlessly interpolates between the extremes of global training and local training by tuning a single regularization parameter, $\lambda$. This adaptability allows practitioners to dynamically balance the trade-off between bias and variance based on the level of statistical heterogeneity, without the need for fixed, hard decisions inherent in the dichotomous strategy.

To further compare FedProx with the dichotomous strategy, we conduct a simple experiment. In the dichotomous setup, each client's dataset is partitioned into two subsets: one subset is used for local training (LocalTrain), where each client trains independently, and the other subset is pooled across all clients for global training (GlobalTrain). The accuracy of the two resulting models is evaluated on a test set, and the better-performing model is reported for each client. By contrast, FedProx is implemented using the experimental setup described earlier, leveraging the full dataset. As shown in Table 4, FedProx consistently outperforms the dichotomous strategy across various configurations. This performance gap can be attributed to the inefficiency of the dichotomous approach, which sacrifices valuable data during the sample splitting process, resulting in reduced sample efficiency.

Even in a hypothetical scenario where the full sample is used for both LocalTrain and GlobalTrain in the dichotomous strategy, the results in Table 3 demonstrate that FedProx achieves a continuous interpolation between the two extremes of LocalTrain and GlobalTrain by adjusting its regularization parameter $\lambda$. This aligns with our theoretical findings.

