# OpenReview forum: "The Effect of Personalization in FedProx: A Fine-grained Analysis on Statistical Accuracy and Communication Efficiency"
_ICLR.cc/2025/Conference — Submitted to ICLR 2025_

### Official Review · Reviewer_han2 · 2024-10-26

**Soundness:** 3
**Presentation:** 3
**Contribution:** 2
**Rating:** 6
**Confidence:** 5

**Summary:**

This paper has studied how personalization in FedProx impacts the statistical error of FedProx and  the trade-off between local computation cost and communication cost. First,  this paper provided sharper statistical error bounds by adaptively tuning the regularization parameter $\lambda$ in FedProx based on the degree of statistical heterogenelity parameter $R$. Then this paper provided a new algorithm to solve FedProx and analyzed the rate of convergence on both local models and global model, showing personalization can increase the computational cost per iteration but also reduce the number of communications. Finally, numerical experiments have been conducted to validate the theoretical findings.

**Strengths:**

This paper studied a fundamental problem that how statistical heterogenelity impacts the process of personalized federated optimization,
gave new statistical error bounds under different levels of statistical heterogenelity,
and established a new algorithm that shows an interesting trade-off between local computation cost and communication cost.

**Weaknesses:**

(1) The authors claimed that their statistical error bounds improve previous results which I do not fully agree with. The main weaknesses of this paper include the unfair comparisons over previous results and the limited improvement in statistical errors over previous results. There are three reasons.

(i) The statistical error bounds in this paper omitted the optimization error of algorithm solving FedProx.
In other words, this paper just analyzed the statistical error of ERM estimators.
However, previous results [1] incorporate the optimization error of algorithm solving FedProx,
that is, the statistical error of approximate ERM estimators.

(ii) The statistical error defined in this paper is slightly different from previous work [1].
While the two definitions are closely related, there exists a gap represented by a problem-dependent constant.

(iii) Except for problem-dpenednt constants,  this paper only improves previous results [1] by a factor of $O(1/\sqrt{m})$
in the case of $R\leq \frac{1}{m\sqrt{n}}$.

(2) This paper aims to provide contributions in theory. In addition to the theoretical results,
it is necessary to explain the technical challenges on the theoretical analyses and the corresponding technical controbutions, both of which are currently unclear to me.
Besides, there are many typos in the current version.

(i) In Eq. (29), (30), (32) and so on, the expectation operator $\mathbb{E}[\cdot]$ are omitted.

(ii) In Eq. (33), (35), (36), I think $\tilde{w}^{(i)}$ should be corrected as $w^{(i)}_*$.

References

[1] Chen et al. Minimax estimation for personalized federated learning: An alternative between fedavg and local training? JMLR, 2023.

**Questions:**

Could the authors elucidate the technical challenges and contributions regarding the theoretical analyses?

**Details Of Ethics Concerns:**

I do not find any ethics concerns.

---

> ### Author Response · Authors · 2024-11-22
> **Response to Reviewer han2 (Part 1)**
>
> We thank the reviewer for their insightful comments and detailed feedback. Below, we address each of the specific concerns:
>
> **Response to Statistical Error Bounds and Comparisons:**
>
> 1. **Inclusion of optimization error:** We would like to point out that the comparison with [1] is fair: the first part of this paper derives the *statistical* convergence rate of the estimator. In Table 1, we provided a comparison with [1] as given by Theorem 11 and 12 therein (cf. Eqs. (28) and (29)). The analysis in [1] relies on the algorithm update specified by Algorithm 2 to obtain the rate of the estimator, whereas our analysis directly targets the problem formulation and is free from the choice of algorithm employed to solve the problem. In other words, our result ensures that **any algorithm** solves the FedProx problem exactly and would attain the rate given by Theorem 1.
> The second part of the paper provides the *algorithmic* convergence rate. Corollary 1 shows a solution that is $\varepsilon$-close to the FedProx minimizer can be obtained with $\mathcal{O}(\kappa \frac{\lambda+\mu}{\lambda+L} \log\frac{1}{\varepsilon})$ communication rounds and $\mathcal{\widetilde{O}}(\kappa \log\frac{1}{\varepsilon} )$ gradient evaluations. Note that our step size is set to a constant, and the rate is linear. If we combine the two parts using the simple relation
> $$
> \\|\boldsymbol{w}\^{(g)}\_t - \boldsymbol{w}\^{(g)}\_\*\\|\_2 \leq \\|\boldsymbol{w}\^{(g)}\_{t}- \boldsymbol{\tilde{w}}\^{(g)}\\|\_2 + \\| \boldsymbol{\tilde{w}}\^{(g)} - \boldsymbol{w}\^{(g)}\_\*\\|\_2, \quad \forall t,
> $$
> we can immediately obtain the conclusion that a solution $\boldsymbol{w}\^{(g)}\_{T}$ of the same statistical rate as $\boldsymbol{\tilde{w}}\^{(g)}$ can be attained if the communication round $R  \gtrsim \kappa \frac{\lambda+\mu}{\lambda+L} \cdot \log\frac{1}{\\| \boldsymbol{\tilde{w}}\^{(g)} - \boldsymbol{w}\^{(g)}\_\*\\|\_2} $ and the total gradient evaluation $T  \gtrsim \kappa  \cdot \log\frac{1}{\\| \boldsymbol{\tilde{w}}^{(g)} - \boldsymbol{w}^{(g)}_*\\|\_2}$. The complexity is better than that provided in [1], where a bounded gradient assumption is imposed, the step size is diminishing, and the computation complexity is sublinear.
>
> 2. **Definition of statistical accuracy:** Indeed, we defined the statistical accuracy using model difference rather than excess risk. However, the two error measures are comparable under strong convexity. Take the error of $\widetilde{\boldsymbol{w}}^{(i)}$ as an example. Our error measure is $\\|\tilde{\boldsymbol{w}}\^{(i)} - \boldsymbol{w}\_{\star}\^{(i)}\\|\^{2}$ whereas the individualized excess risk (IER) defined in [1] is $\mathbb{E}\_{z\_i} \left[ \ell(\tilde{\boldsymbol{w}}\^{(i)}, z\_i) - \ell(\boldsymbol{w}\_\star\^{(i)}, z\_i) \right].$ By strong convexity and the optimality of $\boldsymbol{w}\_\star\^{(i)}$ we have the error bound $\mathbb{E}\_{z\_i} \left[ \ell({\boldsymbol{w}}\^{(i)}, z\_i) - \ell(\boldsymbol{w}\_\star\^{(i)}, z\_i) \right] \geq  \frac{\mu}{2}  \\| {\boldsymbol{w}}\^{(i)} - \boldsymbol{w}\_{\star}\^{(i)} \\|^2$ hold for any $\boldsymbol{w}^{(i)}$. Furthermore, assuming smoothness we have
>     $\mathbb{E}\_{z\_i} \left[ \ell({\boldsymbol{w}}\^{(i)}, z\_i) - \ell(\boldsymbol{w}\_\star\^{(i)}, z\_i) \right] \leq \frac{L}{2} \\| {\boldsymbol{w}}\^{(i)} - \boldsymbol{w}\_{\star}\^{(i)}\\|^2$. Under our common problem assumptions, the two measures are equivalent if $\mu$ and $L$ are treated as constants. In Table 1 of the manuscript, we did a fair comparison by transforming their result into our metrics $\mathbb{E}\\|\boldsymbol{w}_{\star}^{(i)}  - \widetilde{\boldsymbol{w}}^{(i)}\\|^2$.
>
>
>  3. **Improved rate:** As noted by the reviewer, our original result already improves over that in (1) by $O(1/\sqrt{m})$ in the case $R \leq \frac{1}{m\sqrt{n}}$. The improvement is not minor since in federated learning settings where $m$ can be very large. In [4], the cross-device federated learning considers the number of clients as $10^5$. In addition, we proved this rate without assuming the boundedness of the loss function and compactness of the domain (cf. Assumption A (a) and (b) in [1]).
>
>
>
> **Minimax Optimality:**
>     In the revised manuscript, we have improved our bounds to $O\left(\frac{1}{N} + R^2 \wedge \frac{1}{n}\right)$, which matches the lower bound established in [1]. To the best of our knowledge, our work is the first to establish the *minimax optimality* of FedProx.

---

> ### Author Response · Authors · 2024-11-22
> **Response to Reviewer han2 (Part 2)**
>
> **Response to Technical Challenges and Contribution**
>
> Prior work [1] established the rate of FedProx through algorithmic stability, and to obtain bounded stability, the boundedness of the loss function is needed. This assumption (or bounded gradients) is, in fact, commonly imposed for analysis based on the tool of algorithmic stability, e.g., [5-7]. To relax this assumption and obtain a bound independent of the diameter of the domain/gradient norm/loss function norm, we need to jump out of the stability analysis framework  and develop new techniques.
>
> Our analysis does not rely on any algorithm but directly tackles the objective function, establishing rates using purely the properties of the loss. Specifically, using the strong convexity of the loss function, we first proved the estimation error is bounded by the gradient variance and an extra statistical heterogeneity term controlled by $\lambda$, as detailed in Lemma 1. This allows us to show for large $R$, a small $\lambda$ can be chosen to yield rate $O(1/n)$ and match one of the worst cases in the lower bound. For the complementary case $R \leq 1/\sqrt{n}$, we leveraged the GlobalTrain solution $\widetilde{\boldsymbol{w}}\_{GT}$ as a bridge and proved the solutions of FedProx to $\widetilde{\boldsymbol{w}}\_{GT}$ are bounded by a term inversely proportional to $\lambda$ (cf. Lemma 4 and Eq. (49) in the Appendix A.3), implying that they can be made arbitrarily close to $\widetilde{\boldsymbol{w}}\_{GT}$  by setting $\lambda$ small. This, together with the rate of $\widetilde{\boldsymbol{w}}\_{GT}$ as proved in Lemma 5, yields a rate of $O(1/N + R^2)$. Combining the two cases, we proved minimax optimality. Not only each piece of the above results are new, but more importantly, identifying they are the key ingredients to show FedProx is minimax optimal, are the technical novelties of this proof.
>
>
>
> **Response to Typographical Errors:**
>
> We sincerely thank the reviewer for pointing out typos, all of which have been corrected in the revised manuscript.
>
> We hope the above clarifications address the reviewer’s concerns, and we remain happy to provide further details if needed.
>
> ----------------------------
>
> [1] Chen, S., Zheng, Q., Long, Q., and Su, W. J. (2023). Minimax Estimation for Personalized Federated Learning: An Alternative between FedAvg and Local Training?. Journal of Machine Learning Research, 24(262), 1-59.
>
> [2] Niu C, Wu F, Tang S, et al. Billion-scale federated learning on mobile clients: A submodel design with tunable privacy[C]//Proceedings of the 26th Annual International Conference on Mobile Computing and Networking. 2020: 1-14.
>
> [3] Chen D, Gao D, Xie Y, et al. Fs-real: Towards real-world cross-device federated learning[C]//Proceedings of the 29th ACM SIGKDD Conference on Knowledge Discovery and Data Mining. 2023: 3829-3841.
>
>
> [4] Chen D, Gao D, Xie Y, et al. Fs-real: Towards real-world cross-device federated learning[C]//Proceedings of the 29th ACM SIGKDD Conference on Knowledge Discovery and Data Mining. 2023: 3829-3841.
>
> [5] Fallah, A., Mokhtari, A., and Ozdaglar, A. (2021). Generalization of model-agnostic meta-learning algorithms: Recurring and unseen tasks. Advances in Neural Information Processing Systems, 34, 5469-5480.
>
> [6] Denevi, G., Ciliberto, C., Grazzi, R., and Pontil, M. (2019, May). Learning-to-learn stochastic gradient descent with biased regularization. In International Conference on Machine Learning (pp. 1566-1575). PMLR.
>
> [7] Chen, J., Wu, X. M., Li, Y., Li, Q., Zhan, L. M., and Chung, F. L. (2020). A closer look at the training strategy for modern meta-learning. Advances in neural information processing systems, 33, 396-406.

---

> > ### Comment · Reviewer_han2 · 2024-11-22
> >
> > Thanks for the detailed responses.
> > The rebuttal addresses most of my concerns.
> > I agree that this paper enhances the excess risk bound compared to [1] concerning the dependence on $m$, $n$, and $R$.
> > However, the assumptions that $R$, $L$, and $\mu$ are constants are essentially the same as
> > the assumption of the loss function being bounded or the domain being compact.
> > For example, if $\Vert w\Vert $ or $\Vert z\Vert$ is unbounded, then it is difficult to make $R$, $L$ or $\mu$ constant.
> >
> > I will raise my score to 6 as a new theoretical bound has been established in the revised version.
> >
> > [1] Chen et al. Minimax Estimation for Personalized Federated Learning: An Alternative between FedAvg and Local Training?. JMLR, 2023.

---

> > > ### Author Response · Authors · 2024-11-24
> > > **Response to Reviewer han2**
> > >
> > > We sincerely thank the reviewer for going through our new results and acknowledging the contributions, and we are pleased that our responses have addressed most of the concerns.
> > >
> > >
> > > In the following, we would like to make a clarification on the last point regarding  $R$, $L$, and $\mu$. The new result, as stated in Theorem 1, does not assume these parameters to be constants. They are all explicitly included in the bound provided by Theorem 1. In our previous reply, we stated "the two measures are equivalent if $\mu$ and $L$ are treated as constants" for a quick comparison with [1]. Without such a restriction, we can still convert our result (e.g., $\\|\widetilde{\boldsymbol{w}}\^{(i)} - \boldsymbol{w}\_\star\^{(i)}\\|$) to the risk measure as
> > > $$
> > >     \mathbb{E} \left( \mathbb{E}\_{z_i} \left[ \ell({\widetilde{\boldsymbol{w}}}\^{(i)}, z\_i) - \ell(\boldsymbol{w}\_\star\^{(i)}, z\_i) \right] \right)\leq \frac{L}{2} \left(  \frac{C\_3}{N}+ C\_4\left( \frac{1}{n}\wedge R\^2\right) \right), \quad  C\_3  = \frac{192\rho\^2}{\mu\^2}, C\_4   = \left(\frac{\rho(1 + (4C\_2+5)^{\frac{1}{2}})}{\mu}\right)\^2  \vee 198\kappa^2.
> > > $$
> > > and compare with that provided in [1] (cf. Eq. (29)).
> > >
> > > As for the comparison of the boundness of $R$, $L$, and $\mu$ and the boundness of the loss/compactness of the parameter space, there are nontrivial examples where the latter is stronger. For example, consider a linear model $y = x^\top w^\star + \varepsilon$ with $z = (x,y)$ and $\ell(w,z) = \frac{1}{2}\| y - x^\top w\|^2$. The parameter $L$ is $\lambda_{\max} (xx^\top)$ and is bounded if $x$ is. However, the parameter space $w \in \mathbb{R}^d$ is unbounded, and $\ell$ cannot have a bounded infinity norm for any fixed $x$.
> > > Lastly, we would like to note that in the proof the assumption $\ell (\cdot,z)$ is $L$-smooth for all $z$ can be relaxed to each empirical loss $L_i$ is $L$-smooth, which means we do not  need each $x_i$ to be unbounded. In the least squares example, having i.i.d. $x_i$s with bounded covariance and large enough $n$ suffices. Furthermore, the assumption that $R$ is bounded is not necessary. Our theoretical results, arguments, and proofs remain valid even when $R$ is unbounded.
> > >
> > >
> > > We hope this clarifies the reviewer’s concerns, and we remain happy to provide further details if needed.
> > >
> > >
> > > ----------------------
> > > [1] Chen, S., Zheng, Q., Long, Q., and Su, W. J. (2023). Minimax Estimation for Personalized Federated Learning: An Alternative between FedAvg and Local Training?. Journal of Machine Learning Research, 24(262), 1-59.

---

> ### Comment · Area_Chair_XAFr · 2024-11-25
>
> Dear Reviewer han2,
>
> The author discussion phase will be ending soon. The authors have provided detailed responses. Could you please reply to the authors with whether they have addressed your concern and whether you will keep or modify your assessment on this submission?
>
> Thanks.
>
> Area Chair

---

### Official Review · Reviewer_5muf · 2024-10-31

**Soundness:** 4
**Presentation:** 3
**Contribution:** 3
**Rating:** 8
**Confidence:** 2

**Summary:**

The paper investigates the effect of regularization in FedProx, providing a theoretical guideline for setting the regularization parameter for achieving personalization. Experiments conducted on synthetic and real datasets validate the theoretical analyses.

**Strengths:**

1. It is intuitive that GlobalTrain is more suitable for i.i.d. data, while LocalTrain (and personalized FL) is more appropriate for non-i.i.d. data, and FedProx represents a hybrid approach. This paper substantiates this intuition and formally provides theoretical guidelines to demonstrate how the regularization term in FedProx affects model convergence.
2. In comparison to similar works, the theoretical results in this paper are more precise with fewer assumptions. Furthermore, the experimental results perfectly validate the theoretical analyses.

**Weaknesses:**

1. Figures in the paper could be larger for better readability.

**Questions:**

N/A

---

> ### Author Response · Authors · 2024-11-22
> **Response to Reviewer 5muf**
>
> We sincerely thank you for your  comments, and greatly appreciate your recognition of the theoretical contributions and the validation provided through our experimental results.
>
> We have carefully considered your suggestion regarding figure sizes and have adjusted both of them in the revised version to improve readability and enhance visual clarity.
>
> If there are any specific aspects of the manucript that require further clarification, we are more than happy to provide additional explanations. Please feel free to highlight any areas of concern, and we will do our best to address them and ensure our contributions are presented as clearly and precisely as possible.

---

### Official Review · Reviewer_bMut · 2024-11-04

**Soundness:** 3
**Presentation:** 3
**Contribution:** 3
**Rating:** 6
**Confidence:** 3

**Summary:**

This paper proposes a theoretical guideline for adaptively setting the strength of the regularization under different levels of statistical heterogeneity to achieve personalization in FL. The authors demonstrate that by adaptive tuning the regularization to strengthen the personalization, communication complexity can be reduced without increasing the computation cost overhead.

**Strengths:**

1. This paper addresses a practical issue that commonly exists in the FL framework the statistical heterogeneity is prevalent. The motivation for optimizing regularization strength in Fed Prox is compelling.
2. The paper is well-written and has a logical structure. The mathematical derivations are rigorous and precise. The authors also provide code to facilitate reproducibility.

**Weaknesses:**

1. The experiments in this paper use only one real dataset, MNIST, which may not comprehensively illustrate the effect of personalization on statistical accuracy under various levels of heterogeneity. In addition, the model used in the non-convex case is relatively simple. Including more datasets and exploring more complex models could better demonstrate the main contribution of this paper.
2. The experimental results primarily focus on the model's error. However, it would also be beneficial to include accuracy to show the model performance with different lambdas.
3. The descipriotns and analyses of the experiments lack clarity and it would be better to improve the content. For example, L. 492-493 references Fig.4 in the appendix as displaying error-related results, but Fig. 4 (a) actually shows accuracy variation. Improving these experimental explanations would make the results more interpretable for readers.
4. I am curious whether this adaptive regularization approach can be extended to scenarios where clients are sampled in each communication round.

**Questions:**

Refer to the weakness section.

---

> ### Author Response · Authors · 2024-11-22
> **Response to Reviewer bMut (Part 1)**
>
> We greatly appreciate the reviewers’ feedback. We have provided our responses below:
>
> **Weakness 1 Expanded experiments: additional datasets, complex models, and accuracy evaluation**
>
>
> We have conducted additional experiments using more *diverse datasets*, including CIFAR-10 and EMNIST Balanced, which cover a broader spectrum of heterogeneity levels. Furthermore, we have extended our analysis by employing more complex non-convex models, such as *CNNs with different number of layers*. All of the models, as the reviewer requested, are *evaluated in classification accuracy* instead of model's error. These new results are consistent with our theory: as $\lambda$ increases, FedProx transitions from behaving similarly to LocalTrain towards GlobalTrain. While GlobalTrain performs best in this scenario due to the relatively weak statistical heterogeneity, FedProx, with increasing  $\lambda$, moves closer to GlobalTrain performance. We have included these new results below, and their corresponding analyses can be found in the revised version of the paper (see Appendix C.2).
> | Dataset          | Client Class | One-stage FedProx (small) | One-stage FedProx (medium) | One-stage FedProx (large) | GlobalTrain | LocalTrain | pFedMe (small) | pFedMe (medium) | pFedMe (large) |
> |------------------|--------------|---------------------------|----------------------------|---------------------------|-------------|------------|----------------|-----------------|----------------|
> | MNIST CNN2       | 2            | 0.3567                    | 0.4721                     | 0.6019                    | 0.6123      | 0.3019     | 0.3973         | 0.5828          | 0.6172         |
> |       MNIST CNN2           | 6            | 0.7001                    | 0.7833                     | 0.7843                    | 0.8092      | 0.6231     | 0.7918         | 0.8093          | 0.8312         |
> |     MNIST CNN2             | 10           | 0.8753                    | 0.8893                     | 0.9032                    | 0.9098      | 0.8194     | 0.8756         | 0.8771          | 0.8749         |
> | MNIST CNN5       | 2            | 0.8891                    | 0.9000                     | 0.9333                    | 0.9377      | 0.6536     | 0.9000         | 0.9003          | 0.9380         |
> |     MNIST CNN5             | 6            | 0.8451                    | 0.8941                     | 0.9392                    | 0.9446      | 0.7333     | 0.9288         | 0.9186          | 0.9306         |
> |   MNIST CNN5               | 10           | 0.9091                    | 0.9102                     | 0.9421                    | 0.9433      | 0.7633     | 0.9340         | 0.9385          | 0.9440         |
> | EMNIST CNN5      | 30           | 0.6084                    | 0.6089                     | 0.6011                    | 0.6102      | 0.5583     | 0.6133         | 0.6129          | 0.6122         |
> |      EMNIST CNN5            | 40           | 0.6097                    | 0.6123                     | 0.6357                    | 0.6423      | 0.5644     | 0.6012         | 0.6102          | 0.6134         |
> |       EMNIST CNN5           | 47           | 0.6278                    | 0.6415                     | 0.6672                    | 0.6611      | 0.6111     | 0.6345         | 0.6532          | 0.6717         |
> | CIFAR10 CNN3     | 2            | 0.6101                    | 0.6292                     | 0.6340                    | 0.6444      | 0.5801     | 0.6056         | 0.6012          | 0.6033         |
> |      CIFAR10 CNN3             | 6            | 0.6211                    | 0.6311                     | 0.6712                    | 0.6949      | 0.5623     | 0.6163         | 0.6163          | 0.6439         |
> |          CIFAR10 CNN3         | 10           | 0.6102                    | 0.6712                     | 0.7302                    | 0.8041      | 0.6238     | 0.6744         | 0.7443          | 0.7732         |

---

> ### Author Response · Authors · 2024-11-22
> **Response to Reviewer bMut (Part 2)**
>
> **Weakness 2 Lack of clarity in the description and analysis of the experiments**
>
> We thank the reviewer for the comment. We have updated the descriptions to accurately reflect the figures and provide more precise interpretations. We have revised the text on Lines 1664 to 1673 to clearly indicate that Fig. 4(a) displays accuracy variation and to ensure consistency between the figure and the main text. We have also reviewed the entire experimental section to improve clarity and ensure all references are accurate.
>
> **Weakness 3 Extended to partial participation of clients**
>
> Once the regularization strategy for $\lambda$ under data heterogeneity is determined, $\lambda$ remains fixed, and the optimization objective together with the estimators $\\{\widetilde{\boldsymbol{w}}\_i\\}\_{i = 1}\^m$ are fully defined by (1). In other words, the choice of $\lambda$  depends solely on the problem parameters, and the statistical error holds universally for any algorithm that solves the FedProx problem, regardless of whether client sampling is employed. On the computation side, our algorithm that uses full gradient update and full client participation to compute  $\\{\widetilde{\boldsymbol{w}}\_i\\}_{i = 1}\^m$ as the solution to (1) can potentially be extended to client partial participation.   If the participation is modeled as uniformly at random per communication round, we anticipate the convergence analysis to be similar to the extension from using full gradient update to unbiased mini-batch stochastic gradients. To validate the performance empirically, we report the statistical accuracy of the proposed algorithm under varying levels of client participation per round on the MNIST dataset in the table below.
>
> Other participation patterns, such as cyclically or even arbitrarily but just requiring all clients not to leave the system permanently, would require a different technical analysis or algorithmic design to avoid the bias introduced by partial participation.
>
> | Participation Rate   | One-stage FedProx(0.1)     | One-stage FedProx(0.5)    | One-stage FedProx(1)|   LocalTrain   |   GlobalTrain   |
> |--------------|--------------|--------------|--------------|--------------|--------------|
> | 0.5    | 0.7533      | 0.7653      | 0.7917      | 0.6612      |  0.8891   |
> | 0.8    | 0.8037      | 0.8123      | 0.8422      | 0.7011      |  0.8912   |
> | 1      | 0.8753      | 0.8893      | 0.9032      | 0.7311      |  0.9482   |

---

> > ### Comment · Reviewer_bMut · 2024-11-27
> >
> > Thank you for conducting additional experiments and providing updated experimental interpretations! These revisions effectively addressed my concerns about the paper. After carefully reviewing the manuscript and considering the feedback from other referees, I have decided to increase my score.

---

> > > ### Author Response · Authors · 2024-11-27
> > > **Response to Reviewer bMut**
> > >
> > > We sincerely thank the reviewer for the thoughtful comments and re-evaluating our work. We are grateful that the additional experiments and updated interpretations addressed your concerns, and we deeply appreciate your constructive feedback and score adjustment.

---

> ### Comment · Area_Chair_XAFr · 2024-11-25
>
> Dear Reviewer bMut,
>
> The author discussion phase will be ending soon. The authors have provided detailed responses. Could you please reply to the authors with whether they have addressed your concern and whether you will keep or modify your assessment on this submission?
>
> Thanks.
>
> Area Chair

---

### Official Review · Reviewer_KgRk · 2024-11-04

**Soundness:** 2
**Presentation:** 2
**Contribution:** 2
**Rating:** 5
**Confidence:** 2

**Summary:**

The FedProx algorithm enables personalization through penalization on the distances between local models and the global model.
This paper theoretically analyzes the regularization strength influence the statistical rate and show that FedProx can consistently outperform pure local training and achieve a nearly minimax-optimal rate by properly choosing the penalization quantity based on the relative size of heterogeneity $R$ and local sample size $n$.  Experiments on synthetic and MNIST datasets are given.

**Strengths:**

- The rates are reasonable and established under common assumptions in federated learning.
- The paper is written in a way that is easy to follow.

**Weaknesses:**

- It is claimed that FedProx achieves a nearly minimax-optimal statistical accuracy but it is not compared to other works like Scaffold where heterogeneity $R$ does not appear in the upper bound.

- Empirical results are only given for synthetic datasets and MNIST on logistic regression. Other personalization methods are not compared.

- The adaptive strategy of choosing of $\lambda$ relies on unknown quantity and therefore limits its contribution.

**Questions:**

N/A

---

> ### Author Response · Authors · 2024-11-22
> **Response to Reviewer KgRk (Part 1)**
>
> We sincerely thank the reviewers for the comments. Below, we address each point in detail:
>
> **Weakness 1 Comparison with other works like Scaffold**
>
> Scaffold computes the minimizer/stationary point of the finite sum minimization problem
> $$
> \widetilde{\boldsymbol{w}}\_{GT} \in \arg\min\_{\boldsymbol{w}} \frac{1}{m} \sum\_{i = 1}^m L\_i (\boldsymbol{w},S\_i).
> $$
>  It uses a variance reduction technique to eliminate the impact of gradient heterogeneity on the **algorithm convergence**, and consequently, the rate does not depend on $R$. Note the rate here is the algorithmic convergence rate, meaning how many iterations are needed to reach a solution that is $\varepsilon$-close to $\widetilde{\boldsymbol{w}}\_{GT}$. However, under model heterogeneity, $\widetilde{\boldsymbol{w}}\_{GT}$ can be far away from any of the  $\boldsymbol{w}_i^\star$s and the difference does depend on $R$ (cf. Lemma 1). On the other hand, our result directly quantifies the distance of the FedProx solution $\widetilde{\boldsymbol{w}}\_{i}$ to $\boldsymbol{w}\_i\^\star$. Their difference called the statistical rate/accuracy, is minimax optimal.
>
> We would like to clarify in this paper we are considering a personalized federated learning problem, where the major concern is the statistical accuracy for each client's model, i.e., the local statistical error specified in Line 229. Such an error would be naturally related to the statistical heterogeneity measure $R$. On the other hand, Scaffold are concerned with the statistical accuracy of the finite-sum problem and, therefore is not dependent on $R$. As the statistical problem concerned by Scaffold is different from ours, the error bound is not directly comparable.
>
>
> **Weakness 2 Empirical result**
>
> First we would like to note that in the original version, we have already considered both a convex setting (logistic regression) but also a non-convex setting (CNN). Thus, our analysis has already extended beyond logistic regression. We would also like to reiterate that the primary objective of our empirical study is to emphasize the role of personalization in objective (1). To this end, we primarily compare our approach with global training and purely local training under varying levels of data heterogeneity. We acknowledge that exploring other personalization methods aligned with objective (1) could provide additional insights into whether the influence of personalization is universally observed. For instance, we notice that pFedMe [2] focuses on the same objective (1) and can address non-iid data. However, the effect of personalization is not fully exploited in their work. We include their method in our numerical analysis and we refer the reviewer to Appendix C.2 for more details.
>
>
>
> **Weakness 3 Choice of $\lambda$ and  contribution**
>
> We would like to clarify the contribution of this paper is primarily theoretical: providing the performance guarantees FedProx--a well-known personalized FL method. With properly chosen penalty $\lambda$, FedProx **can** achieve a minimax optimal statistical rate. This performance limit was unknown before, and our result improves the state-of-the-art [1] upperbound by (i) enlarging the family of applicable loss functions and (ii) obtaining a tighter rate. Indeed, the choice of $\lambda$ depends to some degree on $R$ (through the cut-off threshold). Such a dependency is intuitive as one should rely on $R$ to decide to what extent the $\boldsymbol{w}_i$s should be pulled together, and also exist in [1] as well as related works [4][5]. How to choose $\lambda$ in a purely data-driven way while not degrading the statistical rate is an important, very challenging problem. We view this as a natural next step for future work, while the current study focuses on addressing the foundational question of whether FedProx serves as an effective personalization strategy under model heterogeneity in the idealized scenario where $R$ is known.
>
>
> -------------------------------
> [1] Chen, S., Zheng, Q., Long, Q., and Su, W. J. (2023). Minimax Estimation for Personalized Federated Learning: An Alternative between FedAvg and Local Training? Journal of Machine Learning Research, 24(262), 1-59.
>
> [2] T Dinh, C., Tran, N., and Nguyen, J. (2020). Personalized federated learning with Moreau envelopes. Advances in Neural Information Processing Systems, 33, 21394-21405.
>
> [3] Li, T., Sahu, A. K., Zaheer, M., Sanjabi, M., Talwalkar, A., and Smith, V. (2020). Federated optimization in heterogeneous networks. Proceedings of Machine Learning and Systems, 2, 429-450.
>
> [4] Li S, Zhang L, Cai T T, et al. Estimation and inference for high-dimensional generalized linear models with knowledge transfer[J]. Journal of the American Statistical Association, 2024, 119(546): 1274-1285.
>
> [5] He, Zelin, Ying Sun, and Runze Li. "Transfusion: Covariate-shift robust transfer learning for high-dimensional regression." International Conference on Artificial Intelligence and Statistics. PMLR, 2024.

---

> > ### Comment · Reviewer_KgRk · 2024-11-26
> > **Reply**
> >
> > I acknowledge that I have read authors' reply. The main concern for me to give a "6" is that the analysis over FedProx for PFL is not that interesting. I am quoting the author of both FedProx and Ditto about the difference between two algorithms:
> >
> > > Remark (Relation to FedProx). We note that the $L_2$ term in Ditto bears resemblance to FedProx, a method which was developed to address heterogeneity in federated optimization (Li et al., 2020d). However, Ditto fundamentally differs from FedProx in that the goal is to learn personalized models vk, while FedProx produces a single global model w. For instance, when the regularization hyperparameter is zero, Ditto reduces to learning separate local models, whereas FedProx would reduce to FedAvg. In fact, Ditto is significantly more general than FedProx in that FedProx could be used as the global model solver in Ditto to optimize G(·). As discussed above, other regularizers beyond the L2 norm may also be used in practice.
> >
> > Note that Ditto is published 3 years ago and FedProx is published 6 years, so they are not that "recent". Personally I am not that excited about the analysis of PFL aspect of FedProx (which is superseded by Ditto). I am not sure how much value it would contribute to the community and I would leave AC to decide if this is a valid concern.

---

> > > ### Author Response · Authors · 2024-11-26
> > > **Response to Reviewer KgRk**
> > >
> > > We sincerely thank the reviewer for carefully considering our new results and for expressing a willingness to potentially reconsider the score. Below, we further address the reviewer’s remaining concerns regarding the contributions of this work.
> > >
> > > **Novelty and Timeliness**
> > >
> > > Although FedProx was introduced in 2018, its rigorous statistical studies were recent, with key advancements established in 2023 [1,2]. Even in these works, the story is incomplete, with strong assumptions such as linear models and boundedness of the loss imposed, and its statistical optimality unknown. As a theoretical work, we addressed a critical gap in the field.
> > >
> > >
> > >
> > > **Difference between FedProx[3] and Ditto[4]**
> > >
> > > The general formulation of Ditto is given by
> > > $$
> > >     \min\_{v_k} h\_k(v\_k;w\^\star) := F\_k(v\_k) + \frac{\lambda}{2} \\| v\_k - w\\|^2, \quad {\rm s.t.}\  w\^\star \in \mathop{\rm argmin}\_{w} \quad G(F_1(w), \ldots, F\_K(w)).
> > > $$
> > > whereas FedProx is
> > > $$
> > >     \min\_{\\{v\_k\\}\_{k = 1}^K, w} \left( \frac{1}{K} \sum\_{k = 1}\^K F\_k (v\_k) + \frac{\lambda}{2} \\|w - v\_k\\|^2 \right).
> > > $$
> > > Note that this is not the objective as stated in [3], but the one Algorithm 2 therein solves under a constant step size and penalty $\lambda$.
> > >
> > > Both formulations produce a global model $w$ and local models $\\{v_k\\}_{k = 1}^K$. By setting $\lambda = 0$, Ditto degenerates to pure local training, and so does FedProx. It is also not immediate to us that Ditto is more general because $G$ is a composite function of $w$ and does not match the objective of FedProx, which involves both $w$ and $v_k$ as variables. The theoretical study of Ditto is restricted to linear models with orthogonal features, $F_k$ being the quadratic loss, and $G$ being the average of its arguments. For the general choice of $G$ and $F$, the statistical guarantee of Ditto is unknown. As such, it is questionable to assert that Ditto is a method that supersedes FedProx.
> > >
> > > Nonetheless, we want to emphasize that the goal of our work is to investigate the **statistical properties** of FedProx, which is a valid, not fully understood question in its own right. Admittedly, there are other personalized FL formulations and algorithms that could outperform FedProx empirically under different simulation settings. Yet, the point of this work is not to prove that FedProx is universally better than all of them (if possible). Rather, the theoretical analysis we developed for FedProx serves as a new perspective that one could take to study the statistical guarantees of regularization-based personalized federated learning methods, for example, generalizing the analysis of Ditto. It also provides the potential to compare and offer guidelines for algorithm recommendations for different problem settings.
> > >
> > >
> > > **Ongoing Relevance of the Objective in Equation (1)**
> > >
> > > Although FedProx and Ditto were published several years ago, they have continued to inspire significant follow-up work in both empirical and theoretical domains, demonstrating their enduring importance in the field. For instance, recent research [5] in 2022 introduced a novel local dissimilarity-invariant convergence theory for FedProx, and recent works [6] in 2024 further refined its theoretical foundations and explored applications in complex federated systems. These advancements highlight the **foundational role of the objective in Equation (1)** in advancing personalized federated learning.
> > >
> > > **Generalizable and Deeper Insights into Regularization-Based PFL**
> > >
> > > The primary aim of our work is to investigate the **statistical properties** of regularization-based personalized federated learning (PFL) objectives, rather than focusing solely on the specific algorithm introduced in [1]. Our analysis extends beyond FedProx to provide insights applicable to a broader class of regularization-based PFL methods. For example, as the reviewer noted, Ditto also falls within this family. We have revised our manuscript to shed light on the point. Additionally, recent approaches [7][8][9] that employ regularization share the same structural foundation as the objective in Equation (1). The analytical techniques we develop are thus generalizable, offering new perspectives for understanding these methods.
> > >
> > > We also wish to emphasize the distinction between the convergence guarantees for optimization algorithms and the statistical guarantees, which is a focus of our work. Most prior studies establish convergence to the optimal solution of Equation (1) but do not examine how this solution relates to the true model parameters (ground truth). Our work fills this gap by providing a **fine-grained analysis of the statistical accuracy** under varying data heterogeneity, bridging the gap between the optimal solution of (1) and the true model parameters. This perspective offers a deeper understanding of regularization-based PFL.

---

> > > ### Author Response · Authors · 2024-11-26
> > > **Response to Reviewer KgRk**
> > >
> > > We deeply appreciate the reviewer’s thoughtful feedback and time. We hope this response provides additional clarity regarding the significance and contributions of our work and kindly invite the reviewer to re-evaluate our submission in light of these clarifications.
> > >
> > > ----------------------
> > >
> > >
> > > [1] Cheng, Gary, Karan Chadha, and John Duchi. "Federated asymptotics: a model to compare federated learning algorithms." International Conference on Artificial Intelligence and Statistics. PMLR, 2023.
> > >
> > >
> > > [2] Chen, Shuxiao, et al. "Minimax Estimation for Personalized Federated Learning: An Alternative between FedAvg and Local Training?." Journal of Machine Learning Research 24.262 (2023): 1-59.
> > >
> > > [3] Li T, Sahu A K, Zaheer M, et al. Federated optimization in heterogeneous networks[J]. Proceedings of Machine learning and systems, 2020, 2: 429-450.
> > >
> > > [4] Li T, Hu S, Beirami A, et al. Ditto: Fair and robust federated learning through personalization[C]//International conference on machine learning. PMLR, 2021: 6357-6368.
> > >
> > >
> > > [5] Yuan X, Li P. On the convergence of FedProx: Local dissimilarity invariant bounds, non-smoothness and beyond[J]. Advances in Neural Information Processing Systems, 2022, 35: 10752-10765.
> > >
> > >
> > > [6] Li H, Richtárik P. On the Convergence of FedProx with Extrapolation and Inexact Prox[J]. arXiv preprint arXiv:2410.01410, 2024.
> > >
> > >
> > >
> > > [7] Hanzely, Filip, and Peter Richtárik. "Federated learning of a mixture of global and local models." arXiv preprint arXiv:2002.05516 (2020).
> > >
> > >
> > >
> > > [8] T Dinh, Canh, Nguyen Tran, and Josh Nguyen. "Personalized federated learning with Moreau envelopes." Advances in neural information processing systems 33 (2020): 21394-21405.
> > >
> > >
> > > [9] Acar D A E, Zhao Y, Navarro R M, et al. Federated learning based on dynamic regularization[J]. arXiv preprint arXiv:2111.04263, 2021.

---

> ### Comment · Area_Chair_XAFr · 2024-11-25
>
> Dear Reviewer KgRk,
>
> The author discussion phase will be ending soon. The authors have provided detailed responses. Could you please reply to the authors with whether they have addressed your concern and whether you will keep or modify your assessment on this submission?
>
> Thanks.
>
> Area Chair

---

> ### Author Response · Authors · 2024-11-30
> **Official Comment by Authors**
>
> Dear Reviewer,
>
> As the author-reviewer discussion period will end soon, we will appreciate it if you could check our response to your review comments. This way, if you have further questions and comments, we can still reply before the author-reviewer discussion period ends. If our response resolves your concerns, we kindly ask you to consider raising the rating of our work. Thank you very much for your time and efforts!

---

### Official Review · Reviewer_iwVj · 2024-11-06

**Soundness:** 1
**Presentation:** 1
**Contribution:** 1
**Rating:** 1
**Confidence:** 5

**Summary:**

The paper studies the FedProx learning rule for multi-distribution learning. It establishes statistical rates as a function of first-order heterogeneity, showing how the regularization (to form consensus) should be set in a heterogeneity-aware manner. The authors also provide an algorithm to solve the empirical FedProx objective, providing its convergence analysis in the noise-less setting. Some experiments are also provided to verify the theory's predictions.

**Strengths:**

The paper identifies an interesting problem in federated learning: the issue of personalizing more v/s less as a function of data heterogeneity.

**Weaknesses:**

**Incorrect units**: Theorem 1 seems wrong, as stated. In particular, the units of the convergence rate are incorrect, i.e., the r.h.s. does not have the units of the squared parameters. For instance, if we blow up both the parameter space and function space by the same factor but very large factor, then r.h.s will grow much more quickly because of the $C_2$ and $C_4$ terms that will blow up due to the additional $\sqrt{1+2\mu}$ factor as opposed to say $\rho/\mu$ which will grow with the same rate as the l.h.s. To put it succinctly, the result is not scale invariant, and the authors are likely either hiding some boundedness assumptions or omitting important problem parameters in the Theorem statement. Even the choice of $\lambda$ is incorrect. $\lambda$ should have the same units as strong convexity, but it does not. It is unclear what the indicator function does in the choice of $\lambda$ because comparing $R$ with a unitless/scaleless quantity such as $1/\sqrt{n}$ makes no sense. For another sanity check, note that when each client's objective is the mean estimation objective (i.e., mean squared error of the estimator), $\mu=1$ and only dependence on the right in (4) should be $\rho^2$ and not $\rho^4$. I request the authors to correct this.

In light of the above issue, it is hard for me to interpret the convergence result, which is the main result of this paper, but I will try to disambiguate what the correct convergence rate should have been.

**Vacuous guarantees**: Let's consider (5) for the particular case of mean estimation so that $\mu=1$. I claim that the upper bound in (5) is worse than the best of two trivial algorithms: (i) estimate the mean using only the samples on machine $i$ (i.e., pure local training), (ii) estimate the mean using all the samples across all the machines (i.e., consensus optimization). To see this, consider the first and third terms of (5), which, when combined, simplify to:$$\min(\frac{\rho^2}{mn} + R^2, \frac{\rho^2}{n}),$$
which is precisely the best of simple algorithms (i) and (ii). Since (5) has another term in the upper bound, the guarantee of Theorem 1 can not beat this trivial baseline and is thus vacuous. Furthermore, the best of (i). (ii) is extremely simple to implement: do the usual federated learning and pure local training side by side by splitting the samples in half. Then, compare these two solutions for each machine and pick the best one.

**Noiseless setting**: It is tough to justify the noiseless setting of the optimization result because, in the absence of noise, there is no benefit of collaboration whatsoever, and even the Theorem statement in the paper suggests setting $\lambda=0$ to do purely local training. Thus, studying optimization with a complex procedure in such a setting is a moot point. The authors brush this away, speaking as if dealing with noise were trivial and techniques like variance reduction can be added without much effort to their algorithm. That does not seem to be the case.

In light of the above issues, I strongly recommend rejecting the work, as it does not meet the rigor requirements for publication at ICLR.

**Questions:**

1. Why is $w_\star^{(g)}$ the appropriate true global solution in the heterogeneous setting? It seems the authors are conflating between the average of optimas and the optima of the average function. In federated learning, the aim is usually to recover the latter, and both these solutions can be further apart. To see the difference between these two solutions, see the section on fixed point discrepancy in [Patel et al.](https://arxiv.org/abs/2405.11667).

2. It is unclear what is $\tilde w^{(i)}$; what do the authors mean by local models? Do they use this to refer to each machine's ERM solutions, which are unique given the strong convexity?

---

> ### Author Response · Authors · 2024-11-22
> **Response to Reviewer iwVj (Part 1)**
>
> We sincerely thank the reviewer for the thoughtful comments, which have inspired us to make substantial improvements to our previous results. Our current result proves FedProx achieves minimax optimality by improving the upper bound from $O\left(\frac{1}{mn} + R^2 \wedge \frac{1}{n} + \frac{R}{\sqrt{n}} \wedge \frac{1}{n}\right) $ in the original submission to $O\left(\frac{1}{mn} + R^2 \wedge \frac{1}{n}\right)$. In addition, we have carefully refined the proofs to address concerns related to units and clarity. Below, we provide detailed responses to each comment:
>
>
> **Incorrect units**
>
> We first discuss this issue based on the results of our original submission.
> In our previous analysis, our focus was on how sample and client number affect the accuracy and considered the limit that $n$, $m$, and $R$ can go to infinity. Problem parameters such as $\mu$ and $L$ were treated as fixed constants. So when choosing $\lambda$ we did not account for the scenario mentioned by the reviewer where $\mu$, $L$, or $\rho$ are also allowed to diverge. Our original choice of $\lambda$ still makes sense if we aim to interpret the result w.r.t. $n$, $m$, and $R$. The comparison of $R$ with $1/\sqrt{n}$ is meaningful, as $1/\sqrt{n}$ represents the critical threshold where the statistical rate transitions from $R^2$ to $1/n$. This transition aligns with the point at which purely local training becomes the optimal strategy. Such a discussion is also reflected in existing literature [1]. Regarding the second sanity check, we have double-checked that the right-hand side of (4), which depends on $\rho$ through $C_1$ and $C_2$ defined respectively as
> $C_1 = \rho^2/\mu^2$ and $C_2 = (\frac{(1 + \sqrt{1 + 2 \mu}) \rho}{2 \mu})^2 \vee (2 \rho /\mu)$, and they
> do not contain any term involving $\rho^4$. Still, we appreciate the reviewer pointing it out and we take it into account in our new analysis. Such a unit issue can be addressed by a modification on the choice of $\lambda$, with the original proof staying valid and both $\lambda$ and the error bound having the correct units.
>
>
>
>
> **Vacuous guarantees**
>
>
>
> We appreciate the reviewer's question and the mean estimation example, which have inspired us to further improve our analysis and the result. The new version of Theorem 1 states if
> $\lambda  \geq \max \left\\{64\kappa^2L, \left(2\kappa\vee5\right)\mu \frac{2L^2R^2 + \rho^2/n}{L^2R^2+\rho^2/N}\right\\} -\mu$ when $R\leq \frac{1}{\sqrt{n}}$, and $ \lambda \leq\frac{\rho^2}{n\mu R^{2}} $ when $R > \frac{1}{\sqrt{n}}$, then we have
> $$\mathbb{E}\left\\|\tilde{\boldsymbol{w}}^{(g)} -\boldsymbol{w}^{(g)}_{\star} \right\\|^2 \leq \frac{C_1}{mn} + C_2 \left(\frac{1}{n}\wedge R^2 \right), \
>         \mathbb{E}\left\\| \widetilde{\boldsymbol{w}}\^{(i)} -\boldsymbol{w}\^{(i)}\_{\star} \right\\|^2 \leq   \frac{C_3}{mn}+ C_4\left( \frac{1}{n}\wedge R^2\right) \quad \text{ for all }i \in [m],$$
> $$ C_1  =  32\frac{\rho^2}{\mu^2}, C_2 = \big(\frac{(1+\sqrt{3})\rho}{\mu}\big)^2\vee \frac{32 L^2}{\mu^2}, C_3  = \frac{192\rho^2}{\mu^2}, C_4   = \left(\frac{\rho(1 + (4C_2+5)^{\frac{1}{2}})}{\mu}\right)^2  \vee 198\kappa^2.$$
> This new upper bound is  $O\left(\frac{1}{mn} + R^2 \wedge \frac{1}{n}\right)$ and matches the rate of the dichotomous strategy proposed by the reviewer across any level of statistical heterogeneity $R$. Both the choice of $\lambda$ and the error bounds have the correct units. Furthermore, we note that the rate $O\left(\frac{1}{mn} + R^2 \wedge \frac{1}{n}\right)$ is in fact the *information-theoretic lower bound* for the estimation problem (1), as established in prior work [1]. Thus, our new result demonstrates that FedProx is *minimax optimal*.
>
> We may reiterate that the key contribution of our work is analyzing the performance of FedProx, a method that has demonstrated great empirical success [2] and whose statistical guarantees were not fully understood.
> This is an important contribution in its own right, even though the rate matches the dichotomous strategy suggested by the reviewer. It remains an open question whether FedProx can outperform this ‘simple baseline’—a baseline that is already minimax optimal, and the point of this paper is not to prove that Fedprox must be better over the baseline. The latest work [1] provided the estimation error of FedProx under more restricted problem assumptions and is larger even compared to the result in our original submission. In [1], it was noted that the minimax optimality of FedProx was an open question (see the discussion at the end of Section 3). In this work, we addressed and closed this gap by establishing its minimax optimality. As a side note, if one seeks to understand why FedProx could be advantageous over the dichotomous strategy in practice, we provide an additional discussion and numerical comparison (See Table 4) in Appendix C.3.

---

> ### Author Response · Authors · 2024-11-22
> **Response to Reviewer iwVj (Part 2)**
>
> **Noiseless setting**
>
> First, we would like to clarify that the "noiseless setting in optimization" refers to using full gradient update but not mini-batch stochastic gradient when solving the FedProx objective (1). It is not the same concept as $\rho =0$, which means the empirical loss $L_i$ is equal to the population loss. In the practical case with $\rho \neq 0$, quantifying how to choose $\lambda$ under data heterogeneity in theoretical analysis is precisely the main contribution of the paper. Under such a choice of $\lambda$, the section on the optimization provides the computation and communication complexity analysis of the proposed algorithm. In particular, our Corollary 1 shows the communication complexity is $\mathcal{O} (\kappa (\lambda + \mu)/(\lambda + L) \log 1/\varepsilon)$. We want to emphasize that this result **does not** "suggest setting $\lambda = 0$ to do purely local training". Instead, our point is that one must first choose the right $\lambda$ based on model heterogeneity to define the right problem to solve (whose optimal solution has high statistical accuracy), as suggested by Theorem 1, and then run an algorithm to find the solution at low costs, whose guarantee is provided by Corollary 1. Even though setting $\lambda = 0$ will require the lowest communication cost algorithmically, it is pointless if the solution generated by the algorithm is far from the ground truth.
>
> In addition, we also want to clarify we have *never* claimed that extension to use stochastic gradients is trivial, but just meant to say proving convergence is potentially doable. For example, [3] considered the analysis of bilevel algorithms in the deterministic case, while stochasticity and variance reduction have been studied in, e.g., [4,5]. That said, we fully acknowledge that incorporating stochastic noise and analyzing its interaction with the deterministic part in the rate analysis requires significant effort. However, our main focus is the effect of personalization on statistical accuracy and communication efficiency. To avoid misunderstanding, we have revised the statement in the manuscript.
>
>
>
>
> **Question 1**
>
> In this work, we call $\boldsymbol{w}\^{(g)}\_{\star}$ the global model and **defined** it to be the average of the $\boldsymbol{w}\_\star^{(i)}$'s. Consistently, our analysis quantifies the difference between $\tilde{\boldsymbol{w}}\^{(g)}$, which is the solution of the FedProx problem defined by (1), and $\boldsymbol{w}\^{(g)}\_{\star}$, as a function of sample size and model heterogeneity $R$. Such a $\boldsymbol{w}\^{(g)}\_{\star}$ represents the population-level counterpart of $\tilde{\boldsymbol{w}}\^{(g)}$.  Similar notation has been used in multiple prior works [1][6] for their analysis. Throughout the paper, we have never indicated that $\boldsymbol{w}\^{(g)}\_{\star}$ or $\tilde{\boldsymbol{w}}\^{(g)}$ should be "the optima of the average function". Indeed, FedAvg aims to recover the optima of the average function, which is defined as the global model therein (we denote it $\boldsymbol{w}\_{GT}\^\star$). However, it is not a consensus that the terminology ``global model'' means exclusively $\boldsymbol{w}\_{GT}\^\star$, and we have never made or used the wrong claim $\boldsymbol{w}\^{(g)}\_{\star} = \boldsymbol{w}\_{GT}\^\star$ throughout this work.
>
>
> **Question 2**
>
> The local models $\widetilde{\boldsymbol{w}}\^{(i)}$, along with $\widetilde{\boldsymbol{w}}\^{(g)}$ are defined as the optimal solutions to the FedProx problem (1).
>
> We sincerely hope that the above clarifications address the reviewer’s concerns, and we would be more than happy to provide additional details if needed.
>
> ---------------------------------
> [1] Chen, S., Zheng, Q., Long, Q., and Su, W. J. (2023). Minimax Estimation for Personalized Federated Learning: An Alternative between FedAvg and Local Training? Journal of Machine Learning Research, 24(262), 1-59.
>
> [2] Li, T., Sahu, A. K., Zaheer, M., Sanjabi, M., Talwalkar, A., and Smith, V. (2020). Federated optimization in heterogeneous networks. Proceedings of Machine Learning and Systems, 2, 429-450.
>
> [3] Ji, K., Liu, M., Liang, Y., and Ying, L. (2022). Will bilevel optimizers benefit from loops. Advances in Neural Information Processing Systems, 35, 3011-3023.
>
> [4] Dagréou, M., Ablin, P., Vaiter, S., and Moreau, T. (2022). A framework for bilevel optimization that enables stochastic and global variance reduction algorithms. Advances in Neural Information Processing Systems, 35, 26698-26710.
>
> [5] Chen, X., Xiao, T., and Balasubramanian, K. (2024). Optimal algorithms for stochastic bilevel optimization under relaxed smoothness conditions. Journal of Machine Learning Research, 25(151), 1-51.
>
> [6] Duan Y, Wang K. Adaptive and robust multi-task learning. The Annals of Statistics, 2023, 51(5): 2015-2039.

---

> > ### Comment · Reviewer_iwVj · 2024-12-02
> >
> > I thank the authors for improving parts of their paper.
> >
> > > We may reiterate that the key contribution of our work is analyzing the performance of FedProx, a method that has demonstrated great empirical success [2] and whose statistical guarantees were not fully understood. This is an important contribution in its own right, even though the rate matches the dichotomous strategy suggested by the reviewer.
> >
> > I disagree with this. I do not believe it is an interesting question to give an analysis of FedProx in a regime where it only matches the simple baseline. Even if the authors' goal is to bridge a theory v/s experiment gap, the interesting result would be to identify assumptions and provide an analysis where FedProx can strictly beat this simple baseline, because in practice it probably does (if it does not, then there is no reason to use FedProx is practice). For inspiration, see [this](https://arxiv.org/pdf/2002.11684) and [this](https://arxiv.org/pdf/2205.13692), where the goal is to understand a shred assumption between the tasks where the simple baselines can be dominated.
> >
> > > In addition, we also want to clarify we have never claimed that extension to use stochastic gradients is trivial, but just meant to say proving convergence is potentially doable.
> >
> > Proving convergence might be doable, but retaining any benefits of the algorithm (in the analysis) while doing so might not be: precisely, why I think it is not reasonable to consider the noiseless regime.
> >
> > Regarding Question 1, I understand that it is unclear if the optima of the average function is the ideal thing to recover, but at least it is well motivated in over-parameterized regimes where clients (approximately) share some optima. Recovering the average of some optimas on each machine does not seem to have any motivation at all...
> >
> > To summarize, I am glad the authors understand my key criticism, but their improvements do not resolve the issue I pointed out. To rephrase, I do not think the problem being studied in this paper is interesting to either drive forward theory or practice.

---

> ### Comment · Area_Chair_XAFr · 2024-11-25
>
> Dear Reviewer iwVj,
>
> The author discussion phase will be ending soon. The authors have provided detailed responses. Could you please reply to the authors with whether they have addressed your concern and whether you will keep or modify your assessment on this submission?
>
> Thanks.
>
> Area Chair

---

> ### Author Response · Authors · 2024-11-30
> **Official Comment by Authors**
>
> Dear Reviewer,
>
> As the author-reviewer discussion period will end soon, we will appreciate it if you could check our response to your review comments. This way, if you have further questions and comments, we can still reply before the author-reviewer discussion period ends. If our response resolves your concerns, we kindly ask you to consider raising the rating of our work. Thank you very much for your time and efforts!

---

> ### Comment · Area_Chair_XAFr · 2024-12-01
>
> Dear Reviewer iwVj,
>
>
> Since you gave the most negative review and the authors have provided detailed responses to try to address your concerns, it is very important to hear your opinions before we can recommend a decision on this paper. Please reply with whether you think the authors have addressed (some of) your concerns and whether you will keep or modify your assessment.
>
> Thank you again.
>
> Area Chair

---

> > ### Comment · Reviewer_iwVj · 2024-12-02
> >
> > I have just replied, sorry for the delay. I will keep my score.

---

### Author Response · Authors · 2024-11-22
**Response to All Reviewers**

We sincerely thank the reviewers for their thoughtful and constructive feedback. Below, we summarize the key improvements made to our manuscript.

1. **Tight Upper Bound and Minimax Optimality** : We have refined the theoretical analysis and established that with a proper choice of the tuning parameter $\lambda$, the solution of FedProx achieves a matching upper bound with the minimax lower bound of the estimation problem, thereby demonstrating the *minimax optimality* of the FedProx method. See Theorem 1 for the new upper bound and Lemma 1 for a discussion on the technical contribution. Additionally, we have refined the choice of $\lambda$ to ensure that the tuning parameter and the resulting statistical bounds are both interpretable and expressed with correct units.


2. **Improved Numerical Studies** : We have expanded our experiments to include more datasets, applied more complex models, and provided a comparison with more baseline methods, including the dichotomous strategy suggested by reviewer $iwVj$, as well as other regularization-based personalized federated learning algorithms. Our results demonstrate the effect of personalization on the statistical accuracy of personalized federated learning method, further validating the theory.

We invite the reviewers to refer to the revised manuscript for further details on these additions.

---

### Meta-Review · Area_Chair_XAFr · 2024-12-13

**Metareview:**

This paper establishes that, by appropriately setting the tuning parameter, the FedProx method achieves an error upper bound that matches the minimax lower bound. This demonstrates the minimax optimality of the FedProx method under the considered setting (smooth and strongly convex).

The primary criticism is that the dichotomous method, a simple baseline, can also achieve the same minimax optimal error bound. One reviewer argued that this undermines the novelty of the paper, as it does not explain why FedProx performs better than the dichotomous method in numerical experiments. The reviewer suggested that the authors explore a different setting where FedProx theoretically outperforms the dichotomous method in terms of the upper bound.

The authors counter that proving the minimax optimal upper bound for FedProx is valuable in itself, as it addresses a gap in understanding the theoretical guarantees of this widely used federated learning method.

Both the reviewer and the authors make valid points, making this decision challenging. I believe the paper does contribute to the field, as the authors claim. However, the fact that the same bound can be achieved by a baseline method diminishes the significance of the result. **If the paper fills an existing gap, it should better justify why this gap is important or interesting.** Specifically, the smooth and strongly convex assumption is relatively strong from an optimization perspective and may not align with the practical scenarios where FedProx is applied and has demonstrated success. Why is this setting a meaningful or interesting one for analyzing the error bound of FedProx? I recommend that the authors either provide a stronger justification for this setting or consider alternative settings where FedProx outperforms the baseline.

One additional minor concern I have is that the minimax optimal error bound was proven in the revised version rather than in the original submission. While the authors have highlighted the changes in blue, I am concerned that the review team may not have had sufficient opportunity to thoroughly evaluate these updates.

Overall, I am slightly inclined toward rejection. A good theory should ideally help explain practical phenomena. However, in this paper, the contribution may not be significant enough to guarantee acceptance. That said, I believe the score 1 given by one reviewer is unfairly low, as the paper does have merit.

**Additional Comments On Reviewer Discussion:**

Reviewer iwVj raised a good point that a simple baseline can achieve the same error bound shown in this paper. This is the main reason I recommend rejection. However, I think the score 1 given by this reviewer is unfairly low as this paper still have some value.

Reviewer KgRk and Reviewer bMut both have concerns on the empirical results which are addressed by the authors. Reviewer 5muf gave the highest score and had almost no concern.

Reviewer han2's main concern is on the theoretical comparison between this paper and the previous works. The authors addressed his/her concern in the revision so Reviewer han2 raise the score.

---

### Decision · Program_Chairs · 2025-01-22

Reject